# Armijo Line-search Can Make (Stochastic) Gradient Descent Provably Faster

**Sharan Vaswani** [1]   **Reza Babanezhad** [2]

## Abstract

Armijo line-search (Armijo-LS) is a standard method to set the step-size for gradient descent (GD). For smooth functions, Armijo-LS alleviates the need to know the global smoothness constant $L$ and adapts to the "local" smoothness, enabling GD to converge faster. Existing theoretical analyses show that GD with Armijo-LS (GD-LS) can result in constant factor improvements over GD with a $1/L$ step-size (denoted as GD(1/L)). We strengthen these results and show that if the objective function satisfies a certain non-uniform smoothness condition, GD-LS can result in a faster convergence rate than GD(1/L). In particular, we prove that for convex objectives corresponding to logistic regression and multi-class classification, GD-LS can converge to the optimum at a linear rate, and hence improves over the sublinear convergence of GD(1/L). Furthermore, for non-convex objectives satisfying gradient domination (e.g. those corresponding to the softmax policy gradient in RL or generalized linear models with a logistic link function), GD-LS can match the fast convergence of algorithms tailored for these specific settings. Finally, we analyze the convergence of stochastic GD with a stochastic line-search on convex losses under the interpolation assumption.

## 1. Introduction

Gradient descent (GD) (Cauchy et al., 1847) and its stochastic variants are the preferred optimization methods in machine learning. The practical effectiveness of gradient methods heavily relies on the choice of the step-size ("learning rate") parameter. Backtracking Armijo line-search (Armijo, 1966; Nocedal & Wright, 2006) (referred to as Armijo-LS) is a standard method to set the step-size for gradient descent.

Given an initial step-size, the simplest form of Armijo-LS "searches" for the largest step-size that guarantees a sufficient decrease in the function value. When minimizing $L$-smooth functions using GD, Armijo-LS alleviates the need to know $L$, the global smoothness constant and enables setting the GD step-size in an adaptive manner. For both $L$-smooth convex and non-convex functions, GD with Armijo-LS (henceforth GD-LS) has been shown to match the favorable theoretical guarantees of GD with a constant $1/L$ step-size (henceforth GD(1/L)). However, empirically, GD-LS typically results in faster convergence and is consequently, the default choice in practice.

One often-cited reason to explain the faster convergence of GD-LS is that it adapts to the "local" smoothness constant $L(\theta)$ near the point $\theta$, and results in an effective step-size of $1/L(\theta)$. In some regions, $L(\theta)$ might be much smaller than the global smoothness $L$, thus allowing GD-LS to use much bigger step-sizes and consequently lead to faster convergence. Existing works that define notions of local smoothness (Scheinberg et al., 2014; Lu & Mei, 2023; Mishkin et al., 2024; Fox & Schmidt, and references there in) can be used to formalize this intuition and show that GD-LS can result in constant factor improvements over GD(1/L), while having the same rate of convergence. In contrast, we consider a special class of objective functions and show that GD-LS (with a large initial step-size) can result in a *provably faster rate of convergence* compared to GD(1/L). In particular, we make the following contributions.

**Contribution 1.** In Sec. 2, we introduce a class of functions that satisfy an $(L_0, L_1)$ non-uniform smoothness condition. In particular, we consider functions where the local smoothness constant around a point $\theta$ is given by $L(\theta) = L_0 + L_1 f(\theta)$ where $L_0$ and $L_1$ are non-negative constants (and $L_1 = 0$ corresponds to the standard uniform smoothness). We show that the proposed condition is satisfied by common objectives; for example, both the logistic and exponential losses used for linear classification satisfy the condition with $L_0 = 0$ and $L_1 \neq 0$. Furthermore, we prove that this condition is also satisfied by non-convex functions corresponding to generalized linear models with a logistic link function, and the softmax policy gradient objective in reinforcement learning.

This condition is similar to that proposed in Zhang et al.

[1]Simon Fraser University [2]Samsung AI, Montreal. Correspondence to: Sharan Vaswani <vaswani.sharan@gmail.com>, Reza Babanezhad <babanezhad@gmail.com>.

*Proceedings of the $42^{nd}$ International Conference on Machine Learning*, Vancouver, Canada. PMLR 267, 2025. Copyright 2025 by the author(s).

(2019) to explain the success of normalization and gradient clipping when training neural networks. The difference is that we consider problems where the smoothness is proportional to $f(\theta)$ rather than $\|\nabla f(\theta)\|$ (Zhang et al., 2019; 2020; Chen et al., 2023). Furthermore, in Sec. 2, we show that non-negative, twice-differentiable functions satisfying the non-uniform smoothness condition in Zhang et al. (2019) satisfy the proposed condition.

**Contribution 2.** In Sec. 3, we analyze the convergence of GD-LS on functions satisfying the proposed conditions. We first prove that the step-size selected by GD-LS around $\theta$ is lower-bounded by $1/L(\theta)$, and hence, GD-LS provably adapts to the local smoothness. We use this to prove Thm. 1, a meta-theorem that quantifies the convergence rate of GD-LS for both convex and non-convex objectives.

**Contribution 3.** In Sec. 4, we instantiate Thm. 1 for non-uniform smooth, convex losses that include logistic regression and multi-class classification with the cross-entropy loss as examples. Specifically, in Cor. 1, we show that GD-LS (with a sufficiently large initial step-size) converges at an $O\left((f^*/\epsilon) \ln(1/\epsilon)\right)$ rate where $f^* := \inf f(\theta)$. Hence, when $f^*$ is $\Theta(\epsilon)$, GD-LS converges at an $O\left(\ln(1/\epsilon)\right)$ rate, compared to the sublinear $O\left(1/\epsilon\right)$ convergence of GD(1/L). We further instantiate this result for logistic regression on linearly separable data, and prove the linear convergence of GD-LS (Cor. 2), thus matching the rate for normalized GD (Axiotis & Sviridenko, 2023). In App. C, we show that GD with the Polyak step-size (Polyak, 1987) can inherit this fast convergence for logistic regression.

In comparison to our results, most work exploiting non-uniform smoothness (Zhang et al., 2019; 2020; Koloskova et al., 2023; Chen et al., 2023; Li et al., 2023) requires the knowledge of the corresponding problem-dependent constants. Notable exceptions include Hübler et al. (2024) and (Vankov et al., 2024; Takezawa et al., 2024; Gorbunov et al., 2024) that consider GD-LS and the Polyak step-size respectively, and aim to minimize the class of non-uniform smooth functions in (Zhang et al., 2019). In particular, the work in Hübler et al. (2024) considers general non-convex functions and does not demonstrate the algorithm's adaptivity to the smoothness, nor does it result in a faster rate than GD(1/L). Moreover, their resulting algorithm requires the knowledge of the non-uniform smoothness constant, making it impractical. On the other hand, (Vankov et al., 2024; Takezawa et al., 2024; Gorbunov et al., 2024) consider minimizing convex, non-uniform smooth functions using GD with the Polyak step-size. In Sec. 4, we show that GD-LS (without any modification) can achieve a similar convergence guarantee as these works.

**Contribution 4.** In Sec. 5, we instantiate Thm. 1 for non-convex functions satisfying non-uniform smoothness and gradient domination conditions that guarantee global opti-

mality. Specifically, in Sec. 5.1, we analyze the convergence of GD-LS for the softmax policy gradient objective in reinforcement learning (Mei et al., 2020). In this setting, the linear convergence rate attained by GD-LS is provably better than the $\Omega(1/\epsilon)$ convergence of GD(1/L) and matches the rate of natural policy gradient (Kakade & Langford, 2002). In Sec. 5.2, we analyze the convergence of GD-LS for functions satisfying the PL condition (Polyak, 1987; Karimi et al., 2016) and instantiate the result for generalized linear models with the logistic link function. Our result demonstrates that GD-LS can converge faster than both constant step-size and normalized GD (Hazan et al., 2015).

**Contribution 5.** Finally, in Sec. 6, we investigate whether the advantages of line-search carry over to the stochastic setting. Specifically, we consider a finite-sum objective $f(\theta) = \frac{1}{n}\sum_{i=1}^{n} f_i(\theta)$, and study the convergence of stochastic gradient descent (SGD) in conjunction with a stochastic line-search proposed in Vaswani et al. (2019b). We restrict our attention to the *interpolation setting* (Vaswani et al., 2019a; Ma et al., 2018; Schmidt & Roux, 2013) which implies that each $f_i$ is minimized at $\theta^* := \arg\min f(\theta)$. The interpolation assumption is of practical interest (Zhang et al., 2017; Belkin et al., 2019), for example, it is satisfied for logistic regression with linearly separable data.

Interpolation enables the fast convergence of SGD, allowing it to match the GD convergence rate, but with an $O(1)$ iteration cost. Under interpolation, SGD with a stochastic line-search (referred to as SGD-SLS) and its variants empirically outperform constant step-size SGD, and have been used to train deep neural networks (Vaswani et al., 2019b; Galli et al., 2024). We further investigate the convergence of SGD-SLS for logistic regression with linearly separable data, and prove that it can leverage non-uniform smoothness to achieve faster rates.

## 2. Problem Formulation

We aim to solve the unconstrained minimization problem: $\min_{\theta \in \mathbb{R}^d} f(\theta)$. We define $\theta^* \in \arg\inf f(\theta)$ as an optimal solution and $f^* := \inf f(\theta)$ as the minimum function value. Throughout, we consider $f$ to be twice-differentiable and satisfying the following assumptions:

**Assumption 1.** $f$ is non-negative i.e. for all $\theta$, $f(\theta) \geq 0$.

**Assumption 2.** $f$ is $(L_0, L_1)$ non-uniform smooth i.e. for constants $L_0, L_1 \geq 0$,

*(a) For all $x, y$ such that $\|x - y\| \leq \frac{q}{L_1}$, where $q \geq 1$ is a constant, if $A := 1 + e^q - \frac{e^q - 1}{q}$ and $B := \frac{e^q - 1}{q}$,*

$$f(y) \leq f(x) + \langle \nabla f(x), y - x \rangle$$
$$+ \frac{(A\,L_0 + B\,L_1\,f(x))}{2}\|y - x\|_2^2 \qquad (1)$$

*(b) For all $\theta$, $\left\|\nabla^2 f(\theta)\right\| \leq L_0 + L_1 f(\theta)$*

The above condition is similar to the non-uniform smoothness conditions proposed in the literature (Zhang et al., 2019; 2020; Chen et al., 2023). Subsequently, we will see that this alternate definition of non-uniform smoothness is more general and enables a simpler analysis for GD-LS.

If $L_1 = 0$, Assn. 2 recovers the standard uniform smoothness condition as a special case. Consequently, common smooth objectives such as linear regression or logistic regression satisfy the above condition. For example, if $X \in \mathbb{R}^{d \times n}$ is the feature matrix, and $y \in \mathbb{R}^n$ is the vector of measurements, then the linear regression objective, $f(\theta) = \frac{1}{2n} \|X\theta - y\|_2^2$ is $(\frac{1}{n}\lambda_{\max}[X^T X], 0)$ non-uniform smooth where $\lambda_{\max}[A]$ is the maximum eigenvalue of the positive semi-definite matrix $A$.

In order to show the benefit of GD with Armijo line-search, we will focus on functions where $L_1 \neq 0$, and the smoothness depends on the function value. We will require these functions to satisfy an additional assumption that relates the gradient norm to the function value. As we will see, such an assumption is true for losses with an exponential tail, even when using a 2 layer neural network (Taheri & Thrampoulidis, 2023; Wu et al., 2024).

**Assumption 3.** *For all $\theta$, there exist constants $\omega, \nu \geq 0$ s.t.*

$$\|\nabla f(\theta)\| \leq \nu\, f(\theta) + \omega. \qquad (2)$$

To motivate the above assumptions, we prove that common convex objectives for supervised learning such as linear logistic regression and linear multi-class classification satisfy Assn. 2 with $L_0 = 0$ and non-zero $L_1$. Moreover, we show that these functions also satisfy Assn. 3 with $\omega = 0$. Below, we state the result for logistic regression, and defer the other results and all proofs to App. A.

**Proposition 1.** *Consider $n$ points where $x_i \in \mathbb{R}^d$ are the features and $y_i \in \{-1, 1\}$ are the corresponding labels. Logistic regression with the objective*

$$f(\theta) = \frac{1}{n} \sum_{i=1}^{n} \ln(1 + \exp(-y_i \langle x_i, \theta \rangle)) \qquad (3)$$

*satisfies Assn. 2 with $L_0 = 0$, $L_1 = 8 \max_{i \in [n]} \|x_i\|_2^2$, and Assn. 3 with $\nu = 8 \max_i \|x_i\|$, $\omega = 0$.*

Note that the logistic regression objective is also uniform smooth, meaning that it simultaneously satisfies Assn. 2 with $L_0 = \frac{1}{4n}\lambda_{\max}[X^T X]$ and $L_1 = 0$, where $X \in \mathbb{R}^{n \times d}$ is the corresponding feature matrix. On the other hand, binary classification with the exponential loss is not uniform smooth on an unbounded domain, but satisfies Assn. 2 with $L_0 = 0$, $L_1 = 8 \max_{i \in [n]} \|x_i\|_2^2$ (see Prop. 6 in App. A).

Next, we show that the non-convex objective corresponding to the generalized linear model (GLM) with a logistic link function also satisfies Assn. 1 to 3. In particular, we prove the following proposition in App. A.

**Proposition 2.** *Consider $n$ points where $x_i \in \mathbb{R}^d$ are the features and $y_i \in [0, 1]$ are the corresponding labels. If $\pi_i(\theta) = \sigma(\langle x_i, \theta \rangle) := \frac{1}{1 + \exp(-\langle x_i, \theta \rangle)}$, the GLM objective,*

$$f(\theta) = \frac{1}{2n} \sum_{i=1}^{n} (\pi_i(\theta) - y_i)^2 , \qquad (4)$$

*satisfies Assn. 2 with $L_0 = \frac{9}{16} \max_{i \in [n]} \|x_i\|_2^2$, $L_1 = 9 \max_{i \in [n]} \|x_i\|_2^2$ and Assn. 3 with $\nu = 9 \max_i \|x_i\|$, $\omega = \max_i \|x_i\|$.*

Finally, in App. A, we show that the objective for softmax policy gradient (Mei et al., 2020) also satisfies the required assumptions for multi-armed bandits and tabular Markov decision processes, and we study these objectives in Sec. 5.

**Connection to non-uniform smoothness in Zhang et al. (2019)**: Assn. 2 and 3 are related to the non-uniform smoothness condition proposed in Zhang et al. (2019) and analyzed in Zhang et al. (2020); Li et al. (2023); Chen et al. (2023); Vankov et al. (2024); Gorbunov et al. (2024). In particular, we prove the following result in App. A.

**Proposition 3.** *For a non-negative, twice-differentiable function $f$, if $f$ is $(L_c, L_g)$ non-uniform smooth according to Zhang et al. (2019) i.e.*

$$\left\|\nabla^2 f(\theta)\right\| \leq L_c + L_g \left\|\nabla f(\theta)\right\| ,$$

*then, it is satisfies Assn. 2 with $L_0 = L_c + L_g \sqrt{2L_c}$, and $L_1 = L_g \left(8L_g + \sqrt{2L_c}\right)$ and Assn. 3 with $\nu = 8L_g + \sqrt{2L_c}$ and $\omega = \sqrt{2L_c}$.*

Hence, the $(L_c, L_g)$ non-uniform smoothness in Zhang et al. (2019) implies Assn. 2 and 3. Consequently, our subsequent results also apply to non-negative, twice-differentiable, $(L_c, L_g)$ non-uniform smooth functions.

In order to further motivate Assn. 2 and 3, consider the logistic regression example in Prop. 1. For logistic regression, the loss corresponding to a *single point* satisfies the notion of $(L_c, L_g)$ non-uniform smoothness with $L_c = 0$ and $L_g = \|x_i\|$ (Gorbunov et al., 2024, Example 1.6). However, the finite-sum over $n$ points ($f(\theta)$ in Eq. (3)) does not necessarily satisfy this assumption with $L_c = 0$ (see Prop. 8 for a simple example with $n = 2$). Consequently, unlike the result in Prop. 1, using the $(L_c, L_g)$ condition in conjunction with Prop. 3 does not directly imply that $L_0 = 0$. In Sec. 4, we will see that, for logistic regression, having $L_0 = 0$ is the key to achieving fast convergence for GD-LS. This further justifies our alternate definition of non-uniform smoothness.

Now that we have motivated Assn. 1 to 3, in the next section, we consider minimizing such non-uniform smooth functions using gradient descent with Armijo line-search.

# 3. GD with Armijo Line-search

The update for gradient descent (GD) with Armijo line-search (Armijo, 1966) (henceforth referred to as `GD-LS`) at iteration $t \in [T]$ is given as: $\theta_{t+1} = \theta_t - \eta_t \nabla f(\theta_t)$, where $\eta_t$ is the step-size returned by the backtracking Armijo line-search (referred to as Armijo-LS). In particular, starting from an initial maximum step-size $\eta_{\max}$, Armijo-LS uses backtracking to select the (approximately) largest step-size that satisfies the Armijo condition,

$$f(\theta_t - \eta_t \nabla f(\theta_t)) \leq f(\theta_t) - c\eta_t \left\| \nabla f(\theta_t) \right\|_2^2, \quad (5)$$

where $c \in (0, 1)$ is a tunable parameter. The complete pseudo-code is described in Alg. 1. The parameter $\beta$ controls the backtracking and is typically set to 0.9, while the parameter $c$ is typically set to a small value such as $10^{-4}$ (Nocedal & Wright, 2006).

---

**Algorithm 1** GD with Armijo Line-search (`GD-LS`)

---

1: **Input**: $\theta_0, \eta_{\max}, c \in (0, 1), \beta \in (0, 1)$
2: **for** $t = 0, \dots, T-1$ **do**
3:    $\tilde{\eta}_t \leftarrow \eta_{\max}$
4:    **while** $f(\theta_t - \tilde{\eta}_t \nabla f(\theta_t)) > f(\theta_t) - c\tilde{\eta}_t \left\| \nabla f(\theta_t) \right\|_2^2$ **do**
5:       $\tilde{\eta}_t \leftarrow \tilde{\eta}_t \beta$
6:    **end while**
7:    $\eta_t \leftarrow \tilde{\eta}_t$
8:    $\theta_{t+1} = \theta_t - \eta_t \nabla f(\theta_t)$
9: **end for**

---

When using `GD-LS` for minimizing $L$ uniformly-smooth functions (corresponding to $L_0 \neq 0$ and $L_1 = 0$ in Assn. 2), $\eta_t$ is constrained to lie in the $\left[ \min \left\{ \eta_{\max}, \frac{2(1-c)\beta}{L} \right\}, \eta_{\max} \right]$ range (Nocedal & Wright, 2006). Note that this bound holds for all $L$ uniformly-smooth functions, does not require convexity, and guarantees that the backtracking line-search will terminate at a non-zero step-size. The parameter $c$ controls the "aggressiveness" of the algorithm; small $c$ values encourage a larger step-size. Hence, Armijo-LS can be seen as method to obtain a step-size proportional to $1/L$, without the knowledge of the global smoothness constant.

These bounds on the step-size can be used to derive the convergence rate for `GD-LS`. For example, for uniformly $L$-smooth and convex functions, the standard analysis shows that `GD-LS` converges to an optimum at an $O(1/T)$ rate (Nocedal & Wright, 2006). It thus matches the rate of GD with a constant step-size equal to $1/L$ (henceforth referred to as `GD(1/L)`). However, as alluded to in Sec. 1, Armijo-LS enables GD to adapt to the "local" smoothness $L(\theta_t)$ (the smoothness around iterate $\theta_t$), and results in faster convergence both in theory (Scheinberg et al., 2014; Lu & Mei, 2023; Mishkin et al., 2024; Fox & Schmidt), and

in practice. In contrast to these works that show constant factor improvements for `GD-LS`, we study non-uniform smooth functions satisfying Assn. 2 and 3, and aim to show that `GD-LS` can result in a faster rate of convergence.

We first show that, when minimizing non-uniform smooth functions satisfying Assn. 2 and 3, `GD-LS` can result in provably larger step-sizes that enable faster convergence. For the subsequent theoretical analysis, we only consider "exact backtracking" i.e. we assume that the backtracking procedure returns the *largest* step-size that satisfies the Armijo condition, meaning that $\beta \approx 1$. Similar to the standard analysis (Nocedal & Wright, 2006), it is straightforward to relax this assumption. We prove the following lemma that lower-bounds the step-size returned by `GD-LS`.

**Lemma 1.** *If $f$ satisfies Assn. 1 to 3, at iteration $t$, `GD-LS` returns a step-size $\eta_t \geq \min \left\{ \eta_{\max}, \frac{1}{\lambda_0 + \lambda_1 f(\theta_t)} \right\}$, where $\lambda_0 := 3\frac{L_0 + L_1 \omega}{(1-c)}$ and $\lambda_1 := 3\frac{L_1(\nu+1)}{(1-c)}$.*

Note that when $L_1 = 0$, the lower-bound on $\eta_t$ is similar to that for uniformly smooth functions. However, when $L_0 = 0$ and $\omega = 0$ (e.g. for logistic regression in Prop. 1), the lower-bound on $\eta_t$ is proportional to $1/f(\theta_t)$, meaning that as $f(\theta_t)$ decreases, the step-size returned by Armijo-LS increases. This enables the faster convergence of `GD-LS`.

We now present a meta-theorem (proved in App. B) that characterizes this fast convergence for both convex and non-convex functions. It requires an additional condition which lower-bounds the gradient norm in terms of the function sub-optimality. In Sec. 4, we use convexity to satisfy this condition, whereas in Sec. 5, we use gradient domination to satisfy it. We also require that the step-size is not constrained by the initial choice, but rather by the properties of the function. This can be achieved by using a large $\eta_{\max}$, which is greater than $\frac{1}{\lambda_0 + \lambda_1 f(\theta_t)}$ for all $t$, or by using a forward-tracking line-search to (approximately) return the largest step-size satisfying the Armijo condition (Fridovich-Keil & Recht, 2019). For conciseness, we express this requirement as $\eta_{\max} = \infty$. We note that it is straightforward to relax it and get an explicit dependence on $\eta_{\max}$, albeit at the cost of clarity in our theoretical results.

**Theorem 1.** *For a fixed $\epsilon > 0$, if $f$ satisfies Assn. 1 to 3, and if for a constant $R > 0$, $\left\| \nabla f(\theta_t) \right\|_2^2 \geq \frac{[f(\theta_t) - f^*]^2}{R}$ for all iterations $t \in [T]$, then, `GD-LS` with $\eta_{\max} = \infty$ requires*

$$T \geq \begin{cases} \max\{2R\lambda_1, 1\} \left( \frac{f^*}{\epsilon} + 1 \right) \ln \left( \frac{f(\theta_0) - f^*}{\epsilon} \right) \\ \quad \text{if } f^* \geq \frac{\lambda_0}{\lambda_1} - \epsilon \quad \textbf{(Case (1))} \\ \\ \frac{2\lambda_0 R}{\epsilon} + \max\{2R\lambda_1, 1\} \left( \frac{f^*}{\epsilon} + 1 \right) \ln \left( \frac{f(\theta_0) - f^*}{\epsilon} \right) \\ \quad \text{otherwise} \quad \textbf{(Case (2))} \end{cases}$$

*iterations to ensure that $f(\theta_T) - f^* \leq \epsilon$.*

*Proof Sketch.* Using the condition $\|\nabla f(\theta)\|_2^2 \geq \frac{[f(\theta) - f^*]^2}{R}$ with the Armijo condition in Eq. (5) and the lower-bound on $\eta_t$ from Lemma 1, we get that

$$f(\theta_{t+1}) \leq f(\theta_t) - \frac{[f(\theta_t) - f^*]^2}{[\lambda_0 + \lambda_1 f(\theta_t)] R}. \quad (6)$$

We now split the proof into two cases: **Case (1)** when $f(\theta_t) \geq \frac{\lambda_0}{\lambda_1}$ for all $t \in [T]$. In this case, $\lambda_1 f(\theta_t) \geq \lambda_0$. Using this relation with Eq. (6) and following the arguments in Axiotis & Sviridenko (2023, Theorem 5.2) allows us to complete the proof of Case (1). **For Case (2)**, we follow the proof of the first case for iterations $t \in [\tau]$ for which $f(\theta_t) \geq \frac{\lambda_0}{\lambda_1}$, and obtain a similar rate. This corresponds to Phase 1. with faster convergence. For all iterations $t \in [\tau, T]$, $f(\theta_t) \leq \frac{\lambda_0}{\lambda_1}$ and hence $\lambda_0 \geq \lambda_1 f(\theta_t)$. Using this relation with Eq. (6) and following the standard proof for uniformly smooth functions completes the proof for Phase 2. which results in slower convergence. Putting the results for both phases together completes the proof of Case (2). $\square$

In order to interpret the above theorem, let us first consider the setting corresponding to $\lambda_1 = 0$. Here, case (2) is active, and the algorithm requires $O(1/\epsilon)$ iterations to achieve the desired sub-optimality, matching the standard result for uniformly smooth functions. Now consider the setting when $\lambda_0 = 0$. Here, case (1) is active, and GD-LS requires $O\left(R\left(\frac{f^*}{\epsilon}\right)\ln\left(\frac{1}{\epsilon}\right)\right)$ iterations. The iteration complexity thus depends on $f^*$, and in cases where $f^*$ is small, GD-LS can result in an improved rate. As a concrete example, consider the case when $\epsilon = \Theta(f^*)$. In this setting, GD-LS will result in a faster $O\left(R\ln\left(\frac{1}{\epsilon}\right)\right)$ convergence. Note that GD(1/L) does not benefit from such adaptivity, and will always result in a sublinear rate. For non-zero $\lambda_0$ and $\lambda_1$, the resulting rate depends on the value of $f^*$. If $f^*$ is larger than the threshold $\frac{2\lambda_0}{\lambda_1}$, GD-LS can result in the potentially fast rate corresponding to case (1), whereas if $f^*$ is smaller than the threshold, the algorithm has a two-phase behaviour: fast convergence until the loss becomes smaller than the threshold, followed by slow convergence to the minimizer.

In the next section, we instantiate the above theorem to prove the fast convergence of GD-LS for convex losses.

## 4. GD-LS for Convex Losses

In this section, we characterize the convergence rate of GD-LS on convex losses satisfying Assn. 1 to 3. Recall that binary classification using logistic regression and multi-class classification using the cross-entropy loss both satisfy Assn. 1 to 3 with $L_0 = 0$ and $\omega = 0$. Consequently, we only instantiate Thm. 1 for this setting and prove the following corollary in App. C.

**Corollary 1.** *For a fixed $\epsilon > 0$, assuming $f(\theta)$ is convex and satisfies Assn. 1 to 3 with $L_0 = 0$ and*

$\omega = 0$, *GD-LS with $\eta_{\max} = \infty$, requires $T \geq \max\{2\lambda_1\|\theta_0 - \theta^*\|_2^2, 1\}\left(\frac{f^*}{\epsilon} + 1\right)\ln\left(\frac{f(\theta_0) - f^*}{\epsilon}\right)$ iterations to ensure that $f(\theta_T) - f^* \leq \epsilon$.*

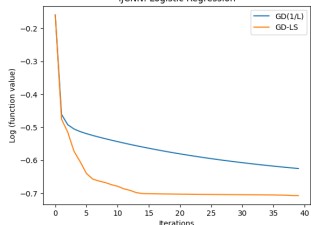

*Figure 1.* Comparing GD-LS with $c = 1/2$ and $\eta_{\max} = 10^8$ and GD(1/L) for unregularized logistic regression on the ijcnn dataset (Chang & Lin, 2011). $f^*$ is small and GD-LS converges faster.

Referring to the explanation following Thm. 1, we conclude tht GD-LS can result in faster convergence than GD(1/L) when $f^*$ is small (see Fig. 1 for an experimental validation).

In order to compare the result in Cor. 1 with existing works, consider the special case of logistic regression. In this case, GD-LS matches the rate for a variant of normalized gradient descent (NGD) in Axiotis & Sviridenko (2023, Theorem 5.2). However, unlike GD-LS, NGD requires the knowledge of $L_1$ making it relatively difficult to be implemented in practice. Furthermore, we note NGD is a specialized algorithm that is helpful to attain faster rates for certain problems (Mei et al., 2021; Hazan et al., 2015; Wilson et al., 2019), whereas GD-LS is universally used and can automatically exploit the problem structure. Moreover, Cor. 1 is more general and also holds for the exponential loss.

While GD(1/L) results in an $\Omega(1/\epsilon)$ rate for general smooth, convex functions (Nesterov et al., 2018), analyses of GD on logistic regression often exploit strong-convexity and prove faster rates (Karimi et al., 2016). In particular, if the iterates lie in a bounded set, the objective is $\mu(\theta)$ strongly-convex where $\mu(\theta) = \lambda_{\min}[X^T X] \min_i \pi_i(\theta)(1 - \pi_i(\theta))$ where $\pi_i = \frac{1}{1 + \exp(-y_i \langle x_i, \theta \rangle)}$. Notice that as $\pi_i(\theta)$ tends to either zero or one i.e. the predictions become deterministic, $\mu(\theta)$ tends to 0. Freund et al. (2018, Theorem 3.3) characterize the resulting rate for GD(1/L) and prove that the suboptimality scales as $O(\exp(-T \exp(-1/\xi)))$ where $\xi$ is the degree of non-separability and tends to zero as the data becomes more separable. Hence, the rate becomes exponentially worse as $\xi$ decreases. However, as the data becomes separable, GD-LS converges at a faster linear rate (see Fig. 2), meaning that strong-convexity cannot explain this behaviour.

Consequently, we consider the special case where the data is linearly separable and $f^* = 0$, and better characterize the fast convergence of GD-LS. Unfortunately, we cannot directly use Cor. 1 since $\|\theta\| \to \infty$ as $f(\theta) \to 0$, making the resulting bound vacuous (Orabona, 2024). Consequently, we use a different technique, and first prove the following theorem in App. C.

**Theorem 2.** *For an initialization $\theta_0$, $\epsilon \in (0, f(\theta_0))$ and comparator $u$ s.t. $f(u) \leq \epsilon$, if $f(\theta)$ is convex, satisfies Assn. 1 to 3 with $L_0 = 0$, $\omega = 0$, then,* `GD-LS` *with $\eta_{\max} = \infty$, $c > \frac{1}{2}$, requires $T \geq \frac{c \lambda_1 \|\theta_0 - u\|_2^2}{(2c-1)} \left[ 1 + \frac{f(u)}{\epsilon} \right]$ iterations to ensure that $f(\theta_T) - f(u) \leq \epsilon$.*

Compared to Cor. 1, the above result only holds for a restricted range of $\epsilon$ and a comparator $u$ s.t. $f(u) \leq \epsilon$. Note that since $f$ is non-negative and `GD-LS` ensures monotonic descent, its sub-optimality is upper-bounded by $f(\theta_0)$. The result only requires the non-uniform smoothness assumptions and thus holds for the exponential loss, logistic regression and multi-class classification.

We now use the above result and prove the following corollary for logistic regression on separable data (similar results hold for the exponential loss and multi-class classification).

**Corollary 2.** *For logistic regression on linearly separable data with margin $\gamma$, if, for all $i$, $\|x_i\| \leq 1$, for an initialization $\theta_0$, an $\epsilon \in (0, f(\theta_0))$,* `GD-LS` *with $\eta_{\max} = \infty$ requires $T \geq O \left( \frac{c}{(1-c)(2c-1)\gamma^2} \left[ \ln \left( \frac{1}{\epsilon} \right) \right]^2 \right)$ iterations to ensure that $f(\theta_T) \leq 2\epsilon$.*

Hence, on linearly separable data, `GD-LS` can achieve a linear rate of convergence for logistic regression (see Fig. 2 for an experimental validation). In this setting, Wu et al. (2024, Theorem 3) show that `GD(1/L)` (or more generally, GD with any constant step-size that guarantees monotonic descent) cannot have a convergence rate faster than $\Omega(1/\epsilon)$. Hence, `GD-LS` can result in an exponential improvement over the rate for `GD(1/L)`. Furthermore, the rate in Cor. 2 is better than the $O(1/\sqrt{\epsilon})$ rate for GD with large (beyond $1/L$) constant step-sizes (Wu et al., 2024). Finally, we note that, in this setting, no algorithm that guarantees monotonic descent in the function values can achieve a rate faster than the linear rate (Zhang et al., 2025, Theorem 2.2), Hence, in this sense, `GD-LS` is optimal for logistic regression.

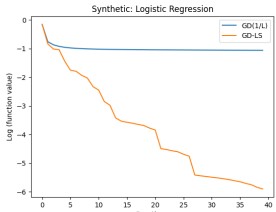 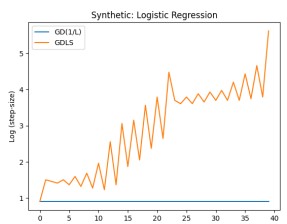

*Figure 2.* Comparing `GD-LS` with $c = 1/2$, $\eta_{\max} = 10^8$ and `GD(1/L)` for unregularized logistic regression on a synthetic separable dataset with $\gamma = 0.1$, $n = 10^4$ and $d = 200$. (Left) Sub-optimality plot: `GD-LS` converges linearly, while `GD(1/L)` has a sublinear convergence. (Right) Step-size plot: The `GD-LS` step-size increases non-monotonically.

Interestingly, in App. C.1, we prove that GD with the Polyak

step-size (Polyak, 1987) (which does not necessarily result in monotonic descent in the function values) can also achieve the linear convergence rate in Thm. 2.

**Convergence under $(L_c, L_g)$ non-uniform smoothness**: Using the reduction in Prop. 3 allows characterizing the convergence of `GD-LS` on twice-differentiable, non-negative, $(L_c, L_g)$ non-uniform smooth (Zhang et al., 2019) and convex functions. In particular, by using Prop. 3 in conjunction with Thm. 1 allows us bound the sub-optimality for the last-iterate of `GD-LS` (see Cor. 5 in App. C for the formal statement) without requiring the knowledge of $L_c$ or $L_g$. In particular, assuming $L_g \geq 1$, $L_c \geq 1$, we prove that `GD-LS` requires $\tilde{O} \left( \frac{L_c L_g^2 R}{\epsilon} + \frac{L_g (L_g^2 + L_c) R f^*}{\epsilon} \right)$ iterations. In contrast, in this same setting, Vankov et al. (2024); Gorbunov et al. (2024) show that GD with the Polyak step-size requires $O \left( \max \left\{ \frac{L_c}{\epsilon}, L_g R \right\} \right)$ iterations for the best-iterate to achieve an $\epsilon$ sub-optimality (Gorbunov et al., 2024, Thm. 4.1). Comparing the two results, we note that, in general, the upper-bound in (Gorbunov et al., 2024) is tighter in terms of $\epsilon$. However, when $\epsilon = \Theta(f^*)$, the two bounds are equivalent in their $\epsilon$ dependence. Moreover, while the results in Gorbunov et al. (2024); Vankov et al. (2024) hold for the best-iterate of GD with the Polyak step-size (that requires the knowledge of $f^*$), Cor. 5 holds for the last-iterate and does not require knowing $f^*$.

Next, we analyze `GD-LS` on non-convex losses.

## 5. `GD-LS` for Non-convex Losses

In this section, we consider non-convex losses that satisfy two alternative gradient domination conditions which enable convergence to the global optimum. In Sec. 5.1, we analyze the convergence of `GD-LS` for objectives corresponding to the softmax policy gradient in reinforcement learning. In Sec. 5.2, we consider objectives satisfying the Polyak-Łojasiewicz (PL) condition (Karimi et al., 2016; Polyak, 1987), and study the generalized linear model with a logistic link function as an example.

**Assumption 4.** *$f$ satisfies a non-uniform gradient domination condition if there exists a constant $\zeta \geq 1$ and $\mu(\theta) > 0$ s.t. for all $\theta$, $\|\nabla f(\theta)\|^\zeta \geq \mu(\theta) [f(\theta) - f^*]$.*

Gradient domination or Łojasiewicz conditions are satisfied for matrix factorization (Ward & Kolda, 2023), policy gradient in reinforcement learning (Mei et al., 2020) and generalized linear models (Mei et al., 2021). These conditions have been exploited to prove global convergence guarantees for first-order methods (Karimi et al., 2016; Mei et al., 2021).

### 5.1. Gradient domination with $\zeta = 1$

We use softmax policy optimization for multi-armed bandits (MAB) as a concrete example that satisfies Assn. 1 to 3

and Assn. 4 with $\zeta = 1$. In particular, we consider an MAB problem in the *exact setting* with deterministic, known rewards. This setting is often used as a testbed to evaluate policy gradient methods (Xiao, 2022; Mei et al., 2020; Lu et al., 2024). We prove the following proposition in App. A.

**Proposition 4.** *Given an MAB problem with $K$ arms and known deterministic rewards $r \in [0,1]^K$, consider the class of softmax policies $\pi_\theta \in \Delta_K$ parameterized by $\theta \in \mathbb{R}^K$ s.t. $\pi_\theta(a) = \frac{\exp(\theta(a))}{\sum_{a'} \exp(\theta(a'))}$. The loss corresponding to the bandit problem is given by: $f(\theta) = r(a^*) - \langle \pi_\theta, r \rangle$, where $a^* := \arg\max_{a \in [K]} r(a)$ is the optimal arm. $f(\theta)$ is non-negative, satisfies Assn. 2 with $L_0 = 0$ and $L_1 = 72$, Assn. 3 with $\nu = 24$ and $\omega = 0$ and Assn. 4 with $\zeta = 1$ and $\mu(\theta) = \pi_\theta(a^*)$.*

Softmax policy gradient methods (Williams, 1992) optimize the above non-convex objective using gradient descent, and have been analyzed recently (Mei et al., 2020; Agarwal et al., 2021). We aim to use GD-LS to optimize the objective defined in Prop. 4 for which the GD update is given by: $\theta_{t+1} = \theta_t - \eta_t \nabla f(\theta_t) = \theta_t + \eta_t \nabla_\theta \langle \pi_\theta, r \rangle$, and the corresponding Armijo condition is given as $\langle \pi_{\theta_{t+1}}, r \rangle \geq \langle \pi_\theta, r \rangle + c\eta_t \|\nabla_\theta \langle \pi_\theta, r \rangle\|_2^2$.

In Prop. 7 in App. A, we show that, under additional assumptions, the softmax policy gradient objective for tabular Markov decision processes (MDPs) also satisfies the Assn. 1 to 4 with $L_0 = 0$ $\omega = 0$ and $\zeta = 1$. In this case, the *exact setting* corresponds to knowing the rewards and the transition probabilities, and is the same setting under which classic RL algorithms such as value iteration or policy iteration are analyzed (Puterman, 2014).

We now characterize the convergence rate of GD-LS.

**Corollary 3.** *For an $\epsilon > 0$, assuming $f(\theta)$ satisfies Assn. 1 to 3 with $L_0 = 0$, $\omega = 0$ and Assn. 4 with $\zeta = 1$, if $\mu := \min_{t \in [T]} \mu(\theta_t)$, then, GD-LS with $\eta_{\max} = \infty$, requires $T \geq \max\left\{1, \frac{2\lambda_1}{\mu^2}\right\}\left(\frac{f^*}{\epsilon} + 1\right)\ln\left(\frac{f(\theta_0) - f^*}{\epsilon}\right)$ iterations to ensure $f(\theta_T) \leq \epsilon$.*

In order to better understand the implications of Cor. 3, we instantiate the above result for MAB and prove the following corollary in App. D.1.

**Corollary 4.** *For an MAB problem with $K$ arms, rewards bounded in $[0,1]$, GD-LS with a uniform initialization i.e. $\forall a$, $\pi_{\theta_0}(a) = 1/K$, $c = \frac{1}{2}$, $\eta_{\max} = \infty$ requires $T = O\left(K^2 \ln(1/\epsilon)\right)$ iterations to guarantee $\langle \pi_{\theta_T}, r \rangle \geq r(a^*) - \epsilon$.*

Hence, for MAB, GD-LS converges at a linear rate. Under additional assumptions, by combining Prop. 7 and Cor. 3, we can prove a similar linear rate for tabular MDPs in the exact setting (see Cor. 6 in App. D.1 for the formal result).

The above result is in contrast to GD(1/L) which can

only attain an $\Omega\left(\frac{1}{\epsilon}\right)$ convergence rate for both bandits and MDPs (Mei et al., 2020, Theorem 9, 10). The convergence rate of GD-LS matches that of algorithms designed for this specific problem, including GD with a specific line-search that requires the knowledge of $f^*$ (Lu et al., 2024), GD with specific increasing step-sizes (Liu et al., 2024), normalized GD (Mei et al., 2021), natural policy gradient (Kakade & Langford, 2002; Xiao, 2022) and mirror descent with a log-sum-exp mirror map (Asad et al., 2024). From a practical perspective, Lu et al. (2024, Figure 1) empirically demonstrate the linear convergence of GD-LS on tabular Markov decision processes, and hence, Cor. 6 substantiates their results theoretically.

**Two-layer neural networks**: We note that Assn. 1 to 3 and Assn. 4 with $\zeta = 1$ are also satisfied when using the (i) exponential loss to train (ii) two-layer neural networks with a smoothed leaky-ReLU non-linearity and (iii) assuming that the training data is linearly separable (Taheri & Thrampoulidis, 2023, Lemmas 3,5). In this setting, Theorem 1 in Taheri & Thrampoulidis (2023) shows that normalized GD can result in linear convergence for the resulting non-convex objective. Since this problem also satisfies the required assumptions for Cor. 3, GD-LS also converges linearly and unlike normalized GD, it does not require knowing problem-dependent constants.

Hence, these results demonstrate the universality of GD-LS.

### 5.2. Gradient domination with $\zeta = 2$

We use the generalized linear model (GLM) with a logistic link function as an example of an objective that satisfies the PL condition (which corresponds to Assn. 4 with $\zeta = 2$).

**Lemma 2** (Lemma 9 in (Mei et al., 2021)). *If $\sigma(\cdot)$ is the sigmoid function and $\pi_i(\theta) := \sigma(\langle x_i, \theta \rangle)$, assuming that for all $i \in [n]$, $\|x\|_i \leq 1$, $y_i = \pi_i(\theta^*)$ such that $\|\theta^*\| \leq D < \infty$ and $\upsilon(\theta) := \min_{i \in [n]} \{\pi_i(\theta) \cdot (1 - \pi_i(\theta))\}$, then the GLM objective in Eq. (4) satisfies Assn. 4 with $\zeta = 2$ and $\mu(\theta) = 64 [\upsilon(\theta)]^2 [\min\{\upsilon(\theta), \upsilon(\theta^*)\}]^2$.*

Similar to logistic regression, the PL constant $\mu$ depends on $\upsilon(\theta)$. However, unlike logistic regression where $y_i \in \{0, 1\}$ and $\|\theta^*\|$ can be infinite for separable data, for GLMs, $y_i \in (0, 1)$, $\|\theta^*\|$ is bounded and consequently, $\mu(\theta^*) > 0$. Hence, as long as $\|\theta_t\| < \infty$ for all iterates $t \in [T]$, $\mu(\theta)$ is bounded away from zero. However, we note that this does not preclude the case where the iterates initially diverge away from the solution, resulting in large $\|\theta_t\|$ and small (but non-zero) $\mu(\theta)$.

Recall that in Prop. 2, we have seen that the GLM objective satisfies Assn. 1 to 3 with non-zero $L_0, L_1, \nu, \omega$. Furthermore, since the targets $y_i = \pi_i(\theta^*)$ are assumed to be realizable in Lemma 2, $f^* = 0$. Given these considerations, we prove the following theorem in App. D and characterize

the convergence of `GD-LS` for such an objective.

**Theorem 3.** *For a fixed $\epsilon \in \left(0, \frac{\lambda_0}{\lambda_1}\right)$, if $f$ satisfies Assn. 1 to 3 and Assn. 4 with $\zeta = 2$, $f^* = 0$ and if $\mu := \min_{t \in [T]} \mu(\theta_t) > 0$, then, `GD-LS` with $\eta_{\max} = \infty$, requires $T \geq \frac{2}{\mu} \left[\lambda_1 f(\theta_0) + \lambda_0 \ln\left(\frac{\lambda_0}{\lambda_1 \epsilon}\right)\right]$ iterations to ensure that $f(\theta_T) \leq \epsilon$.*

The above result shows that `GD-LS` converges linearly, where the convergence rate depends on the ratio $\lambda_0/\lambda_1$. On the other hand, for an $L$ uniformly-smooth function satisfying the PL condition, `GD(1/L)` requires $O\left(\frac{L}{\mu} \ln\left(\frac{f(\theta_0)}{\epsilon}\right)\right)$ iterations (Karimi et al., 2016). Ignoring the constant first term which is independent of $\epsilon$ and assuming $\lambda_0 \approx L$, we can see that the result in Thm. 3 is better than the standard rate when $\lambda_0/\lambda_1$ is smaller than $f(\theta_0)$. It is important to note that since `GD-LS` can automatically (without any change in the algorithm) exploit the uniform smoothness and obtain the standard result as well, the number of iterations required for `GD-LS` is $\min\left\{O\left(\frac{L}{\mu} \ln\left(\frac{f(\theta_0)}{\epsilon}\right)\right), O\left(\frac{\lambda_0}{\mu} \ln\left(\frac{\lambda_0}{\lambda_1 \epsilon}\right)\right)\right\}$, meaning that `GD-LS` converges at least as fast as `GD(1/L)`. Since the GLM objective is also uniformly smooth (Mei et al., 2021, Lemma 10), in Fig. 3, we empirically compare to `GD(1/L)`, and verify the faster convergence of `GD-LS`.

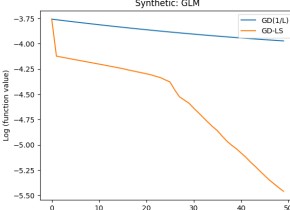

*Figure 3.* Comparing `GD-LS` with $c = 1/2$, $\eta_{\max} = 10^4$ and `GD(1/L)` for GLM on a synthetic dataset with $n = 10^4$, $d = 200$, $\|\theta^*\| = 1$.

Comparing Thm. 3 to the existing results for the GLM objective, we note that Hazan et al. (2015, Lemma 3.1) show that the objective is locally quasi-convex, and use this property to derive a slower $O(1/\epsilon^2)$ convergence rate for normalized GD with a decreasing step-size (Hazan et al., 2015, Theorem 4.1). On the other hand, Mei et al. (2021) propose a novel variant of normalized GD and prove that it converges linearly, with a better constant dependence compared to GD. However, their method requires the knowledge of $\mu$, making it difficult to implement. On the other hand, `GD-LS` does not require the knowledge of any problem-dependent constants, and achieves similar or better theoretical results compared to these specialized methods.

In the next section, we show the benefits of using a line-search in the stochastic setting.

## 6. SGD with Stochastic Line-search

In this section, we analyze the convergence of stochastic gradient descent (SGD) (Robbins & Monro, 1951) with a

stochastic variant of the Armijo line-search (referred to as `SGD-SLS`) proposed in Vaswani et al. (2019b).

We focus on the convex, finite-sum setting, and consider minimizing $f(\theta) = \frac{1}{n} \sum_{i=1}^{n} f_i(\theta)$ where each $f_i$ is convex and satisfies Assn. 1 to 3. Logistic regression and multiclass classification using the cross-entropy loss are examples of such an objective. For ease of exposition, we assume that each $f_i$ is $(L_0, L_1)$ non-uniform smooth, and note that it is straightforward to analyze the case where the $f_i$'s have different smoothness constants.

At iteration $t \in [T]$, SGD randomly samples a function $f_t$ from the $n$ functions in the finite-sum, computes its gradient and updates the parameters. Specifically,

$$\theta_{t+1} = \theta_t - \eta_t \nabla f_t(\theta_t), \tag{7}$$

where $\nabla f_t(\theta_t)$ is the gradient of the loss function chosen at iteration $t$. Each stochastic gradient $\nabla f_t(\theta_t)$ is unbiased, implying that $\mathbb{E}_t[\nabla f_t(\theta)] = \nabla f(\theta)$. In order to estimate $\eta_t$, `SGD-SLS` uses the stochastic analog of the Armijo condition in Eq. (5). In particular, starting from $\eta_{\max}$, SLS uses a backtracking procedure and returns the largest step-size $\eta_t$ that satisfies: $\eta_t \leq \eta_{\max}$ and,

$$f_t(\theta_t - \eta_t \nabla f_t(\theta_t)) \leq f_t(\theta_t) - c\,\eta_t\,\|\nabla f_t(\theta_t)\|_2^2. \tag{8}$$

Note that the above stochastic Armijo condition only involves the sampled function and its gradient.

In order to analyze the convergence of `SGD-SLS`, we define $f_i^* := \min f_i(\theta)$ as the minimum of function $i$ in the finite-sum and $\chi^2(\theta^*) := \mathbb{E}_i[f_i(\theta^*) - f_i^*]$ as the measure of the stochasticity at the optimum (Loizou et al., 2021). In particular, if $\chi^2 = 0$, then each $f_i$ is minimized at $\theta^*$ implying that $\nabla f_i(\theta^*) = 0$. This special case is referred to as the *interpolation* setting (Vaswani et al., 2019a; Ma et al., 2018; Schmidt & Roux, 2013) and is useful in practical machine learning; for example, it is approximately satisfied by over-parameterized neural networks (Zhang et al., 2017) or non-parametric regression (Liang & Rakhlin, 2018; Belkin et al., 2019). Furthermore, logistic regression on linearly separable data is an example of a smooth convex loss that satisfies the interpolation condition and is the main motivation for the subsequent analysis.

When minimizing uniformly-smooth convex functions in the interpolation setting, Vaswani et al. (2019a) proved that `SGD-SLS` converges to the optimum at an $O(1/\epsilon)$ rate, matching GD and is faster than the standard $O(1/\epsilon^2)$ rate for SGD (Bottou et al., 2018). Motivated by this and the results in Sec. 4, we analyze `SGD-SLS` for logistic regression which satisfies Assn. 1 to 3. We first note the results in Vaswani et al. (2019a) (and subsequent papers analyzing the interpolation setting) do not directly apply to logistic regression. In particular, these results have a dependence

on $\|\theta^*\|$ which is infinite in the logistic regression example (Orabona, 2024). Consequently, we first prove that SGD-SLS can exploit the uniform smoothness in logistic regression and prove a $O(1/\epsilon)$ rate in App. E.

**Theorem 4.** *For logistic regression on linearly separable data with margin $\gamma$, if, for all $i$, $\|x_i\| \leq 1$, for a fixed $\epsilon \in (0, 1)$, SGD-SLS with $\eta_{\max} = \frac{1}{\epsilon}$ and $c = \frac{2}{3}$ requires $T = O\left(\frac{1}{\epsilon\gamma^2} \left[\ln\left(\frac{1}{\epsilon^2}\right)\right]^2\right)$ iterations to ensure that $\mathbb{E}[\inf_{t \in [T]} f(\theta_t)] \leq \frac{5\epsilon}{2}$.*

Next, we exploit the non-uniform smoothness of logistic regression and analyze the convergence of SGD-SLS. Note that since SLS involves an Armijo line-search for one (randomly chosen) function in each iteration, we can follow the same argument as in Lemma 1 and show that the step-size in each iteration is lower-bounded i.e. $\eta_t \geq \min\left\{\eta_{\max}, \frac{1}{\lambda_0 + \lambda_1 f_t(\theta_t)}\right\}$. Given this result, we prove the following theorem in App. E.

**Theorem 5.** *For logistic regression on linearly separable data with margin $\gamma$, if, for all $i$, $\|x_i\| \leq 1$, for a fixed $\epsilon \in \left(0, \frac{C'}{8}\right)$ where $C' = O(1)$, SGD-SLS with $\eta_{\max} = \frac{1}{\epsilon}$ and $c = \frac{2}{3}$ requires $T = O\left(\frac{n}{\gamma^2} \left[\ln\left(\frac{n}{\epsilon^2}\right)\right]^2\right)$ iterations to ensure that $\mathbb{E}[\inf_{t \in [T]} f(\theta_t)] \leq \frac{5\epsilon}{2}$.*

Note that the above $O(n(\ln(n/\epsilon))^2)$ convergence rate is slower than that of GD-LS. However, since SGD-SLS has an $O(1)$ cost per iteration (as compared to the $O(n)$ cost for GD-LS), both algorithms have the same gradient complexity. In order to intuitively understand why it is difficult to prove a faster (independent of $n$) rate for SGD-SLS, first note that the stochastic setting does not allow using arbitrarily large values of $\eta_{\max}$. Second, note that individual losses $f_i$ in the finite sum can become much smaller than $f$. Specifically, consider iteration $t$ of SGD-SLS and consider a point $i$ such that $f_i(\theta_t) \leq \delta << \epsilon$ where $\epsilon$ is the desired sub-optimality. If this point is sampled at iteration $t$, the corresponding size of the update is $\|\theta_{t+1} - \theta_t\| \approx \delta \eta_{\max} = \delta/\epsilon$ which can be small. On the other hand, for $i$ such that $f_i(\theta_t) \geq \delta$, the size of the update is $O(1)$. Hence, if at iteration $t$, there are $n - 1$ "small" losses and the algorithm samples a point uniformly at random, it has an $O(1)$ update with probability $1/n$. Hence, in expectation, the rate depends on $n$ in the worst-case.

However, note that the same SGD-SLS algorithm can achieve the rates in Thm. 4 and 5 and hence, the algorithm has an $O\left(\min\{n, 1/\epsilon\} (\ln(n/\epsilon))^2\right)$ convergence rate. We conjecture that by formalizing the above intuition, we can prove a matching lower-bound and leave this to future work. Note that the resulting gradient complexity for SGD-SLS is smaller than that of GD-LS. Interestingly, this is also smaller than the $O\left((n + 1/\epsilon) \ln(1/\epsilon)\right)$ gradient complexity

for variance reduced methods such as SARAH (Nguyen et al., 2017) on general uniformly-smooth convex losses.

Given the above intuition, a natural question is whether SGD-SLS can attain faster rates if the algorithm can ensure that it samples points that have a relatively large function values. We formalize this in the following theorem proved in App. E.

**Theorem 6.** *For logistic regression on linearly separable data with margin $\gamma$, if, for all $i$, $\|x_i\| \leq 1$, for a fixed $\epsilon \in \left(0, \frac{\min\left\{\frac{1}{2}, C'\right\}}{8}\right)$ where $C' := \frac{1}{\lambda_1} = \frac{1}{648}$, SGD-SLS with $\eta_{\max} = \frac{1}{\epsilon}$ and $c = \frac{2}{3}$ and guaranteeing that for all $t \in [T]$, $\Pr\left[f_t(\theta_t) \geq \frac{\epsilon}{2}\right] = 1$ requires $T = O\left(\frac{1}{\gamma^2} \left[\ln\left(\frac{1}{\epsilon^2}\right)\right]^2\right)$ iterations to ensure that $\mathbb{E}[f(\theta_T)] \leq \frac{3\epsilon}{2}$.*

Hence, if the algorithm can ensure that the sampled function $f_t$ at iteration $t$ has a loss greater than $\epsilon/2$, then, SGD-SLS can indeed achieve a faster $O((\ln(1/\epsilon))^2)$ convergence rate which is independent of $n$. Skipping updates when the sampled point has a small loss, or using a projection step to ensure that no loss becomes smaller than $\epsilon/2$ are two potential ways to ensure that $\Pr\left[f_t(\theta_t) \geq \frac{\epsilon}{2}\right] = 1$. Implementing such approaches in a computationally efficient manner and analyzing the resulting algorithm is challenging, and we leave this interesting direction to future work. Finally, we note that by combining the proof techniques in App. C.1 and App. E, the above convergence guarantees also hold for the stochastic Polyak step-size (Loizou et al., 2021).

# 7. Conclusion

We analyzed GD-LS for a class of functions satisfying non-uniform smoothness. For a range of practical convex and non-convex functions, we proved that Armijo-LS can enable GD to adapt to the objective's properties and result in faster convergence. In particular, we showed that, for specific problems in machine learning, GD-LS can (i) either match or provably improve upon the sublinear rate of GD (1/L), (ii) do so without relying on the knowledge of problem-dependent constants and (iii) match the fast convergence of algorithms tailored for these problems. Our results thus show the universal effectiveness of GD-LS.

We believe that analyzing GD-LS for a broader class of non-convex functions, and characterizing the advantage of using Armijo-LS for other algorithms (such as Nesterov accelerated gradient) are important future directions. We also plan to investigate whether other adaptive step-size schemes such as AdaGrad (Duchi et al., 2011) or Adam (Kingma & Ba, 2014) can also provably adapt to the non-uniform smoothness and result in faster convergence rates.

## Impact Statement

This paper presents work whose goal is to advance the field of Machine Learning. There are many potential societal consequences of our work, none which we feel must be specifically highlighted here.

## Acknowledgments

We thank Damien Scieur, Mark Schmidt, Curtis Fox, Michael Lu and Siyi Meng for helpful discussions and feedback, and Anh Dang for help with the initial experiments. This research was partially supported by the Natural Sciences and Engineering Research Council of Canada (NSERC) Discovery Grant RGPIN-2022-04816.

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

# Supplementary Material

## Organization of the Appendix

## A. Proofs for Sec. 2

**Assumption 5.** *If $f$ is twice differentiable and $(L_c, L_g)$-non-uniform smooth as defined in Zhang et al. (2019), then, for constants $L_c$, $L_g > 0$,*

$$\left\|\nabla^2 f(\theta)\right\| \leq L_c + L_g \left\|\nabla f(\theta)\right\|. \tag{9}$$

**Proposition 3.** *For a non-negative, twice-differentiable function $f$, if $f$ is $(L_c, L_g)$ non-uniform smooth according to Zhang et al. (2019) i.e.*

$$\left\|\nabla^2 f(\theta)\right\| \leq L_c + L_g \left\|\nabla f(\theta)\right\|,$$

*then, it is satisfies Assn. 2 with $L_0 = L_c + L_g \sqrt{2L_c}$, and $L_1 = L_g \left(8L_g + \sqrt{2L_c}\right)$ and Assn. 3 with $\nu = 8L_g + \sqrt{2L_c}$ and $\omega = \sqrt{2L_c}$.*

*Proof.* Using Li et al. (2023, Lemma 3.5), we know that $(L_c, L_g)$ non-uniform smoothness also implies that,

$$\begin{aligned}
\left\|\nabla f(\theta)\right\|_2^2 &\leq 2\left[L_c + 2L_g \left\|\nabla f(\theta)\right\|\right] \left(f(\theta) - f^*\right) \\
&\leq 2\left[L_c + 2L_g \left\|\nabla f(\theta)\right\|\right] f(\theta) &&\text{(Since } f \text{ is non-negative)} \\
\implies \left\|\nabla f(\theta)\right\| &\leq 4L_g f(\theta) + \sqrt{2L_c} \sqrt{f(\theta)} &&\text{(By completing the square w.r.t } \left\|\nabla f(\theta)\right\|)
\end{aligned}$$

Consider two cases depending on whether $f(\theta)$ is larger than 1.

**Case 1**: If $f(\theta) \leq 1 \implies \sqrt{f(\theta)} \leq 1$. The above inequality can be simplified as: $\left\|\nabla f(\theta)\right\| \leq 4L_g f(\theta) + \sqrt{2L_c}$.

**Case 2**: If $f(\theta) > 1 \implies \sqrt{f(\theta)} \leq f(\theta)$. The above inequality can be simplified as: $\left\|\nabla f(\theta)\right\| \leq 4L_g f(\theta) + \sqrt{2L_c} f(\theta)$.

Combining both cases, we have that,

$$\left\|\nabla f(\theta)\right\| \leq \left(8L_g + \sqrt{2L_c}\right) f(\theta) + \sqrt{2L_c} \tag{10}$$

Hence, Assn. 3 is satisfied with $\nu = 8L_g + \sqrt{2L_c}$ and $\omega = \sqrt{2L_c}$. Using this result in the definition of non-uniform smoothness in Assn. 5,

$$\left\|\nabla^2 f(\theta)\right\| \leq L_c + L_g \left[\left(8L_g + \sqrt{2L_c}\right) f(\theta) + \sqrt{2L_c}\right] = \left(L_c + L_g \sqrt{2L_c}\right) + L_g \left(8L_g + \sqrt{2L_c}\right) f(\theta) \tag{11}$$

Hence, Assn. 2 (b) is satisfied with $L_0 = L_c + L_g \sqrt{2L_c}$ and $L_1 = L_g \left(8L_g + \sqrt{2L_c}\right)$.

Using Zhang et al. (2020, Lemma A.3), if $f$ satisfies Assn. 5, then, for all $x, y$ such that $\|x - y\| \leq \frac{q}{L_g}$ where $q > 0$ is a constant,

$$f(y) \leq f(x) + \langle \nabla f(x), y - x \rangle + \frac{AL_c + B\, L_g\, \|\nabla f(x)\|}{2} \|y - x\|_2^2 \,,$$

where $A := 1 + e^q - \frac{e^q - 1}{q}$ and $B := \frac{e^q - 1}{q}$. Combining the above inequality with Eq. (10),

$$f(y) \leq f(x) + \langle \nabla f(x), y - x \rangle + \frac{AL_c + B\, L_g\, \left[\left(8L_g + \sqrt{2L_c}\right) f(x) + \sqrt{2L_c}\right]}{2} \|y - x\|_2^2$$

$$= f(x) + \langle \nabla f(x), y - x \rangle + \frac{A\left(L_c + \frac{B}{A} L_g \sqrt{2L_c}\right) + B\, L_g \left(8L_g + \sqrt{2L_c}\right) f(x)}{2} \|y - x\|_2^2$$

$$\leq f(x) + \langle \nabla f(x), y - x \rangle + \frac{A\left(L_c + L_g \sqrt{2L_c}\right) + B\, L_g \left(8L_g + \sqrt{2L_c}\right) f(x)}{2} \|y - x\|_2^2 \qquad \text{(Since } B \leq A)$$

Hence, Assn. 2 (a) is satisfied with $L_0 = L_c + L_g \sqrt{2L_c}$ and $L_1 = L_g \left(8L_g + \sqrt{2L_c}\right)$. $\qquad \square$

In order to prove that commonly used functions in machine learning satisfy the assumptions in Sec. 2, we will require the following lemma.

**Lemma 3.** *For a finite-sum objective, $f(\theta) = \frac{1}{n} \sum_{i=1}^{n} f_i(\theta)$, if, for all $i$, $f_i$ is non-negative and satisfies Assn. 5 with constants $L_c$ and $L_g$, then, $f$ satisfies Assn. 1 to 3 with constants $L_0 = L_c + L_g \sqrt{2L_c}$, $L_1 = L_g \left(8L_g + \sqrt{2L_c}\right)$, $\nu = 8L_g + \sqrt{2L_c}$ and $\omega = \sqrt{2L_c}$.*

*Proof.* If $f_i$ is non-negative for all $i$, then, $f$ is non-negative, thus satisfying Assn. 1. We will first prove that if $f_i$ is non-negative and satisfies Assn. 2 and 3, then $f$ also satisfies these assumptions with the same constants.

If $f_i$ satisfies Assn. 2 (a), then,

$$f_i(y) \leq f_i(x) + \langle \nabla f_i(x), y - x \rangle + \frac{AL_0 + B\, L_1\, f_i(x)}{2} \|y - x\|_2^2$$

Summing the LHS and RHS for $i = 1$ to $n$ and dividing by $n$ completes the proof that $f$ satisfies Assn. 2 (a). If $f_i$ satisfies Assn. 2 (b), then,

$$\|\nabla^2 f(\theta)\| = \left\| \frac{1}{n} \sum_i \nabla^2 f_i(\theta) \right\| \leq \frac{1}{n} \sum_i \|\nabla^2 f_i(\theta)\| \leq L_0 + \frac{L_1}{n} \sum_i f_i(\theta) = L_0 + L_1\, f(\theta)$$

Hence, $f$ satisfies Assn. 2 with the same constants $L_0$ and $L_1$. If $f_i$ satisfies Assn. 3, then,

$$\|\nabla f(\theta)\| = \left\| \frac{1}{n} \sum_i \nabla f_i(\theta) \right\| \leq \frac{1}{n} \sum_i \|\nabla f_i(\theta)\| \leq \frac{\nu}{n} \sum_i f_i(\theta) + \omega = \nu\, f(\theta) + \omega\,.$$

Hence, $f$ satisfies Assn. 3 with the same constants $\nu$ and $\omega$. From Prop. 3, if $f_i$ satisfies Assn. 5 with constants $L_c, L_g$, then, $f_i$ and consequently $f$ satisfies Assn. 2 and 3 with constants $L_0 = L_c + L_g \sqrt{2L_c}$, $L_1 = L_g \left(8L_g + \sqrt{2L_c}\right)$, $\nu = 8L_g + \sqrt{2L_c}$ and $\omega = \sqrt{2L_c}$. $\qquad \square$

### A.1. Examples satisfying Assn. 1 to 3

We will use the above lemma to prove that linear logistic regression, exponential loss with a linear model, linear multi-class classification using the cross-entropy loss, generalized linear models with a logistic link function and the softmax policy gradient objective for multi-armed bandits and tabular MDPs satisfy Assn. 1 to 3.

**Proposition 1.** *Consider $n$ points where $x_i \in \mathbb{R}^d$ are the features and $y_i \in \{-1, 1\}$ are the corresponding labels. Logistic regression with the objective*

$$f(\theta) = \frac{1}{n} \sum_{i=1}^{n} \ln(1 + \exp(-y_i \langle x_i, \theta \rangle)) \tag{3}$$

*satisfies Assn. 2 with $L_0 = 0$, $L_1 = 8 \max_{i \in [n]} \|x_i\|_2^2$, and Assn. 3 with $\nu = 8 \max_i \|x_i\|$, $\omega = 0$.*

*Proof.* Clearly, $f_i(\theta) \geq 0$ for all $\theta$. Calculating the gradient and hessian for $f_i(\theta) := \ln(1 + \exp(-y_i \langle x_i, \theta \rangle))$,

$$\nabla f_i(\theta) = \frac{-\exp(-y_i \langle x_i, \theta \rangle)}{1 + \exp(-y_i \langle x_i, \theta \rangle)} y_i \, x_i \quad ; \quad \nabla^2 f_i(\theta) = \frac{1}{1 + \exp(-y_i \langle x_i, \theta \rangle)} \frac{\exp(-y_i \langle x_i, \theta \rangle)}{1 + \exp(-y_i \langle x_i, \theta \rangle)} y_i^2 \, x_i \, x_i^T$$

Bounding the Hessian,

$$\nabla^2 f_i(\theta) \preceq \frac{\exp(-y_i \langle x_i, \theta \rangle)}{1 + \exp(-y_i \langle x_i, \theta \rangle)} y_i^2 x_i x_i^T = \frac{\exp(-y_i \langle x_i, \theta \rangle)}{1 + \exp(-y_i \langle x_i, \theta \rangle)} x_i x_i^T \quad \text{(For all } x, \frac{1}{1+e^x} \leq 1 \text{ and } y_i^2 = 1\text{)}$$

$$\implies \left\| \nabla^2 f_i(\theta) \right\| \leq \frac{\exp(-y_i \langle x_i, \theta \rangle)}{1 + \exp(-y_i \langle x_i, \theta \rangle)} \|x_i\|_2^2 = \|x_i\| \, \|\nabla f_i(\theta)\|$$

Hence, for all $i$, $f_i$ satisfies Assn. 5 with $L_c = 0$ and $L_g = \max_i \|x_i\|$. Using Lemma 3, we conclude that $f(\theta)$ satisfies Assn. 2 with $L_0 = 0$ and $L_1 = 8 \max_i \|x_i\|_2^2$, and Assn. 3 with $\nu = 8 \max_i \|x_i\|$ and $\omega = 0$. $\qquad\square$

**Proposition 2.** *Consider $n$ points where $x_i \in \mathbb{R}^d$ are the features and $y_i \in [0, 1]$ are the corresponding labels. If $\pi_i(\theta) = \sigma(\langle x_i, \theta \rangle) := \frac{1}{1 + \exp(-\langle x_i, \theta \rangle)}$, the GLM objective,*

$$f(\theta) = \frac{1}{2n} \sum_{i=1}^{n} (\pi_i(\theta) - y_i)^2 \, , \tag{4}$$

*satisfies Assn. 2 with $L_0 = \frac{9}{16} \max_{i \in [n]} \|x_i\|_2^2$, $L_1 = 9 \max_{i \in [n]} \|x_i\|_2^2$ and Assn. 3 with $\nu = 9 \max_i \|x_i\|$, $\omega = \max_i \|x_i\|$.*

*Proof.* Clearly, $f_i(\theta) \geq 0$ and hence $f(\theta) \geq 0$ for all $\theta$. $f(\theta)$ is a finite-sum objective. Calculating the gradient and hessian for $f_i(\theta) = \frac{1}{2} (\pi_i(\theta) - y_i)^2$,

$$\nabla f_i(\theta) = (\pi_i(\theta) - y_i) \frac{1}{1 + \exp(-\langle x_i, \theta \rangle)} \frac{\exp(-\langle x_i, \theta \rangle)}{1 + \exp(-\langle x_i, \theta \rangle)} x_i$$

$$\nabla^2 f_i(\theta) = [1 - 2\pi_i(\theta)] \pi_i(\theta) [1 - \pi_i(\theta)] [\pi_i(\theta) - y_i] x_i x_i^T + [\pi_i(\theta)]^2 [1 - \pi_i(\theta)]^2 x_i x_i^T$$

$$\implies \left\| \nabla^2 f_i(\theta) \right\| = \left[ \underbrace{|1 - 2\pi_i(\theta)|}_{\leq 1} \underbrace{\pi_i(\theta) [1 - \pi_i(\theta)] |\pi_i(\theta) - y_i| \|x_i\|}_{=\|\nabla f_i(\theta)\|} + \underbrace{[\pi_i(\theta)]^2 [1 - \pi_i(\theta)]^2 \|x_i\|}_{\leq \frac{1}{16}} \right] \|x_i\|$$

$$\text{(Triangle Inequality)}$$

$$\implies \left\| \nabla^2 f_i(\theta) \right\| \leq \|x_i\| \, \|\nabla f_i(\theta)\| + \frac{1}{16} \|x_i\|_2^2$$

Hence, for all $i$, $f_i(\theta)$ satisfies Assn. 5 with $L_c = \frac{1}{16} \max_{i \in [n]} \|x_i\|_2^2$ and $L_g = \max_{i \in [n]} \|x_i\|$.

Using Lemma 3, we conclude that $f(\theta)$ satisfies Assn. 2 with $L_0 = \frac{8 + \sqrt{2}}{16\sqrt{2}} \max_{i \in [n]} \|x_i\|_2^2$ and $L_1 = \frac{16\sqrt{2} + 1}{\sqrt{8}} \max_{i \in [n]} \|x_i\|_2^2$, and Assn. 3 with $\nu = \frac{16\sqrt{2} + 1}{\sqrt{8}} \max_i \|x_i\|$ and $\omega = \frac{1}{\sqrt{8}} \max_i \|x_i\|$. $\qquad\square$

**Proposition 5.** *Consider $n$ points where $x_i \in \mathbb{R}^d$ are the features and $y_i \in \{0,1\}^C$ are the corresponding one-hot label vectors for $C$ classes. Multi-class classification with the cross-entropy objective is given as:*

$$f(\theta) = \frac{1}{n} \sum_{m=1}^{n} KL(y^m || \pi_\theta^m), \text{ where } \forall m \in [n], \pi_\theta^m \in \Delta_C \text{ s.t. } \forall i \in [C], \pi_\theta^m(i) = \frac{\exp(\langle x_m, \theta_i \rangle)}{\sum_{k=1}^{C} \exp(\langle x_m, \theta_k \rangle)},$$

*where $\theta_i \in \mathbb{R}^d$ for $i \in [C]$ and $\theta = [\theta_1, \theta_2, \ldots, \theta_C]$. Multi-class logistic regression satisfies Assn. 2 with $L_0 = 0$ and $L_1 = 32 \max_{m \in [n]} \|x\|_1^2$, and Assn. 3 with $\nu = 16 \max_i \|x_i\|$ and $\omega = 0$.*

*Proof.* Let us consider a single input-output pair $(x, y)$ and calculate the gradient for a single function in the finite-sum. Define $\ell(\theta) := KL(y || \pi_\theta)$ where $y$ is a $C$-dimensional one-hot vector, $x \in \mathbb{R}^d$ and $\pi_\theta \in \Delta_C$ s.t. $\pi_\theta(i) = \frac{\exp(\langle x, \theta_i \rangle)}{\sum_{k=1}^{C} \exp(\langle x, \theta_k \rangle)}$.
Clearly, $\ell(\theta) \geq 0$. Calculating its gradient and Hessian,

$$\frac{\partial \ell(\theta)}{\partial \theta_i} = [\pi_\theta(i) - y_i] x.$$

The Hessian can be written as a Kronecker product of a $C \times C$ matrix which corresponds to the Jacobian of the softmax function, and a $d \times d$ rank-one matrix formed using the features. Specifically,

$$\nabla^2 \ell(\theta) = \underbrace{H}_{C \times C} \underbrace{xx^T}_{d \times d} \text{ where, } H := \text{diag}(\pi_\theta) - \pi_\theta \, \pi_\theta^T$$

$$\implies \left\| \nabla^2 \ell(\theta) \right\| \leq \|x\|_2^2 \, \|H\|$$

Since $H$ is a square symmetric PSD matrix, $\|H\| = \lambda_{\max}[H]$. By the Gershgorin circle theorem, $\lambda_{\max}[H] \leq \max_i \sum_{j=1}^{C} |H_{i,j}|$. Calculating the row sums, we conclude that $\|H\| \leq \lambda_{\max}[H] \leq 2 \max_i \pi_\theta(i) (1 - \pi_\theta(i))$. Hence,

$$\left\| \nabla^2 \ell(\theta) \right\| \leq 2 \, \|x\|_2^2 \, \max_i \pi_\theta(i) (1 - \pi_\theta(i)) \leq 2 \, \|x\| \, \frac{\max_i \pi_\theta(i) (1 - \pi_\theta(i))}{\sum_{i=1}^{C} |\pi_\theta(i) - y_i|} \, \|\nabla \ell(\theta)\|_1$$

Let $j^* := \arg\max \pi_\theta(i) (1 - \pi_\theta(i))$. Using that $\sum_{i=1}^{C} |\pi_\theta(i) - y_i| \geq |\pi_\theta(j^*) - y_{j^*}|$ and $\|x\|_2 \leq \|x\|_1$,

$$\leq 2 \, \|x\|_1 \, \frac{\pi_\theta(j^*) (1 - \pi_\theta(j^*))}{|\pi_\theta(j^*) - y_{j^*}|} \, \|\nabla \ell(\theta)\|_1$$

$$\implies \left\| \nabla^2 \ell(\theta) \right\| \leq 2 \, \|x\|_1 \, \|\nabla \ell(\theta)\|_1 \qquad \text{(Since } y_{j^*} \in \{0,1\} \text{ and } \pi_\theta(j^*) \in [0,1])$$

Hence, for a single $(x, y)$ pair, we can conclude that, $\ell(\theta)$ satisfies Assn. 5 with $L_g = 2 \|x\|_1$. Hence, $f_i$ satisfies Assn. 5 with $L_c = 0$ and $L_g = 2 \max_{i \in [n]} \|x_i\|_1$.

Using Lemma 3, we can conclude that $f(\theta)$ satisfies Assn. 2 with $L_0 = 0$ and $L_1 = 32 \max_{i \in [n]} \|x_i\|_1^2$, and Assn. 3 with $\nu = 16 \max_i \|x_i\|$ and $\omega = 0$. $\qquad \square$

**Proposition 6.** *Consider $n$ points where $x_i \in \mathbb{R}^d$ are the features and $y_i \in \{0,1\}$ are the corresponding labels. Binary classification with an exponential loss with the objective*

$$f(\theta) := \frac{1}{n} \sum_{i=1}^{n} \exp(-y_i \langle x_i, \theta \rangle),$$

*satisfies Assn. 2 with $L_0 = 0$ and $L_1 = 8 \max_{i \in [n]} \|x_i\|_2^2$, and Assn. 3 with $\nu = 8 \max_i \|x_i\|$ and $\omega = 0$.*

*Proof.* Clearly, $f_i(\theta) \geq 0$ and hence $f(\theta) \geq 0$ for all $\theta$. Calculating the gradient and hessian for $f_i(\theta) := \exp(-y_i \langle x_i, \theta \rangle)$,

$$\nabla f_i(\theta) = -\exp(-y_i \langle x_i, \theta \rangle) y_i \, x_i$$

$$\nabla^2 f_i(\theta) = \exp(-y_i \langle x_i, \theta \rangle) \, y_i^2 \, x_i \, x_i^T = \exp(-y_i \langle x_i, \theta \rangle) \, x_i \, x_i^T \qquad (y_i^2 = 1)$$

$$\implies \left\| \nabla^2 f_i(\theta) \right\| \leq \|x_i\| \, \|\nabla f_i(\theta)\|$$

Hence, for all $i$, $f_i$ satisfies Assn. 5 with $L_c = 0$ and $L_g = \max_i \|x_i\|$. Using Lemma 3, we conclude that $f(\theta)$ satisfies Assn. 2 with $L_0 = 0$ and $L_1 = 8 \max_i \|x_i\|_2^2$ and Assn. 3 with $\nu = 8 \max_i \|x_i\|$ and $\omega = 0$. $\qquad \square$

**Proposition 4.** *Given an MAB problem with $K$ arms and known deterministic rewards $r \in [0, 1]^K$, consider the class of softmax policies $\pi_\theta \in \Delta_K$ parameterized by $\theta \in \mathbb{R}^K$ s.t. $\pi_\theta(a) = \frac{\exp(\theta(a))}{\sum_{a'} \exp(\theta(a'))}$. The loss corresponding to the bandit problem is given by: $f(\theta) = r(a^*) - \langle \pi_\theta, r \rangle$, where $a^* := \arg\max_{a \in [K]} r(a)$ is the optimal arm. $f(\theta)$ is non-negative, satisfies Assn. 2 with $L_0 = 0$ and $L_1 = 72$, Assn. 3 with $\nu = 24$ and $\omega = 0$ and Assn. 4 with $\zeta = 1$ and $\mu(\theta) = \pi_\theta(a^*)$.*

*Proof.* From Mei et al. (2021, Lemma 2), we know that

$$\left\| \nabla^2 \ell(\theta) \right\| = \left\| \nabla^2 \langle \pi_\theta, r \rangle \right\| \leq 3 \left\| \nabla \langle \pi_\theta, r \rangle \right\| = 3 \left\| \nabla \ell(\theta) \right\|$$

Hence, the loss for the bandit problem satisfies Assn. 5 with $L_c = 0$ and $L_g = 3$. Using Lemma 3 with $n = 1$, we can conclude the the loss for the bandit problem satisfies Assn. 2 with $L_0 = 0$ and $L_1 = 72$ and Assn. 3 with $\nu = 24$ and $\omega = 0$. From Mei et al. (2020, Lemma 3), we know that,

$$\left\| \nabla \ell(\theta) \right\| = \left\| \nabla_\theta \langle \pi_\theta, r \rangle \right\| \geq \pi_\theta(a^*) \left[ r(a^*) - \langle \pi_t, r \rangle \right] = \pi_\theta(a^*) \, f(\theta)$$

Hence, the loss for the bandit problem satisfies Assn. 4 with $\mu(\theta) = \pi_\theta(a^*)$. $\qquad\square$

**Proposition 7.** *Consider an infinite-horizon discounted Markov decision process (MDP) defined by $\langle \mathcal{S}, \mathcal{A}, \mathcal{P}, r, \rho, \gamma \rangle$, where $\mathcal{S}$ and $\mathcal{A}$ represent the states and actions, $\mathcal{P} : \mathcal{S} \times \mathcal{A} \to \Delta_{\mathcal{S}}$ is the transition probability function, $r : \mathcal{S} \times \mathcal{A} \to [0, 1]$ is the reward function, $\rho \in \Delta_{\mathcal{S}}$ is the initial state distribution, and $\gamma \in [0, 1)$ represents the discount factor. If $V^\pi(s) := \mathbb{E}[\sum_{t=0}^\infty \gamma^t r(s_t, a_t) | s_0 = s]$ where $s_t \sim p(.|s_{t-1}, a_{t-1})$, and $a_t \sim \pi(.|s_t)$ for $t \geq 1$ is the expected discounted cumulative reward for a policy $\pi$ starting at state $s$, we define $V^\pi(\rho) := \mathbb{E}_{s \sim \rho}[V^\pi(s)]$.*

*Consider a policy $\pi_\theta$ parameterized by $\theta \in \mathbb{R}^{|\mathcal{S}| \times |\mathcal{A}|}$ s.t. $\pi_\theta(s, \cdot) \in \Delta_K$ for all $s \in \mathcal{S}$ and $\pi_\theta(s, a) \propto \exp(\theta(s, a))$. The loss corresponding to the tabular MDP problem is given by:*

$$f(\theta) = V^{\pi^*}(\rho) - V^{\pi_\theta}(\rho),$$

*where $\pi^*$ is the optimal policy. $f(\theta)$ satisfies Assn. 2 with $L_0 = 0$ and $L_1 = 8 \left[ 3 + \frac{4 \cdot \left( \min_s \frac{1}{\rho(s)} - (1-\gamma) \right)}{1-\gamma} \right]^2 S$, Assn. 3 with $\nu = 8 \left[ 3 + \frac{4 \cdot \left( \min_s \frac{1}{\rho(s)} - (1-\gamma) \right)}{1-\gamma} \right] \sqrt{S}$ and $\omega = 0$ and Assn. 4 with $\mu(\theta) = \frac{\min_s \pi_\theta(a^*(s)|s)}{\sqrt{S} \, \min_{s \in \mathcal{S}} \rho(s)}$ where $a^*(s)$ is the action that a deterministic optimal policy $\pi^*$ selects in state $s$.*

*Proof.* Assuming that the starting state distribution has full support, i.e. $\rho(s) > 0$, from Mei et al. (2021, Lemma 6), we know that,

$$\left\| \nabla^2 f(\theta) \right\| \leq \left[ 3 + \frac{4 \cdot \left( \min_s \frac{1}{\rho(s)} - (1-\gamma) \right)}{1-\gamma} \right] \cdot \sqrt{S} \cdot \left\| \nabla f(\theta) \right\|$$

Hence, the loss for the tabular MDP problem satisfies Assn. 5 with $L_c = 0$ and $L_g = \left[ 3 + \frac{4 \cdot \left( \min_s \frac{1}{\rho(s)} - (1-\gamma) \right)}{1-\gamma} \right] \sqrt{S}$. Using Lemma 3 with $n = 1$, we can conclude the the loss for the tabular MDP problem satisfies Assn. 2 with $L_0 = 0$ and $L_1 = 8 \left[ 3 + \frac{4 \cdot \left( \min_s \frac{1}{\rho(s)} - (1-\gamma) \right)}{1-\gamma} \right]^2 S$ and Assn. 3 with $\nu = 8 \left[ 3 + \frac{4 \cdot \left( \min_s \frac{1}{\rho(s)} - (1-\gamma) \right)}{1-\gamma} \right] \cdot \sqrt{S}$ and $\omega = 0$. From Mei et al. (2020, Lemma 8), we know that

$$\left\| \nabla f(\theta) \right\| \geq \frac{\min_s \pi_\theta(a^*(s)|s)}{\sqrt{S} \, \min_{s \in \mathcal{S}} \rho(s)} \, f(\theta)$$

Hence, the loss for the tabular MDP problem satisfies Assn. 4 with $\mu(\theta) = \frac{\min_s \pi_\theta(a^*(s)|s)}{\sqrt{S} \, \min_{s \in \mathcal{S}} \rho(s)}$. $\qquad\square$

**Proposition 8.** *Consider the logistic regression objective in Eq. (3) with $n = 2$ and $d = 1$. Consider the two points to be such that $y_1 x_1 = 2$ and $y_2 x_2 = -2$. For this problem, the non-uniformness assumption in Zhang et al. (2019): $\left\| \nabla^2 f(\theta) \right\| \leq L_0 + L_1 \left\| \nabla f(\theta) \right\|$ cannot hold for $L_0 = 0$ and any $L_1 \neq 0$.*

*Proof.* Using the proof of Prop. 1 to calculate the gradient and hessian, we get that $\nabla f_1(0) = 0$ and $\nabla f_2(0) = 0$ which implies $\nabla f(0) = 0$. Similarly, for Hessian, we get $\nabla^2 f_1(0) = 1$ and $\nabla^2 f_2(0) = 1$ which implies $\nabla^2 f(0) = 1$. Since $\nabla f(\theta) = 0$ and $\nabla^2 f(\theta) \neq 0$, the assumption cannot hold with $L_0 \neq 0$ and any $L_1 \neq 0$. □

## B. Proofs for Sec. 3

**Lemma 1.** *If $f$ satisfies Assn. 1 to 3, at iteration $t$, GD-LS returns a step-size $\eta_t \geq \min\left\{\eta_{\max}, \frac{1}{\lambda_0 + \lambda_1 f(\theta_t)}\right\}$, where* $\lambda_0 := 3\frac{L_0 + L_1\omega}{(1-c)}$ *and* $\lambda_1 := 3\frac{L_1(\nu+1)}{(1-c)}$.

*Proof.* **Case 1**: If $L_1 = 0$, Assn. 2 is equivalent to the standard $L_0$-uniform smoothness condition. In this case, we can follow the standard analysis of GD-LS (Nocedal & Wright, 2006) and conclude that $\eta_t \geq \min\left\{\eta_{\max}, \frac{2(1-c)}{L_0}\right\} \geq \min\left\{\eta_{\max}, \frac{(1-c)}{3L_0}\right\}$.

In this special case, $\lambda_0 = \frac{3L_0}{1-c}$ and $\lambda_1 = 0$, meaning that $\eta_t \geq \min\left\{\eta_{\max}, \frac{1}{\lambda_0 + \lambda_1 f(\theta_t)}\right\}$. This concludes the proof.

**Case 2**: If $L_1 \neq 0$ and since $f(\theta)$ is non-negative, we define the log-loss as follows.

$$g(\theta) := \ln(L_0 + L_1 f(\theta))$$

Using Assn. 2, $\nabla^2 f(\theta) \preceq [L_0 + L_1 f(\theta)] I_d$. Using this result, we bound the Hessian of $g(\theta)$.

$$\nabla g(\theta) = \frac{L_1 \nabla f(\theta)}{L_0 + L_1 f(\theta)}$$

$$\nabla^2 g(\theta) = \frac{L_1 \nabla^2 f(\theta)}{L_0 + L_1 f(\theta)} - \frac{L_1^2 [\nabla f(\theta)][\nabla f(\theta)]^T}{(L_0 + L_1 f(\theta))^2} \preceq \frac{L_1 \nabla^2 f(\theta)}{L_0 + L_1 f(\theta)} \quad \text{(Since the second term is PSD)}$$

$$\implies \nabla^2 g(\theta) \preceq L_1 I_d$$

Hence, $g(\theta)$ is $L_1$-globally smooth. Using this result, we know that for all $u, v$,

$$g(u) \leq g(v) + \langle \nabla g(v), u - v \rangle + \frac{L_1}{2}\|u - v\|_2^2$$

Using this result for $u = \theta_{t+1}$ and $v = \theta_t$,

$$g(\theta_{t+1}) \leq g(\theta_t) + \langle \nabla g(\theta_t), \theta_{t+1} - \theta_t \rangle + \frac{L_1}{2}\|\theta_{t+1} - \theta_t\|_2^2$$

$$= g(\theta_t) - \eta_t \langle \nabla f(\theta_t), \nabla g(\theta_t) \rangle + \frac{L_1 \eta_t^2}{2}\|\nabla f(\theta_t)\|_2^2$$

$$\text{(Using the update that } \theta_{t+1} = \theta_t - \eta_t \nabla f(\theta_t))$$

$$= g(\theta_t) - \eta_t \left\langle \nabla f(\theta_t), \frac{L_1 \nabla f(\theta_t)}{L_0 + L_1 f(\theta_t)} \right\rangle + \frac{L_1 \eta_t^2}{2}\|\nabla f(\theta_t)\|_2^2 \quad \text{(Since } \nabla g(\theta) = \frac{L_1 \nabla f(\theta)}{L_0 + L_1 f(\theta)})$$

$$\implies g(\theta_t - \eta_t \nabla f(\theta_t)) \leq \underbrace{g(\theta_t) - \eta_t \frac{L_1 \|\nabla f(\theta_t)\|_2^2}{L_0 + L_1 f(\theta_t)} + \frac{L_1 \eta_t^2}{2}\|\nabla f(\theta_t)\|_2^2}_{:=h_Q(\eta_t)} \quad (12)$$

Next, we will compare the above inequality with what we obtain from the Armijo line-search.

$$f(\theta_t - \eta_t \nabla f(\theta_t)) \leq f(\theta_t) - c\eta_t \|\nabla f(\theta_t)\|_2^2$$

$$L_0 + L_1 f(\theta_t - \eta_t \nabla f(\theta_t)) \leq L_0 + L_1 f(\theta_t) - c\eta_t L_1 \|\nabla f(\theta_t)\|_2^2$$

Note that $L_0 + L_1 f(\theta_t) - c\eta_t L_1 \|\nabla f(\theta_t)\|_2^2 \geq 0$. Since $f, L_0, L_1 \geq 0$, $1 - c\eta_t \frac{L_1 \|\nabla f(\theta_t)\|_2^2}{L_0 + L_1 f(\theta_t)} \geq 0$. Taking log on both

sides,

$$\ln\left(L_0 + L_1\,f(\theta_t - \eta_t\nabla f(\theta_t))\right) \leq \ln\left(L_0 + L_1\,f(\theta_t) - c\eta_t\,L_1\,\|\nabla f(\theta_t)\|_2^2\right) \quad \text{(Since } \ln \text{ is monotonically increasing)}$$

$$\implies g(\theta_t - \eta_t\nabla f(\theta_t)) \leq \ln\left(L_0 + L_1\,f(\theta_t) - c\eta_t\,L_1\,\|\nabla f(\theta_t)\|_2^2\right) \quad \text{(By definition of } g\text{)}$$

$$= \ln\left((L_0 + L_1\,f(\theta_t))\left(1 - c\eta_t\,\frac{L_1\,\|\nabla f(\theta_t)\|_2^2}{L_0 + L_1\,f(\theta_t)}\right)\right) \tag{13}$$

$$= g(\theta_t) + \ln\left(1 - c\eta_t\,\frac{L_1\,\|\nabla f(\theta_t)\|_2^2}{L_0 + L_1\,f(\theta_t)}\right) \quad \text{(By definition of } g\text{)}$$

$$\leq g(\theta_t) + \left(1 - c\eta_t\,\frac{L_1\,\|\nabla f(\theta_t)\|_2^2}{L_0 + L_1\,f(\theta_t)}\right) - 1 \quad \text{(For all } x > 0,\ \ln(x) \leq x - 1\text{)}$$

$$\implies g(\theta_{t+1}) \leq \underbrace{g(\theta_t) - c\eta_t\,\frac{L_1\,\|\nabla f(\theta_t)\|_2^2}{L_0 + L_1\,f(\theta_t)}}_{:=h_L(\eta_t)} \tag{14}$$

Hence, assuming exact back-tracking, if $\eta_t$ is a step-size that satisfies Eq. (5), then Eq. (14) will also be satisfied.

If the Armijo condition is satisfied for an $\eta_t$ s.t. $h_L(\eta_t) \leq h_Q(\eta_t)$, then,

$$g(\theta_t) - c\eta_t\,\frac{L_1\,\|\nabla f(\theta_t)\|_2^2}{L_0 + L_1\,f(\theta_t)} \leq g(\theta_t) - \eta_t\,\frac{L_1\,\|\nabla f(\theta_t)\|_2^2}{L_0 + L_1\,f(\theta_t)} + \frac{L_1\,{\eta_t}^2}{2}\,\|\nabla f(\theta_t)\|_2^2$$

$$\implies \eta_t \geq \frac{2\,(1-c)}{L_0 + L_1\,f(\theta_t)}$$

If the Armijo condition is satisfied for an $\eta_t$ s.t. $h_Q(\eta_t) \leq h_L(\eta_t)$, it implies that $\eta_t \leq \frac{2(1-c)}{L_0 + L_1 f(\theta_t)}$.

However, we show that the resulting step-size cannot be too small. In particular, we will prove that the Armijo condition is satisfied for

$$\eta_t = \frac{2(1-c)}{6(L_0 + L_1\omega) + 6\,L_1\,(\nu+1)\,f(\theta_t)}.$$

To show this, we use Assn. 2. In order to use this inequality, we have to ensure that $\|\theta_{t+1} - \theta_t\| = \eta_t\,\|\nabla f(\theta_t)\| \leq \frac{q}{L_1}$. Since based on Assn. 3, $\|\nabla f(\theta_t)\| \leq \nu\,f(\theta_t) + \omega$, it suffices to ensure that $q \geq \eta_t\,L_1(\nu\,f(\theta_t) + \omega)$.

Using Assn. 2, we get that:

$$f(\theta_{t+1}) \leq f(\theta_t) - \eta_t\,\|\nabla f(\theta_t)\|_2^2 + \frac{AL_0 + B\,L_1\,f(\theta_t)}{2}\,{\eta_t}^2\,\|\nabla f(\theta_t)\|_2^2$$

The Armijo condition is definitely satisfied if:

$$f(\theta_t) - \eta_t\,\|\nabla f(\theta_t)\|_2^2 + \frac{\left(1 + e^q - \frac{e^q-1}{q}\right)L_0 + \left(\frac{e^q-1}{q}\right)L_1\,f(\theta_t)}{2}\,{\eta_t}^2\,\|\nabla f(\theta_t)\|_2^2 \leq f(\theta_t) - c\eta_t\,\|\nabla f(\theta_t)\|_2^2$$

Hence, the Armijo condition is satisfies for all $\eta_t$ s.t.

$$\eta_t \leq \frac{2\,(1-c)}{\left(1 + e^q - \frac{e^q-1}{q}\right)L_0 + \left(\frac{e^q-1}{q}\right)L_1\,f(\theta_t)},$$

Since,

$$\frac{2\,(1-c)}{\left(1 + e^q - \frac{e^q-1}{q}\right)L_0 + \left(\frac{e^q-1}{q}\right)L_1\,f(\theta_t) + \left(\frac{e^q-1}{q}\right)L_1\,(\nu f(\theta_t) + \omega)} \leq \frac{2\,(1-c)}{\left(1 + e^q - \frac{e^q-1}{q}\right)L_0 + \left(\frac{e^q-1}{q}\right)L_1\,f(\theta_t)},$$

the Armijo condition will be satisfied for the smaller step-size.

Moreover, for

$$\eta_t' := \frac{2(1-c)}{\left(1 + e^q - \frac{e^q-1}{q}\right) L_0 + \left(\frac{e^q-1}{q}\right) L_1 f(\theta_t) + \left(\frac{e^q-1}{q}\right) L_1 (\nu f(\theta_t) + \omega)},$$

we need to ensure that $q \geq \eta_t' L_1(\nu f(\theta_t) + \omega)$. Hence, we want to find a $q$ s.t.

$$q \geq \frac{2(1-c) L_1(\nu f(\theta_t) + \omega)}{\left(1 + e^q - \frac{e^q-1}{q}\right) L_0 + \left(\frac{e^q-1}{q}\right) L_1 f(\theta_t) + \left(\frac{e^q-1}{q}\right) L_1 (\nu f(\theta_t) + \omega)}$$

Since $\left(1 + e^q - \frac{e^q-1}{q}\right) L_0 + \left(\frac{e^q-1}{q}\right) L_1 f(\theta_t) > 0$, it suffices to choose $q$ s.t.

$$\implies q \geq \frac{2(1-c) L_1(\nu f(\theta_t) + \omega)}{\left(\frac{e^q-1}{q}\right) L_1 (\nu f(\theta_t) + \omega)} = \frac{2(1-c)}{\left(\frac{e^q-1}{q}\right)}$$

Finally, since $1 + x \leq \exp(x)$ for all $x$, it suffices to choose $q$ s.t.

$$q \geq 2(1-c)$$

Hence, $q = 2$ satisfies the required conditions. Therefore, for $q = 2$ we have,

$$\eta_t' = \frac{2(1-c)}{\left(1 + e^2 - \frac{e^2-1}{2}\right) L_0 + \left(\frac{e^2-1}{2}\right) L_1 f(\theta_t) + \left(\frac{e^2-1}{2}\right) L_1 (\nu f(\theta_t) + \omega)}$$

Therefore for any $\eta_t \leq \eta_t'$, we have $q = 2 \geq \eta_t L_1 (\nu f(\theta_t) + \omega)$. Since $\frac{e^2-1}{2} \leq 6$ and $1 + e^2 - \frac{e^2-1}{2} \leq 6$ we can set

$$\eta_t = \frac{2(1-c)}{6 L_0 + 6 L_1 f(\theta_t) + 6 L_1 (\nu f(\theta_t) + \omega)} = \frac{2(1-c)}{6 (L_0 + L_1\omega) + 6 L_1(\nu + 1) f(\theta_t)}.$$

Based on above argument, we can conclude that the $\eta_t$, the step-size returned by the Armijo line-search is lower-bounded as

$$\eta_t \geq \frac{2(1-c)}{6 (L_0 + L_1\omega) + 6 L_1(\nu + 1) f(\theta_t)}.$$

Moreover if

$$\eta_{\max} \leq \frac{2(1-c)}{6 (L_0 + L_1\omega) + 6 L_1(\nu + 1) f(\theta_t)},$$

then $\eta_{\max}$ satisfies both Armijo condition and $h_Q(\eta_{\max}) \leq h_L(\eta_{\max})$, in which case, the line-search would terminate immediately and return $\eta_{\max}$. Therefore

$$\eta_t \geq \min\{\eta_{\max}, \frac{2(1-c)}{6 (L_0 + L_1\omega) + 6 L_1(\nu + 1) f(\theta_t)}\}.$$

$\square$

**Theorem 1.** *For a fixed $\epsilon > 0$, if $f$ satisfies Assn. 1 to 3, and if for a constant $R > 0$, $\|\nabla f(\theta_t)\|_2^2 \geq \frac{[f(\theta_t)-f^*]^2}{R}$ for all iterations $t \in [T]$, then, GD-LS with $\eta_{\max} = \infty$ requires*

$$T \geq \begin{cases} \max\{2 R\lambda_1, 1\} \left(\frac{f^*}{\epsilon} + 1\right) \ln\left(\frac{f(\theta_0)-f^*}{\epsilon}\right) \\ \text{if } f^* \geq \frac{\lambda_0}{\lambda_1} - \epsilon \quad \textbf{(Case (1))} \\ \\ \frac{2\lambda_0 R}{\epsilon} + \max\{2 R\lambda_1, 1\} \left(\frac{f^*}{\epsilon} + 1\right) \ln\left(\frac{f(\theta_0)-f^*}{\epsilon}\right) \\ \text{otherwise} \quad \textbf{(Case (2))} \end{cases}$$

*iterations to ensure that $f(\theta_T) - f^* \leq \epsilon$.*

*Proof.* Using the Armijo line-search condition in Eq. (5), and combining it with the lower-bound in Lemma 1,

$$f(\theta_{t+1}) \le f(\theta_t) - \frac{1}{\lambda_0 + \lambda_1 f(\theta_t)} \|\nabla f(\theta_t)\|_2^2 \tag{15}$$

We now follow a proof similar to that of Axiotis & Sviridenko (2023, Theorem 5.2) and derive a linear rate of convergence. From the theorem assumption, we know that $\|\nabla f(\theta_t)\|_2^2 \ge \frac{[f(\theta_t) - f^*]^2}{R}$. Combining these relations,

$$f(\theta_{t+1}) \le f(\theta_t) - \frac{1}{\lambda_0 + \lambda_1 f(\theta_t)} \frac{[f(\theta_t) - f^*]^2}{R}$$

Let us define $\tau := \max\{t \text{ s.t } \lambda_0 \le \lambda_1 f(\theta_t)\}$. Hence, for all $t \le \tau$, $f(\theta_t) \ge \frac{\lambda_0}{\lambda_1}$.

Consider two cases:
**Case (1)**: If $f^* \ge \frac{\lambda_0}{\lambda_1} - \epsilon$. Since $\{f(\theta_t)\}_{t=0}^{t=\tau}$ is monotonically decreasing due to the Armijo line-search and converging to $\frac{\lambda_0}{\lambda_1}$. Hence, there exists a $\tau'$ s.t. $\tau' \le \tau$ such that $f(\theta_{\tau'}) - f^* \le \epsilon$ and $f(\theta_{\tau'-1}) - f^* \ge \epsilon$, i.e. $\tau'$ is the iteration index when the desired sub-optimality criterion is satisfied for the first time. This implies that for all $t < \tau'$, $\delta_t := f(\theta_t) - f^* > \epsilon$. Hence, $\frac{f(\theta_{\tau'})}{f^*} \le \alpha := 1 + \frac{\epsilon}{f^*}$. Since $f^* > 0$ and $\epsilon > 0$, $\alpha > 1$. Hence for all $t < \tau'$,

$$\frac{f(\theta_t)}{f^*} > \alpha \implies \frac{\delta_t}{f(\theta_t)} = 1 - \frac{f^*}{f(\theta_t)} > 1 - \frac{1}{\alpha} > 0.$$

Using the condition of Case (1), we get

$$\delta_{t+1} \le \delta_t - \underbrace{\frac{1}{2\lambda_1 R}}_{:=\alpha} \frac{[f(\theta_t) - f^*]^2}{f(\theta_t)}$$

$$\le \delta_t - \bar{\alpha} \frac{[f(\theta_t) - f^*]^2}{f(\theta_t)} \qquad \text{(where } \bar{\alpha} := \max\{1, \alpha\}\text{)}$$

$$\le \delta_t - \bar{\alpha} \frac{[f(\theta_t) - f^*]}{f(\theta_t)} \delta_t$$

$$= \delta_t - \bar{\alpha} \left(1 - \frac{f^*}{f(\theta_t)}\right) \delta_t$$

Combining the above relations, for all $t < \tau'$,

$$\delta_{t+1} \le \left(1 - \underbrace{\bar{\alpha} \left(1 - \frac{1}{\alpha}\right)}_{:=\rho}\right) \delta_t$$

Since $\bar{\alpha} \in (0, 1)$ and $\left(1 - \frac{1}{\alpha}\right) \in (0, 1)$, $\rho := \bar{\alpha} \left(1 - \frac{1}{\alpha}\right) \in (0, 1)$. Recursing from $t = 0$ to $t = \tau' - 1$,

$$\delta_{\tau'} \le \exp(-\rho \tau') \delta_0$$

In order to ensure that $f(\theta_{\tau'}) - f^* \le \epsilon$, we require,

$$\tau' \ge \frac{1}{\rho} \ln\left(\frac{\delta_0}{\epsilon}\right) = \frac{1}{\min\{\alpha, 1\}} \left(\frac{f^*}{\epsilon} + 1\right) \ln\left(\frac{\delta_0}{\epsilon}\right) = \max\{2 R\lambda_1, 1\} \left(\frac{f^*}{\epsilon} + 1\right) \ln\left(\frac{f(\theta_0) - f^*}{\epsilon}\right)$$

**Case (2)**: If $f^* < \frac{\lambda_0}{\lambda_1} - \epsilon$. We will divide the subsequent analysis into two phases.

**Phase (1)**: For all $t \leq \tau$, s.t. $\lambda_0 + \lambda_1 f(\theta_t) \leq 2\lambda_1 f(\theta_t)$ holds, by a similar analysis as above, we can conclude that,

$$\delta_\tau \leq \exp\left(-\rho \tau\right) \delta_0 \implies f(\theta_\tau) - f^* \leq \exp\left(-\rho \tau\right) \left[f(\theta_0) - f^*\right]$$

Since $\delta_\tau = f(\theta_\tau) - f^* \geq \frac{\lambda_0}{\lambda_1} - f^* = \epsilon$. Hence,

$$\exp\left(-\rho \tau\right) \left[f(\theta_0) - f^*\right] \geq \epsilon \implies \tau \leq \frac{1}{\rho} \ln\left(\frac{f(\theta_0) - f^*}{\epsilon}\right)$$

**Phase (2)**: For all $t > \tau$, $\lambda_0 \geq \lambda_1 f(\theta_t)$ which implies $\lambda_0 + \lambda_1 f(\theta_t) \leq 2\lambda_0$. In this case,

$$f(\theta_{t+1}) - f^* \leq \underbrace{\left[f(\theta_t) - f^*\right]}_{:=\delta_t} - \frac{1}{2\lambda_0 R}\left[f(\theta_t) - f^*\right]^2 \implies \delta_{t+1} \leq \delta_t - \frac{1}{2\lambda_0 R}\delta_t^2$$

Following the standard approach, we divide both sides by $\delta_{t+1}\delta_t$ and rearranging we get

$$\frac{1}{2\lambda_0 R} \leq \frac{1}{2\lambda_0 R}\frac{\delta_t}{\delta_{t+1}} \qquad \text{(since } \frac{\delta_t}{\delta_{t+1}} \geq 1\text{)}$$

$$\leq \frac{1}{\delta_{t+1}} - \frac{1}{\delta_t}$$

Summing the above for $t = \tau$ to $t = T - 1$, we get

$$\frac{T - \tau}{2\lambda_0 R} \leq \frac{1}{\delta_T} - \frac{1}{\delta_\tau}$$

$$\implies \delta_T \leq \frac{1}{\frac{T-\tau}{2\lambda_0 R} + \frac{1}{\delta_\tau}}$$

We need to find $T$ such that $\delta_T \leq \epsilon$, which means

$$\frac{1}{\frac{T-\tau}{2\lambda_0 R} + \frac{1}{\delta_\tau}} \leq \epsilon \implies T - \tau \geq \frac{2\lambda_0 R}{\epsilon} - \frac{2\lambda_0 R}{\delta_\tau} \implies T \geq \frac{2\lambda_0 R}{\epsilon} + \frac{1}{\rho}\ln(\delta_0/\epsilon)$$

Putting everything together,

$$T \geq \frac{2\lambda_0 R}{\epsilon} + \max\{2 R\lambda_1, 1\}\left(\frac{f^*}{\epsilon} + 1\right)\ln\left(\frac{f(\theta_0) - f^*}{\epsilon}\right)$$

$\square$

## C. Proofs for Sec. 4

**Corollary 1.** *For a fixed $\epsilon > 0$, assuming $f(\theta)$ is convex and satisfies Assn. 1 to 3 with $L_0 = 0$ and $\omega = 0$, `GD-LS` with $\eta_{\max} = \infty$, requires $T \geq \max\{2\lambda_1 \|\theta_0 - \theta^*\|_2^2, 1\}\left(\frac{f^*}{\epsilon} + 1\right)\ln\left(\frac{f(\theta_0) - f^*}{\epsilon}\right)$ iterations to ensure that $f(\theta_T) - f^* \leq \epsilon$.*

*Proof.* Using the convexity of $f$,

$$f(\theta_t) - f^* \leq \langle \nabla f(\theta_t), \theta_t - \theta^* \rangle \leq \|\nabla f(\theta_t)\| \|\theta_t - \theta^*\|$$

$$\implies \|\nabla f(\theta_t)\|_2^2 \geq \frac{[f(\theta_t) - f^*]^2}{\|\theta_t - \theta^*\|_2^2}$$

Next, we show that $\|\theta_{t+1} - \theta^*\| \leq \|\theta_t - \theta^*\|$ for all $t$, and hence $\|\theta_t - \theta^*\| \leq \|\theta_0 - \theta^*\|$.

$$\|\theta_{t+1} - \theta^*\|_2^2 = \|\theta_t - \theta^* - \eta_t \nabla f(\theta_t)\|_2^2 = \|\theta_t - \theta^*\|_2^2 - 2\eta_t \langle \nabla f(\theta_t), \theta_t - \theta^* \rangle + \eta_t^2 \|\nabla f(\theta_t)\|_2^2$$

$$\leq \|\theta_t - \theta^*\|_2^2 - 2\eta_t[f(\theta_t) - f^*] + \eta_t^2 \|\nabla f(\theta_t)\|_2^2 \qquad \text{(By convexity of } f\text{)}$$

$$\leq \|\theta_t - \theta^*\|_2^2 - 2\eta_t[f(\theta_t) - f^*] + 2\eta_t [f(\theta_t) - f(\theta_{t+1})]$$

$$\text{(Using the Armijo line-search with } c = \frac{1}{2}\text{)}$$

$$\implies \|\theta_{t+1} - \theta^*\|_2^2 \leq \|\theta_t - \theta^*\|_2^2 - 2\eta_t [f(\theta_{t+1}) - f^*] + 2\eta_t [f(\theta_t) - f^*]$$

$$\leq \|\theta_t - \theta^*\|_2^2 \qquad \text{(By the definition of } \theta^* \text{ and using that for all } t, f^* < f(\theta_T) \leq f(\theta_t)\text{)}$$

Combining the above inequalities,

$$\|\nabla f(\theta_t)\|_2^2 \geq \frac{[f(\theta_t) - f^*]^2}{\|\theta_0 - \theta^*\|_2^2} \tag{16}$$

Using Thm. 1 with $R = \|\theta_0 - \theta^*\|_2^2$ and setting $L_0 = 0$ completes the proof. $\qquad\square$

**Theorem 2.** *For an initialization $\theta_0$, $\epsilon \in (0, f(\theta_0))$ and comparator $u$ s.t. $f(u) \leq \epsilon$, if $f(\theta)$ is convex, satisfies Assn. 1 to 3 with $L_0 = 0$, $\omega = 0$, then, $\mathtt{GD\text{-}LS}$ with $\eta_{\max} = \infty$, $c > \frac{1}{2}$, requires $T \geq \frac{c\,\lambda_1\,\|\theta_0 - u\|_2^2}{(2c-1)} \left[1 + \frac{f(u)}{\epsilon}\right]$ iterations to ensure that $f(\theta_T) - f(u) \leq \epsilon$.*

*Proof.* For an arbitrary comparator $u$ s.t. $f(u) \leq f(\theta_T)$,

$$\|\theta_{t+1} - u\|_2^2 = \|\theta_t - u\|_2^2 - 2\,\eta_t\,\langle\nabla f(\theta_t), \theta_t - u\rangle + \eta_t^2\,\|\nabla f(\theta_t)\|_2^2 \leq \|\theta_t - u\|_2^2 - 2\,\eta_t\,[f(\theta_t) - f(u)] + \eta_t^2\,\|\nabla f(\theta_t)\|_2^2$$
$$\text{(Convexity)}$$

$$\leq \|\theta_t - u\|_2^2 - 2\,\eta_t\,[f(\theta_t) - f(u)] + \frac{\eta_t}{c}\,[f(\theta_t) - f(\theta_{t+1})]$$
$$\text{(Using the Armijo line-search with } c > \tfrac{1}{2})$$

$$\leq \|\theta_t - u\|_2^2 - 2\,\eta_t\,[f(\theta_t) - f(u)] + \frac{\eta_t}{c}\,[f(\theta_t) - f(u)]$$
$$\text{(Since } f(u) \leq f(\theta_t))$$

$$= \|\theta_t - u\|_2^2 - \left(2 - \frac{1}{c}\right)\eta_t\,[f(\theta_t) - f(u)] \tag{17}$$

$$\implies \|\theta_{t+1} - u\|_2^2 \leq \|\theta_t - u\|_2^2 - \left(2 - \frac{1}{c}\right)\frac{1}{\lambda_0 + \lambda_1 f(\theta_t)}[f(\theta_t) - f(u)]$$
$$\text{(Using Lemma 1)}$$

$$= \|\theta_t - u\|_2^2 - \left(2 - \frac{1}{c}\right)\frac{1}{\lambda_1}\frac{f(\theta_t) - f(u)}{f(\theta_t)}$$
$$\text{( using } \lambda_0 = 0 \text{ since } L_0, \omega = 0 \text{ and by defining } C := \left(2 - \tfrac{1}{c}\right)(1/\lambda_1))$$

By recursing from $t = 0$ to $T - 1$,

$$\|\theta_T - u\|_2^2 \leq \|\theta_0 - u\|_2^2 - C\sum_{t=0}^{T-1}\frac{f(\theta_t) - f(u)}{f(\theta_t)}$$

Assume $T$ is the first iteration s.t. $f(\theta_T) - f(u) \leq \epsilon$. Hence, $\frac{f(\theta_T)}{f(u)} \leq \alpha := 1 + \frac{\epsilon}{f(u)}$. Hence, for all $t < T$, $f(\theta_t) - f(u) > \epsilon$ and $\frac{f(\theta_t)}{f(u)} > \alpha$. Consequently, $\frac{f(\theta_t) - f(u)}{f(\theta_t)} \geq 1 - \frac{1}{\alpha}$. Combining the above relations,

$$\|\theta_T - u\|_2^2 \leq \|\theta_0 - u\|_2^2 - C\,T\left(1 - \frac{1}{\alpha}\right)$$

Since $f$ satisfies Assn. 1 to 3 and $f(u) \leq \epsilon$, then, using Lemma 4 with $M = f(\theta_0)$ and $B = \frac{e^q - 1}{q}$ where $q$ is such that $\|\theta_T - u\| \leq \frac{q}{L_1}$. Since $\|\theta_T - u\| \leq \|\theta_0 - u\|$, we can set $q = L_1\,\|\theta_0 - u\|$. Hence,

$$f(\theta_T) - f(u) \leq \frac{\epsilon}{2} + \underbrace{[\nu^2\,f(\theta_0) + B\,L_1\,f(\theta_0)]}_{:=L}\frac{\|\theta_T - u\|_2^2}{2}$$

$$\implies f(\theta_T) - f(u) \leq \frac{\epsilon}{2} + \frac{L}{2}\,\|\theta_T - u\|_2^2 \leq \frac{\epsilon}{2} + \frac{L}{2}\left[\|\theta_0 - u\|_2^2 - C\,T\left(1 - \frac{1}{\alpha}\right)\right] = \frac{\epsilon}{2} + \frac{L}{2}\left[\|\theta_0 - u\|_2^2 - C\,T\frac{\epsilon}{f(u) + \epsilon}\right] \tag{18}$$

Hence, in order to ensure that $f(\theta_T) - f(u) \leq \epsilon$, it is sufficient to guarantee that $\|\theta_T - u\|_2^2 \leq \frac{\epsilon}{L}$. In order to guarantee this, it is sufficient to set $T$ as follows.

$$T \geq \frac{\|\theta_0 - u\|_2^2 - \frac{\epsilon}{L}}{C}\left[1 + \frac{f(u)}{\epsilon}\right].$$
$$\text{(Using the definition of } \alpha)$$

Using the definition of $C$, we conclude that it is sufficient to set $T$ as:

$$T \geq \frac{c\lambda_1 \|\theta_0 - u\|_2^2}{(2c-1)} \left[1 + \frac{f(u)}{\epsilon}\right].$$

$\square$

**Corollary 2.** *For logistic regression on linearly separable data with margin $\gamma$, if, for all $i$, $\|x_i\| \leq 1$, for an initialization $\theta_0$, an $\epsilon \in (0, f(\theta_0))$, GD-LS with $\eta_{\max} = \infty$ requires $T \geq O\left(\frac{c}{(1-c)(2c-1)\gamma^2} \left[\ln\left(\frac{1}{\epsilon}\right)\right]^2\right)$ iterations to ensure that $f(\theta_T) \leq 2\epsilon$.*

*Proof.* Define $u^*$ to be the max-margin solution i.e. $\|u^*\| = 1$ and $\gamma$ to be the corresponding margin, i.e.

$$\gamma := \min_i y_i \langle x_i, u^* \rangle \tag{19}$$

For a scalar $\beta > 0$,

$$f(\beta u^*) = \frac{1}{n} \sum_{i=1}^n \ln(1 + \exp(-y_i \langle x_i, \beta u^* \rangle)) \leq \frac{1}{n} \sum_{i=1}^n \exp(-y_i \langle x_i, u^* \rangle) \leq \exp(-\beta\gamma) \tag{20}$$

For normalized data, s.t. $\|x_i\| \leq 1$, the logistic regression loss is convex, is uniform smooth with $L = \frac{\lambda_{\max}[X^T X]}{4n} \leq 1$.
We set $\beta = \frac{1}{\gamma} \ln\left(\frac{1}{\epsilon}\right)$ implies that $f(\beta u^*) \leq \epsilon$. For the $T$ defined in the theorem statement, consider two cases:

**Case (I):** $f(\theta_T) < f(\beta u^*) \leq \epsilon$. This gives the desired result immediately.

**Case (II):** $f(\theta_T) > f(\beta u^*)$. In this case, we can use the result in Eq. (18) in Thm. 2 with the comparator $u = \beta u^*$ where $f(u) \leq \epsilon$. Hence, GD with Armijo line-search with $c$, $\eta_{\max} = \infty$ and $\theta_0 = 0$ ensures that when $T$ is the first iteration s.t. $f(\theta_T) - f(u) \leq \epsilon \implies f(\theta_T) \leq 2\epsilon$, then, for $C := \frac{2c-1}{c\lambda_1}$, $L := [\nu^2 f(\theta_0) + B L_1 f(\theta_0)]$ where $B = \frac{e^q - 1}{q}$ and $q = L_1 \|\theta_0 - u\|$,

$$f(\theta_T) \leq f(\beta u^*) + \frac{L}{2}\left[\beta^2 - CT\left(\frac{\epsilon}{\epsilon + f(\beta u^*)}\right)\right]$$

$$\implies f(\theta_T) \leq \epsilon + \frac{L}{2}\left[\frac{1}{\gamma^2}\left[\ln\left(\frac{1}{\epsilon}\right)\right]^2 - \frac{CT}{2}\right] \qquad \text{(Since } f(\beta u^*) \leq \epsilon \text{ and } \beta = \frac{1}{\gamma}\ln\left(\frac{1}{\epsilon}\right))$$

Hence, in order to ensure that $f(\theta_T) \leq 2\epsilon$, it is sufficient to set $T$ as:

$$T \geq \frac{1}{C\gamma^2}\left[\ln\left(\frac{1}{\epsilon}\right)\right]^2$$

$$= \frac{c\lambda_1}{(2c-1)\gamma^2}\left[\ln\left(\frac{1}{\epsilon}\right)\right]^2$$

$$= \frac{3cL_1(\nu+1)}{(2c-1)(1-c)\gamma^2}\left[\ln\left(\frac{1}{\epsilon}\right)\right]^2 \qquad \text{(using the value of } \lambda_1)$$

$$= \frac{216c}{(2c-1)(1-c)\gamma^2}\left[\ln\left(\frac{1}{\epsilon}\right)\right]^2 \qquad \text{(using Prop. 1 for the value of } \nu \text{ and } L_1)$$

Combining the two cases, we conclude that $f(\theta_T) \leq 2\epsilon$. $\square$

**Corollary 5.** *Assume $f$ is convex and satisfies Assn. 5, then GD-LS with $\eta_{max} = \infty$ requires*

$$T = \mathcal{O}\left(\frac{R(L_c + L_g\sqrt{L_c} + L_g^2\sqrt{L_c} + L_g L_c)}{\epsilon} + \left(R(L_g(Lg + \sqrt{L_c})^2 + L_g(L_g + \sqrt{L_c}))\right)\frac{f^*}{\epsilon}\ln\left(\frac{f^*}{\epsilon}\right)\right)$$

*iterations to ensure that $f(\theta_T) - f^* \leq \epsilon$.*

*Proof.* Since $f$ is convex, we can follow a similar argument as in the proof of Cor. 1, to establish that $\|\nabla f(\theta_t)\|_2^2 \geq \frac{[f(\theta_t)-f^*]^2}{R}$ where $R = \|\theta_0 - \theta^*\|_2^2$. This allows us to apply the result from Thm. 1, to obtain a bound on $T$. Since the bound in **Case 2** is bigger than **Case 1**, we consider the bound in **Case 2**.

$$T \geq \frac{2\lambda_0 R}{\epsilon} + \max\{2 R\lambda_1, 1\} \left(\frac{f^*}{\epsilon} + 1\right) \ln\left(\frac{f(\theta_0) - f^*}{\epsilon}\right)$$

$$\implies T = \mathcal{O}\left(\frac{\lambda_0 R}{\epsilon} + \left(\frac{R\lambda_1}{\epsilon}\right) \ln\left(\frac{1}{\epsilon}\right)\right)$$

From Thm. 1, we have $\lambda_0 = \frac{3}{1-c}(L_0 + L_1\omega)$ and $\lambda_1 = \frac{3}{1-c}(L_1(\nu + 1))$. By using Prop. 3, we can substitute $L_0, L_1, \nu$, and $\omega$ with their respective expressions in terms of $L_c$ and $L_g$, yielding

$$T = \mathcal{O}\left(\frac{R(L_c + L_g\sqrt{L_c} + L_g^2\sqrt{L_c} + L_g L_c)}{\epsilon} + \left(R\left(L_g(Lg + \sqrt{L_c})^2 + L_g(L_g + \sqrt{L_c})\right)\right) \frac{f^*}{\epsilon} \ln\left(\frac{f^*}{\epsilon}\right)\right)$$

$\square$

## C.1. Proofs for the Polyak Step-size

For an arbitrary comparator $u$, we generalize the Polyak step-size (Polyak, 1987) at iteration $t \in [T]$ as:

$$\eta_t = \frac{f(\theta_t) - f(u)}{c \|\nabla f(\theta_t)\|_2^2}, \tag{21}$$

where, $c \in (0, 1)$ is a hyper-parameter. Note that when $u = \theta^* = \arg\min f(\theta)$, $\eta_t = \frac{f(\theta_t)-f^*}{c\|\nabla f(\theta_t)\|_2^2}$ recovers the standard Polyak step-size in Polyak (1987).

We analyze the convergence of GD with the step-size in Eq. (21) under Assn. 3 with $\omega = 0$ i.e. we will assume that $f$ is $L$ uniform smooth and that for all $\theta$, $\|\nabla f(\theta)\| \leq \nu f(\theta)$. From Prop. 1, 5 and 6, we know that this property is true from binary classification with the logistic loss, as well as for multi-class classification with the cross-entropy loss.

**Theorem 7.** *For an initialization $\theta_0$, $\epsilon \in (0, f(\theta_0))$ and comparator $u$ s.t. $f(u) \leq \epsilon$, if $f(\theta)$ is convex, satisfies Assn. 1 to 3 with $L_0 = 0$, $\omega = 0$, GD with the Polyak step-size in Eq. (21) and $c > \frac{1}{2}$, requires*

$$T \geq \frac{c\nu^2 \|\theta_0 - u\|_2^2}{(2c - 1)} \left[1 + \frac{f(u)}{\epsilon}\right]^2$$

*iterations to ensure that $f(\theta_T) - f(u) \leq \epsilon$.*

*Proof.* Following a proof similar to that for Thm. 2, for an arbitrary comparator $u$ s.t. $f(u) \leq f(\theta_T)$,

$$\|\theta_{t+1} - u\|_2^2 = \|\theta_t - u\|_2^2 - 2\eta_t \langle\nabla f(\theta_t), \theta_t - u\rangle + \eta_t^2 \|\nabla f(\theta_t)\|_2^2 \leq \|\theta_t - u\|_2^2 - 2\eta_t [f(\theta_t) - f(u)] + \eta_t^2 \|\nabla f(\theta_t)\|_2^2$$

$$\text{(Convexity)}$$

$$\leq \|\theta_t - u\|_2^2 - 2\eta_t [f(\theta_t) - f(u)] + \frac{\eta_t}{c} [f(\theta_t) - f(u)]$$

$$\text{(Using the Polyak step-size in Eq. (21) with } c > \tfrac{1}{2} \text{ to simplify the third term)}$$

$$= \|\theta_t - u\|_2^2 - \left(2 - \frac{1}{c}\right) \frac{[f(\theta_t) - f(u)]^2}{c \|\nabla f(\theta_t)\|_2^2}$$

$$\text{(Using the Polyak step-size in Eq. (21))}$$

$$\leq \|\theta_t - u\|_2^2 - \left(2 - \frac{1}{c}\right) \frac{[f(\theta_t) - f(u)]^2}{c\nu^2 [f(\theta_t)]^2}$$

$$\text{(Using Assn. 3)}$$

$$\implies \|\theta_{t+1} - u\|_2^2 \leq \|\theta_t - u\|_2^2 - C \left(\frac{f(\theta_t) - f(u)}{f(\theta_t)}\right)^2$$

$$\text{(Using Lemma 1 and defining } C := \tfrac{(2c-1)}{c\nu^2})$$

By recursing from $t = 0$ to $T - 1$,

$$\|\theta_T - u\|_2^2 \le \|\theta_0 - u\|_2^2 - C \sum_{t=0}^{T-1} \left( \frac{f(\theta_t) - f(u)}{f(\theta_t)} \right)^2$$

Assume $T$ is the first iteration s.t. $f(\theta_T) - f(u) \le \epsilon$. Hence, $\frac{f(\theta_T)}{f(u)} \le \alpha := 1 + \frac{\epsilon}{f(u)}$. Hence, for all $t < T$, $f(\theta_t) - f(u) > \epsilon$ and $\frac{f(\theta_t)}{f(u)} > \alpha$. Consequently, $\frac{f(\theta_t) - f(u)}{f(\theta_t)} \ge 1 - \frac{1}{\alpha}$. Combining the above relations,

$$\|\theta_T - u\|_2^2 \le \|\theta_0 - u\|_2^2 - C\,T \left( 1 - \frac{1}{\alpha} \right)^2$$

Proceeding in the same manner as the proof of Thm. 2, where $B = \frac{e^q - 1}{q}$ and $q = L_1 \|\theta_0 - u\|$, we obtain that,

$$f(\theta_T) - f(u) \le \frac{\epsilon}{2} + \underbrace{\left[ \nu^2 f(\theta_0) + B\,L_1\,f(\theta_0) \right]}_{:=L} \frac{\|\theta_T - u\|_2^2}{2}$$

$$\implies f(\theta_T) - f(u) \le \frac{\epsilon}{2} + \frac{L}{2} \|\theta_T - u\|_2^2 \le \frac{\epsilon}{2} + \frac{L}{2} \left[ \|\theta_0 - u\|_2^2 - C\,T \left( 1 - \frac{1}{\alpha} \right) \right] = \frac{\epsilon}{2} + \frac{L}{2} \left[ \|\theta_0 - u\|_2^2 - C\,T \frac{\epsilon}{f(u) + \epsilon} \right] \tag{22}$$

Hence, in order to ensure that $f(\theta_T) - f(u) \le \epsilon$, it is sufficient to guarantee that $\|\theta_T - u\|_2^2 \le \frac{\epsilon}{L}$. In order to guarantee this, it is sufficient to set $T$ as follows.

$$T \ge \frac{c\,\nu^2 \|\theta_0 - u\|_2^2}{(2c - 1)} \left[ 1 + \frac{f(u)}{\epsilon} \right]^2$$

$\square$

**Theorem 8.** *For logistic regression on linearly separable data with margin $\gamma$, if, for all $i$, $\|x_i\| \le 1$, for an initialization $\theta_0 = 0$, a fixed $\epsilon \in (0, f(\theta_0))$, GD with the Polyak step-size $\eta_t = \min \left\{ \frac{f(\theta_t)}{c\,\|\nabla f(\theta_t)\|_2^2}, \frac{1}{2\epsilon} \right\}$ for some $c > 2$ requires*

$$T \ge \frac{\beta^2}{C} = \frac{64\,c^2}{(c-1)\,\gamma^2} \left[ \ln \left( \frac{64\,c}{\epsilon} \right) \right]^2$$

*to ensure that $f(\theta_T) \le 2\,\epsilon$.*

*Proof.* The logistic loss on linearly separable data is convex, satisfies Assn. 1 to 3 with $L_0 = 0$, $\omega = 0$, $\nu = 8$, and $f^* = 0$. By Assn. 3, $\|\nabla f(\theta)\| \le \nu\,f(\theta)$ and we can bound the Polyak step-size as:

$$\eta_t \in \left[ \min \left\{ \frac{1}{c\,\nu^2\,f(\theta_t)}, \frac{1}{2\epsilon} \right\}, \frac{1}{2\epsilon} \right]. \tag{23}$$

Using the GD update: $\theta_{t+1} = \theta_t - \eta_t \nabla f(\theta_t)$, consider a comparator $u$ s.t. $f(u) \le \frac{\epsilon}{\max\{c\,\nu^2, 2\}}$ and $f(u) \le f(\theta_t)$ for all

$t \in [T]$. Assuming that $T$ is the first iteration such that $f(\theta_T) - f(u) \leq \epsilon$, we have that,

$$
\begin{aligned}
\|\theta_{t+1} - u\|_2^2 &= \|\theta_t - u\|_2^2 - 2\,\eta_t \, \langle \nabla f(\theta_t), \theta_t - u \rangle + {\eta_t}^2 \, \|\nabla f(\theta_t)\|_2^2 \\
&\leq \|\theta_t - u\|_2^2 - 2\,\eta_t \, [f(\theta_t) - f(u)] + {\eta_t}^2 \, \|\nabla f(\theta_t)\|_2^2 && \text{(Convexity)} \\
&\leq \|\theta_t - u\|_2^2 - 2\,\eta_t \, [f(\theta_t) - f(u)] + \frac{\eta_t}{c} \, [f(\theta_t)] && \text{(Using the Polyak step-size with } c > 2\text{)} \\
&= \|\theta_t - u\|_2^2 - \left(2 - \frac{1}{c}\right) \eta_t \, f(\theta_t) + 2\,\eta_t \, f(u) \\
&\leq \|\theta_t - u\|_2^2 - \left(2 - \frac{1}{c}\right) \min\left\{\frac{1}{c\,\nu^2}, \frac{f(\theta_t)}{2\epsilon}\right\} + 2\,\eta_t \, f(u) && \text{(Using Eq. (23))} \\
&\leq \|\theta_t - u\|_2^2 - \left(2 - \frac{1}{c}\right) \frac{1}{\max\{c\,\nu^2, 2\}} + 2\,\eta_t \, f(u) && \text{(Since } f(\theta_t) \geq \epsilon \text{ for all } t < T\text{)} \\
&\leq \|\theta_t - u\|_2^2 - \left(2 - \frac{1}{c}\right) \frac{1}{\max\{c\,\nu^2, 2\}} + \frac{f(u)}{\epsilon} && \text{(Since } \eta_t \leq \frac{1}{2\epsilon} \text{ from Eq. (23))} \\
&\leq \|\theta_t - u\|_2^2 - \left(2 - \frac{1}{c}\right) \frac{1}{\max\{c\,\nu^2, 2\}} + \frac{1}{\max\{c\,\nu^2, 2\}} && \text{(Since } f(u) \leq \frac{\epsilon}{\max\{c\,\nu^2,2\}}\text{)} \\
&= \|\theta_t - u\|_2^2 - \underbrace{\left(1 - \frac{1}{c}\right) \frac{1}{\max\{c\,\nu^2, 2\}}}_{:=C}
\end{aligned}
$$

Summing up from $t = 0$ to $t = T - 1$,

$$
\|\theta_T - u\|_2^2 \leq \|\theta_0 - u\|_2^2 - CT = \|u\|_2^2 - C\,Tu \qquad \text{(Since } \theta_0 = 0\text{)}
$$

Since $f$ is 1 uniformly smooth, using Lemma 4 with $M = f(\theta_0)$, we get

$$
f(\theta_T) - f(u) \leq \frac{\epsilon}{2} + \underbrace{[\nu^2 \, f(\theta_0) + 1]}_{:=L} \frac{\|\theta_T - u\|_2^2}{2} \leq \frac{\epsilon}{2} + L \left[\|u\|_2^2 - C\,T\right]
$$

To ensure that $f(\theta_T) - f(u) \leq \epsilon$, it is sufficient to set

$$
T \geq \frac{\|u\|_2^2}{C}. \tag{24}
$$

In order to bound $\|u\|$, we define $u^*$ to be the max-margin solution i.e. $\|u^*\| = 1$ and $\gamma$ to be the corresponding margin, i.e.

$$
\gamma := \min_i y_i \langle x_i, u^* \rangle \tag{25}
$$

Consider $u = \beta \, u^*$, for a scalar $\beta = \frac{1}{\gamma} \ln\left(\frac{\max\{c\nu^2, 2\}}{\epsilon}\right)$,

$$
f(u) = \frac{1}{n} \sum_{i=1}^{n} \ln(1 + \exp(-y_i \langle x_i, \beta u^* \rangle)) \leq \frac{1}{n} \sum_{i=1}^{n} \exp(-y_i \langle x_i, u^* \rangle) \leq \exp(-\beta\gamma) = \frac{\epsilon}{\max\{c\nu^2, 2\}} \tag{26}
$$

This satisfies the requirement on $f(u)$. For logistic regression, we $\nu = 8$ and since $c > 2$, therefore $\max\{c\nu^2, 2\} = c\nu^2$. Using this to bound $T$, we get that,

$$
T \geq \frac{\beta^2}{C} = \frac{c^2\,\nu^2}{(c-1)\,\gamma^2} \left[\ln\left(\frac{c\,\nu^2}{\epsilon}\right)\right]^2
$$

Finally, we conclude that after $T = \frac{\beta^2}{C} = \frac{c^2\,\nu^2}{(c-1)\,\gamma^2} \left[\ln\left(\frac{c\,\nu^2}{\epsilon}\right)\right]^2$ iterations we have $f(\theta_T) - f(u) \leq \epsilon$ and since $f(u) \leq \epsilon$ then $f(\theta_T) \leq 2\,\epsilon$ $\qquad\qquad \square$

### C.2. Helper Lemmas

**Lemma 4.** *For $\epsilon \in (0, M)$ and a comparator $u$ s.t. $f(u) \leq \epsilon$, if $f$ satisfies Assn. 1 to 3 with $L_0 = 0$ and $\omega = 0$, then, for all $\theta$ s.t. $\|\theta - u\| \leq \frac{q}{L_1}$,*

$$f(\theta) - f(u) \leq \frac{\epsilon}{2} + \left[\nu^2 M + B L_1 M\right] \frac{\|\theta - u\|_2^2}{2},$$

*where $B := \frac{e^q - 1}{q}$.*

*Furthermore, if $f$ is also $L$ uniform smooth, then, for all $\theta$,*

$$f(\theta) - f(u) \leq \frac{\epsilon}{2} + \left[\nu^2 M + L\right] \frac{\|\theta - u\|_2^2}{2},$$

*Proof.* Since $f$ satisfies Assn. 2 with $L_0 = 0$, using Eq. (1), we have that for all $\theta$ s.t. $\|\theta - u\| \leq \frac{q}{L_1}$ and $B := \frac{e^q - 1}{q}$,

$$
\begin{aligned}
f(\theta) - f(u) &\leq \langle \nabla f(u), \theta - u \rangle + \frac{B L_1 f(u)}{2} \|\theta - u\|_2^2 \\
&\leq \frac{\|\nabla f(u)\|_2^2}{2 M \nu^2} + \frac{\nu^2 M \|\theta - u\|_2^2}{2} + \frac{B L_1 f(u)}{2} \|\theta - u\|_2^2 && \text{(Using Young's inequality)} \\
&\leq \frac{[f(u)]^2}{2 M} + \left[\nu^2 M + B L_1 f(u)\right] \frac{\|\theta - u\|_2^2}{2} && \text{(Since $f$ satisfies Assn. 3 with $\omega = 0$)} \\
&\leq \frac{\epsilon^2}{2 M} + \left[\nu^2 M + B L_1 \epsilon\right] \frac{\|\theta - u\|_2^2}{2} && \text{(Since $f(u) \leq \epsilon$)} \\
\implies f(\theta) - f(u) &\leq \frac{\epsilon}{2} + \left[\nu^2 M + B L_1 M\right] \frac{\|\theta - u\|_2^2}{2} && \text{(Since $\frac{\epsilon^2}{2 M} \leq \frac{\epsilon}{2}$ for $\epsilon \leq M$)}
\end{aligned}
$$

If $f$ is $L$-uniform smooth, we can use the standard descent lemma to get, that for all $\theta$,

$$f(\theta) - f(u) \leq \langle \nabla f(u), \theta - u \rangle + \frac{L}{2} \|\theta - u\|_2^2$$

Using this inequality and following the same proof gives the result. $\square$

# D. Proofs for Sec. 5

## D.1. Proofs for Sec. 5.1

**Corollary 3.** *For an $\epsilon > 0$, assuming $f(\theta)$ satisfies Assn. 1 to 3 with $L_0 = 0$, $\omega = 0$ and Assn. 4 with $\zeta = 1$, if $\mu := \min_{t \in [T]} \mu(\theta_t)$, then, GD-LS with $\eta_{\max} = \infty$, requires $T \geq \max\left\{1, \frac{2\lambda_1}{\mu^2}\right\} \left(\frac{f^*}{\epsilon} + 1\right) \ln\left(\frac{f(\theta_0) - f^*}{\epsilon}\right)$ iterations to ensure $f(\theta_T) \leq \epsilon$.*

*Proof.* Using Assn. 4, we know that,

$$\|\nabla f(\theta)\|_2^2 \geq [\mu(\theta)]^2 [f(\theta) - f^*]^2 \geq \mu^2 [f(\theta) - f^*]^2$$

Using Thm. 1 with $R = \frac{1}{\mu^2}$ and $L_0 = 0$ completes the proof. $\qquad\square$

**Corollary 4.** *For an MAB problem with $K$ arms, rewards bounded in $[0, 1]$, GD-LS with a uniform initialization i.e. $\forall a$, $\pi_{\theta_0}(a) = 1/K$, $c = \frac{1}{2}$, $\eta_{\max} = \infty$ requires $T = O\left(K^2 \ln\left(1/\epsilon\right)\right)$ iterations to guarantee $\langle \pi_{\theta_T}, r \rangle \geq r(a^*) - \epsilon$.*

*Proof.* From Prop. 4, we know that the MAB problem satisfies both Assn. 2 and Assn. 3 with the parameters $L_0 = 0$, $L_1 = 72$, $\nu = 24$, and $\omega = 0$. It also satisfies Assn. 4 with $\zeta = 1$. Since GD-LS guarantees monotonic decrease of the objective, Mei et al. (2020, Lemma 5 and Proposition 2), implies that for the uniform initialization where $\pi_0(a) = \frac{1}{K}$ for all $a$, $\pi_{\theta_t}(a^*) \geq \frac{1}{K}$, $\forall t \geq 0$. Therefore, we have $\mu = \min_t \pi_{\theta_t}(a^*) = \frac{1}{K}$. With these parameters, we can instantiate the result of Cor. 3, and get:

$$T > \max\left\{1, 2^5 \, 3^3 \, 5^2 \, K^2\right\} \left(\frac{f^*}{\epsilon} + 1\right) \ln\left(\frac{f(\theta_0) - f^*}{\epsilon}\right) \qquad \text{(since } \lambda_1 = 3\frac{L_1(\nu+1)}{1-c} \text{ and } c = \frac{1}{2})$$

$$\implies T = O\left(K^2 \ln\left(\frac{1}{\epsilon}\right)\right) \qquad \text{(since } f^* = 0 \text{ and } f(\theta_0) - f^* \leq 1)$$

$\qquad\square$

**Corollary 6.** *For the tabular MDP problem defined in Prop. 7, GD-LS, with $c = \frac{1}{2}$, and $\eta_{max} = \infty$ requires*

$$T \geq \frac{2^7 3}{\mu} \left(\left[3 + \frac{4 \cdot \left(\min_s \frac{1}{\rho(s)} - (1-\gamma)\right)}{1-\gamma}\right]^3 S^{\frac{3}{2}}\right) \ln\left(\frac{1}{(1-\gamma)\epsilon}\right)$$

*iterations to guarantee $f(\theta_T) = V^{\pi^*}(\rho) - V^{\pi_{\theta_T}}(\rho) \leq \epsilon$, where $\mu = \inf_{t \in [T]} \mu(\theta_t)$ and $\mu(\theta) = \frac{\min_s \pi_\theta(a^*(s)|s)}{\sqrt{S} \min_{s \in S} \rho(s)}$, $a^*(s)$ is the action corresponding to the optimal policy $\pi^*$ in state $s$, $\gamma$ is the discount factor, $\rho$ is the initial state distribution, and $S$ is the size of state space.*

*Proof.* As proved in Prop. 7, the function $f(\theta)$ satisfies Assn. 2 with $L_0 = 0$ and $L_1 = 8\left[3 + \frac{4 \cdot \left(\min_s \frac{1}{\rho(s)} - (1-\gamma)\right)}{1-\gamma}\right]^2 S$,

Assn. 3 with $\nu = 8\left[3 + \frac{4 \cdot \left(\min_s \frac{1}{\rho(s)} - (1-\gamma)\right)}{1-\gamma}\right] \sqrt{S}$ and $\omega = 0$ and Assn. 4 with . Since GD-LS guarantees monotonic decrease in the objective, Mei et al. (2020, Lemma 9), implies that $\mu > 0$. With these parameters and noting that $f^* = 0$,

we can instantiate the result of Cor. 3 and get:

$$T \geq \max\left\{1, \frac{48}{\mu}\left(\left[3 + \frac{4 \cdot \left(\min_s \frac{1}{\rho(s)} - (1-\gamma)\right)}{1-\gamma}\right]^2 S\left(8\left[3 + \frac{4 \cdot \left(\min_s \frac{1}{\rho(s)} - (1-\gamma)\right)}{1-\gamma}\right]\sqrt{S} + 1\right)\right)\right\} \ln\left(\frac{f(\theta_0)}{\epsilon}\right)$$

$$\left(\text{since } \lambda_1 = 3\frac{L_1(\nu+1)}{1-c} \text{ and } c = \tfrac{1}{2}\right)$$

$$\implies T \geq \frac{48}{\mu}\left(\left[3 + \frac{4 \cdot \left(\min_s \frac{1}{\rho(s)} - (1-\gamma)\right)}{1-\gamma}\right]^2 S\left(8\left[3 + \frac{4 \cdot \left(\min_s \frac{1}{\rho(s)} - (1-\gamma)\right)}{1-\gamma}\right]\sqrt{S} + 1\right)\right) \ln\left(\frac{1}{(1-\gamma)\epsilon}\right)$$

$$\left(\text{since } V^\pi(\rho) \leq \tfrac{1}{1-\gamma} \text{ for all policies } \pi\right)$$

$$\implies T \geq \frac{2^7 3}{\mu}\left(\left[3 + \frac{4 \cdot \left(\min_s \frac{1}{\rho(s)} - (1-\gamma)\right)}{1-\gamma}\right]^3 S^{\frac{3}{2}}\right) \ln\left(\frac{1}{(1-\gamma)\epsilon}\right)$$

$$\square$$

### D.2. Proofs for Sec. 5.2

**Theorem 3.** *For a fixed $\epsilon \in \left(0, \frac{\lambda_0}{\lambda_1}\right)$, if $f$ satisfies Assn. 1 to 3 and Assn. 4 with $\zeta = 2$, $f^* = 0$ and if $\mu := \min_{t\in[T]} \mu(\theta_t) > 0$, then, GD-LS with $\eta_{\max} = \infty$, requires $T \geq \frac{2}{\mu}\left[\lambda_1 f(\theta_0) + \lambda_0 \ln\left(\frac{\lambda_0}{\lambda_1 \epsilon}\right)\right]$ iterations to ensure that $f(\theta_T) \leq \epsilon$.*

*Proof.* Using the Armijo line-search condition in Eq. (5), and combining it with the lower-bound in Lemma 1,

$$f(\theta_{t+1}) \leq f(\theta_t) - \frac{1}{\lambda_0 + \lambda_1 f(\theta_t)} \|\nabla f(\theta_t)\|_2^2 \tag{27}$$

**Phase 1**: Let us define $\tau := \max\{t \text{ s.t } \lambda_0 \leq \lambda_1 f(\theta_t)\}$. Hence, for all $t \leq \tau$, $f(\theta_t) \geq \frac{\lambda_0}{\lambda_1}$, and hence,

$$f(\theta_{t+1}) \leq f(\theta_t) - \frac{1}{2\lambda_1 f(\theta_t)} \|\nabla f(\theta_t)\|_2^2 \ .$$

From Assn. 4 with $\zeta = 2$, we know that $\|\nabla f(\theta)\|_2^2 \geq \mu(\theta)\left[f(\theta) - f^*\right]$ and that $f^* = 0$. Combining these relations, we have that for all $t \leq \tau$,

$$f(\theta_{t+1}) \leq f(\theta_t) - \frac{f(\theta_t)}{2\lambda_1 R f(\theta_t)} = f(\theta_t) - \underbrace{\frac{\mu}{2\lambda_1}}_{:=\alpha} \qquad\qquad (\text{Since } \mu = \min_{t\in[T]} \mu(\theta_t))$$

Recursing from $t = 0$ to $t = \tau$,

$$f(\theta_\tau) \leq f(\theta_0) - \tau\alpha$$

Since $f(\theta_\tau) \geq \frac{\lambda_0}{\lambda_1}$, we get that,

$$\tau \leq \frac{1}{\alpha}\left[f(\theta_0) - \frac{\lambda_0}{\lambda_1}\right] = \frac{2\lambda_1}{\mu}\left[f(\theta_0) - \frac{\lambda_0}{\lambda_1}\right]$$

**Phase 2**: For $t \geq \tau$, $f(\theta_t) \leq \frac{\lambda_0}{\lambda_1}$. Combining this with Eq. (27) and using that $\|\nabla f(\theta_t)\|_2^2 \geq \mu f(\theta_t)$,

$$f(\theta_{t+1}) \leq f(\theta_t) - \frac{\mu f(\theta_t)}{2\lambda_0} = f(\theta_t)\left(1 - \frac{\mu}{2\lambda_0}\right)$$

Recursing from $t = \tau$ to $t = T$ and using that $1 - x \leq \exp(-x)$,

$$f(\theta_T) \leq \exp\left(-\frac{\mu(T-\tau)}{2\lambda_0}\right) f(\theta_\tau)$$

Hence, in order for $f(\theta_T) \leq \epsilon$, we require

$$T \geq \tau + 2\,\frac{\lambda_0}{\mu}\,\ln\left(\frac{f(\theta_\tau)}{\epsilon}\right)$$

Since $f(\theta_\tau) \leq f(\theta_0) - \tau\,\alpha$, is sufficient to set $T$ as:

$$T \geq \underbrace{\tau + 2\,\frac{\lambda_0}{\mu}\,\ln\left(\frac{f(\theta_0) - \tau\,\alpha}{\epsilon}\right)}_{:=h(\tau)}$$

Hence, it is sufficient to set $T$ as:

$$T \geq \max_\tau h(\tau) \quad \text{s.t} \quad \tau \leq 2\frac{\lambda_1}{\mu}\left[f(\theta_0) - \frac{\lambda_0}{\lambda_1}\right].$$

Calculating the first and second derivatives of $h(\tau)$,

$$h'(\tau) = 1 - \frac{2\lambda_0\,\alpha}{\mu\,(f(\theta_0) - \tau\,\alpha)} \quad ; \quad h''(\tau) = -\frac{2\lambda_0\,\alpha^2}{\mu\,(f(\theta_0) - \tau\alpha)^2}$$

Hence, $h(\tau)$ is maximized when at $\tau^* := 2\lambda_1\mu\left[f(\theta_0) - \frac{\lambda_0}{\lambda_1}\right]$. Calculating $h(\tau^*)$, we conclude that it is sufficient to set $T$ as:

$$T \geq \frac{2}{\mu}\left[\lambda_1 f(\theta_0) + \lambda_0\left(\ln\left(\frac{\lambda_0}{\lambda_1\,\epsilon}\right)\right)\right]$$

$\square$

## E. Proofs for Sec. 6

**Lemma 5.** *For a finite sum* $f(\theta) = \frac{1}{n}\sum_{i=1}^{n}$, *if* $\min f_i(\theta) = 0$ *for all* $i$ *and an arbitrary comparator* $u$, *then iteration* $t$ *of* `SGD-SLS` *satisfies the following bound:*

$$\|\theta_{t+1} - u\|_2^2 \leq \|\theta_t - u\|_2^2 - \left(2 - \frac{1}{c}\right)\eta_t\left[f_t(\theta_t)\right] + 2\eta_t[f_t(u)].$$

*Proof.* Using the SGD update: $\theta_{t+1} = \theta_t - \eta_t\,\nabla f_t(\theta_t)$, for a comparator $u$,

$$
\begin{aligned}
\|\theta_{t+1} - u\|_2^2 &= \|\theta_t - u\|_2^2 - 2\,\eta_t\,\langle\nabla f_t(\theta_t), \theta_t - u\rangle + {\eta_t}^2\,\|\nabla f_t(\theta_t)\|_2^2 \\
&\leq \|\theta_t - u\|_2^2 - 2\,\eta_t\,[f_t(\theta_t) - f_t(u)] + {\eta_t}^2\,\|\nabla f_t(\theta_t)\|_2^2 && \text{(Convexity)} \\
&\leq \|\theta_t - u\|_2^2 - 2\,\eta_t\,[f_t(\theta_t) - f_t(u)] + \frac{\eta_t}{c}\,[f_t(\theta_t) - f_t(\theta_{t+1})] \\
&&& \hspace{-4cm} \text{(Using the stochastic Armijo line-search with } c > \tfrac{1}{2}) \\
&\leq \|\theta_t - u\|_2^2 - 2\,\eta_t\,[f_t(\theta_t) - f_t(u)] + \frac{\eta_t}{c}\,[f_t(\theta_t) - {f_t}^*] && \text{(where } {f_t}^* := \min f_t(\theta)) \\
&= \|\theta_t - u\|_2^2 - 2\,\eta_t\,[f_t(\theta_t) - {f_t}^*] - 2\eta_t[{f_t}^* - f_t(u)] + \frac{\eta_t}{c}\,[f_t(\theta_t) - {f_t}^*] && \text{(Add/subtract } {f_t}^*) \\
&= \|\theta_t - u\|_2^2 - \left(2 - \frac{1}{c}\right)\eta_t\,[f_t(\theta_t) - {f_t}^*] + 2\eta_t[f_t(u) - {f_t}^*] \\
\|\theta_{t+1} - u\|_2^2 &\leq \|\theta_t - u\|_2^2 - \left(2 - \frac{1}{c}\right)\eta_t\,[f_t(\theta_t)] + 2\eta_t[f_t(u)] && \text{(Since } {f_t}^* = 0)
\end{aligned}
$$

$\square$

**Theorem 4.** *For logistic regression on linearly separable data with margin* $\gamma$, *if, for all* $i$, $\|x_i\| \leq 1$, *for a fixed* $\epsilon \in (0, 1)$, `SGD-SLS` *with* $\eta_{\max} = \frac{1}{\epsilon}$ *and* $c = \frac{2}{3}$ *requires* $T = O\left(\frac{1}{\epsilon\gamma^2}\left[\ln\left(\frac{1}{\epsilon^2}\right)\right]^2\right)$ *iterations to ensure that* $\mathbb{E}[\inf_{t\in[T]} f(\theta_t)] \leq \frac{5\epsilon}{2}$.

*Proof.* For the $\epsilon$ and $T$ defined in the theorem statement, define the event $G = \{\inf_{t\in[T]}[f(\theta_t)] \leq \epsilon\}$. Using the law of total expectation,

$$\mathbb{E}\left[\inf_{t\in[T]} f(\theta_t)\right] = \mathbb{E}\left[\inf_{t\in[T]} f(\theta_t)|G\right]\Pr[G] + \mathbb{E}\left[\inf_{t\in[T]} f(\theta_t)|G^c\right]\Pr[G^c] \tag{28}$$

$$\leq \epsilon + \mathbb{E}\left[\inf_{t\in[T]} f(\theta_t)|G^c\right] \qquad\qquad \text{(By definition of } G)$$

Hence, we need to bound $\mathbb{E}\left[\inf_{t\in[T]} f(\theta_t)\right]$ conditioned on the event $G^c = \{\inf_{t\in[T]} f(\theta_t) \geq \epsilon\}$.

**Conditioned on** $G^c$**:** For normalized data, s.t. $\|x_i\| \leq 1$, the logistic regression loss is convex, is uniform smooth with $L = \frac{\lambda_{\max}[X^T X]}{4n} \leq 1$. Using the standard guarantee for Armijo line-search, we know that $\eta_{\max} \geq \min\left\{\eta_{\max}, \frac{2\,(1-c)}{L}\right\}$. Since $f_t(\theta) \geq 0$, combining this with Lemma 5,

$$
\begin{aligned}
\|\theta_{t+1} - u\|_2^2 &\leq \|\theta_t - u\|_2^2 - \left(2 - \frac{1}{c}\right)\min\left\{\eta_{\max}, \frac{2\,(1-c)}{L}\right\}f_t(\theta_t) + 2\eta_{\max}[f_t(u)] \\
&\leq \|\theta_t - u\|_2^2 - \frac{1}{2}\min\left\{\eta_{\max}, \frac{2}{3}\right\}f_t(\theta_t) + 2\eta_{\max}[f_t(u)] && \text{(Since } L \leq 1 \text{ and setting } c = \tfrac{2}{3}) \\
&= \|\theta_t - u\|_2^2 - \frac{1}{3}f_t(\theta_t) + \frac{2f_t(u)}{\epsilon} && \text{(Setting } \eta_{\max} = \tfrac{1}{\epsilon} \text{ and since } \epsilon \leq 1)
\end{aligned}
$$

Taking an expectation over the randomness in iteration $t$, conditioned on the past iterates,

$$\implies \mathbb{E}_t[\|\theta_{t+1} - u\|_2^2] \leq \|\theta_t - u\|_2^2 - \frac{f(\theta_t)}{3} + \frac{2f(u)}{\epsilon} \qquad \text{(Since } u \text{ is independent of the randomness at iteration } t)$$

Consider $u = \beta u^*$ where $\beta = \frac{1}{\gamma} \ln\left(\frac{12}{\epsilon^2}\right)$ implies that $f(u) \leq \frac{\epsilon^2}{12}$. Since $\epsilon \leq 1$, $f(u) \leq \epsilon$. Using this relation with the above inequality, we get,

$$\mathbb{E}_t[\|\theta_{t+1} - u\|_2^2] \leq \|\theta_t - u\|_2^2 - \frac{f(\theta_t)}{3} + \frac{\epsilon}{6}$$

From the conditioning on $G^c$, we know that $f(\theta_t) \geq \epsilon$. Hence,

$$\implies \mathbb{E}_t[\|\theta_{t+1} - u\|_2^2] \leq \|\theta_t - u\|_2^2 - \frac{\epsilon}{3} + \frac{\epsilon}{6} = \|\theta_t - u\|_2^2 - \frac{\epsilon}{6}$$

Taking expectation w.r.t. the randomness from iterations $t = 0$ to $T - 1$,

$$\mathbb{E}[\|\theta_{t+1} - u\|_2^2] \leq \mathbb{E}[\|\theta_t - u\|_2^2] - \frac{\epsilon}{6}$$

$$\implies \mathbb{E}[\|\theta_T - u\|_2^2] \leq \|\theta_0 - u\|_2^2 - \frac{\epsilon T}{6} \qquad \text{(Summing from } t = 0 \text{ to } T - 1)$$

$$\leq \beta^2 - \underbrace{\frac{\epsilon}{6}}_{:=C} T \qquad \text{(Since } \theta_0 = 0 \text{ and } \|u\| = \beta)$$

Since $f$ satisfies Assn. 1 to 3, $f$ is 1 uniform smooth and $f(u) \leq \epsilon$, then, using Lemma 4 with $M = \frac{1}{8}$,

$$\mathbb{E}[f(\theta_T) - f(u)] \leq \frac{\epsilon}{2} + \underbrace{[\nu^2 + 1]}_{:=L} \mathbb{E}\frac{\|\theta_T - u\|_2^2}{2} \leq \frac{\epsilon}{2} + \frac{L}{2}\left[\beta^2 - C T\right]$$

$$\implies \mathbb{E}[f(\theta_T)] \leq \frac{3\epsilon}{2} + \frac{L}{2}\left[\beta^2 - C T\right] \qquad \text{(Since } f(u) \leq \epsilon)$$

In order to ensure that $\mathbb{E}[f(\theta_T)] \leq \frac{3\epsilon}{2}$, it is sufficient to set $T$ as:

$$T \geq \frac{\beta^2}{C} = \frac{6}{\epsilon \gamma^2}\left[\ln\left(\frac{12}{\epsilon^2}\right)\right]^2$$

Since the above calculations were conditioned on $G^c$, we can conclude that after $T = \frac{6}{\epsilon \gamma^2}\left[\ln\left(\frac{12}{\epsilon^2}\right)\right]^2$ iterations, $\mathbb{E}[\inf_{t \in [T]} f(\theta_T)|G^c] \leq \frac{3\epsilon}{2}$. Hence, from Eq. (29)

$$\mathbb{E}\left[\inf_{t \in [T]} f(\theta_t)\right] \leq \frac{5\epsilon}{2}$$

$\square$

**Theorem 5.** *For logistic regression on linearly separable data with margin $\gamma$, if, for all $i$, $\|x_i\| \leq 1$, for a fixed $\epsilon \in \left(0, \frac{C'}{8}\right)$ where $C' = O(1)$, SGD-SLS with $\eta_{\max} = \frac{1}{\epsilon}$ and $c = \frac{2}{3}$ requires $T = O\left(\frac{n}{\gamma^2}\left[\ln\left(\frac{n}{\epsilon^2}\right)\right]^2\right)$ iterations to ensure that $\mathbb{E}[\inf_{t \in [T]} f(\theta_t)] \leq \frac{5\epsilon}{2}$.*

*Proof.* For the $\epsilon$ and $T$ defined in the theorem statement, define the event $G = \{\inf_{t \in [T]}[f(\theta_t)] \leq \epsilon\}$. Using the law of total expectation,

$$\mathbb{E}\left[\inf_{t \in [T]} f(\theta_t)\right] = \mathbb{E}\left[\inf_{t \in [T]} f(\theta_t)|G\right] \Pr[G] + \mathbb{E}\left[\inf_{t \in [T]} f(\theta_t)|G^c\right] \Pr[G^c] \qquad (29)$$

$$\leq \epsilon + \mathbb{E}\left[\inf_{t \in [T]} f(\theta_t)|G^c\right] \qquad \text{(By definition of } G)$$

Hence, we need to bound $\mathbb{E}\left[\inf_{t \in [T]} f(\theta_t)\right]$ conditioned on the event $G^c = \{\inf_{t \in [T]} f(\theta_t) \geq \epsilon\}$.

**Conditioned on $G^c$:** Note that from Lemma 1, we know that $\eta_t \geq \min\left\{\eta_{\max}, \frac{C'}{f_t(\theta_t)}\right\}$ where $C' := \frac{1}{\lambda_1}$. Since $f_t(\theta) \geq 0$, combining this with Lemma 5, we get that,

$$\|\theta_{t+1} - u\|_2^2 \leq \|\theta_t - u\|_2^2 - \left(2 - \frac{1}{c}\right) \min\{\eta_{\max} f_t(\theta_t), C'\} + 2\eta_{\max}[f_t(u)]$$

$$= \|\theta_t - u\|_2^2 - \underbrace{\frac{\min\{\eta_{\max} f_t(\theta_t), C'\}}{2}}_{:=(*)} + 2\eta_{\max}[f_t(u)] \qquad \text{(Setting } c = \frac{2}{3}\text{)}$$

In order to simplify (*), we consider two cases that depend on whether or not $f_t(\theta_t) \leq \epsilon$.

$$(*) = \frac{\min\{\eta_{\max} f_t(\theta_t), C'\}}{2} \mathcal{I}\{f_t(\theta_t) \geq \epsilon\} + \frac{\min\{\eta_{\max} f_t(\theta_t), C'\}}{2} \mathcal{I}\{f_t(\theta_t) < \epsilon\}$$

$$\implies (*) \geq \frac{\min\{\eta_{\max} f_t(\theta_t), C'\}}{2} \mathcal{I}\{f_t(\theta_t) \geq \epsilon\} \geq \frac{\min\{\eta_{\max}\epsilon, C'\}}{2} \mathcal{I}\{f_t(\theta_t) \geq \epsilon\}$$

Combining the above inequalities, we get that,

$$\|\theta_{t+1} - u\|_2^2 \leq \|\theta_t - u\|_2^2 - \frac{\min\{\eta_{\max}\epsilon, C'\}}{2} \mathcal{I}\{f_t(\theta_t) \geq \epsilon\} + 2\eta_{\max}[f_t(u)]$$

Taking an expectation over the randomness in iteration $t$, conditioned on the past iterates,

$$\mathbb{E}_t[\|\theta_{t+1} - u\|_2^2] \leq \|\theta_t - u\|_2^2 - \frac{\min\{\eta_{\max}\epsilon, C'\}}{2} \Pr[f_t(\theta_t) \geq \epsilon] + 2\eta_{\max}[f(u)]$$
$$\text{(Since } \eta_{\max} \text{ and } u \text{ are both independent of the randomness at iteration } t\text{)}$$

From the conditioning on $G^c$, we know that $f(\theta_t) \geq \epsilon$. Since $f$ is the average of the $f_i$ losses, we know that there exists at least one $j$ s.t. $f_j(\theta_t) \geq \epsilon$. Hence, conditioned on the past iterates, $\Pr[f_t(\theta_t)] \geq \epsilon \geq \frac{1}{n}$. Combining this lower bound with the above inequality, we get that,

$$\mathbb{E}_t[\|\theta_{t+1} - u\|_2^2] \leq \|\theta_t - u\|_2^2 - \frac{\min\{\eta_{\max}\epsilon, C'\}}{2n} + 2\eta_{\max}[f(u)]$$

$$= \|\theta_t - u\|_2^2 - \frac{\min\{1, C'\}}{2n} + \frac{2 f(u)}{\epsilon} \qquad \text{(Setting } \eta_{\max} = \frac{1}{\epsilon}\text{)}$$

Define $u^*$ to be the max-margin solution i.e. $\|u^*\| = 1$ and $\gamma$ to be the corresponding margin, i.e.

$$\gamma := \min_i y_i \langle x_i, u^* \rangle \qquad (30)$$

For a scalar $\beta > 0$,

$$f(\beta u^*) = \frac{1}{n} \sum_{i=1}^n \ln(1 + \exp(-y_i \langle x_i, \beta u^* \rangle)) \leq \frac{1}{n} \sum_{i=1}^n \exp(-y_i \langle x_i, u^* \rangle) \leq \exp(-\beta\gamma) \qquad (31)$$

Consider $u = \beta u^*$ where $\beta = \frac{1}{\gamma} \ln\left(\frac{n}{\epsilon^2}\right)$ implies that $f(u) \leq \frac{\epsilon^2}{n}$. Since $\epsilon \leq 1$ and $n \geq 1$, $f(u) \leq \epsilon$. Using this relation with the above inequality, we get,

$$\mathbb{E}_t[\|\theta_{t+1} - u\|_2^2] \leq \|\theta_t - u\|_2^2 - \frac{\min\{1, C'\}}{2n} + \frac{2\epsilon}{n}$$

For $\epsilon \leq \frac{\min\{1, C'\}}{8} \leq 1$,

$$\mathbb{E}_t[\|\theta_{t+1} - u\|_2^2] \leq \|\theta_t - u\|_2^2 - \frac{\min\{1, C'\}}{4n}$$

Taking expectation w.r.t. the randomness from iterations $t = 0$ to $T - 1$,

$$\mathbb{E}[\|\theta_{t+1} - u\|_2^2] \leq \mathbb{E}[\|\theta_t - u\|_2^2] - \frac{\min\{1, C'\}}{4n}$$

$$\implies \mathbb{E}[\|\theta_T - u\|_2^2] \leq \|\theta_0 - u\|_2^2 - \frac{\min\{1, C'\}}{4n} T \qquad \text{(Summing from } t = 0 \text{ to } T - 1\text{)}$$

$$\leq \beta^2 - \underbrace{\frac{\min\{1, C'\}}{4n}}_{:=C} T \qquad \text{(Since } \theta_0 = 0 \text{ and } \|u\| = \beta\text{)}$$

Since $f$ satisfies Assn. 1 to 3, $f$ is 1 uniform smooth and $f(u) \leq \epsilon$, then, using Lemma 4 with $M = \frac{1}{8}$,

$$\mathbb{E}[f(\theta_T) - f(u)] \leq \frac{\epsilon}{2} + \underbrace{[\nu^2 + 1]}_{:=L} \mathbb{E}\frac{\|\theta_T - u\|_2^2}{2} \leq \frac{\epsilon}{2} + \frac{L}{2}\left[\beta^2 - C\,T\right]$$

$$\implies \mathbb{E}[f(\theta_T)] \leq \frac{3\epsilon}{2} + \frac{L}{2}\left[\beta^2 - C\,T\right] \qquad \text{(Since } f(u) \leq \epsilon\text{)}$$

In order to ensure that $\mathbb{E}[f(\theta_T)] \leq \frac{3\epsilon}{2}$, it is sufficient to set $T$ as:

$$T \geq \frac{\beta^2}{C} = \frac{4\,n}{\min\{C', 1\}\,\gamma^2}\left[\ln\left(\frac{n}{\epsilon^2}\right)\right]^2$$

Since the above calculations were conditioned on $G^c$, we can conclude that after $T = \frac{4\,n}{\min\{C', 1\}\,\gamma^2}\left[\ln\left(\frac{n}{\epsilon^2}\right)\right]^2$ iterations, $\mathbb{E}[\inf_{t \in [T]} f(\theta_T)|G^c] \leq \frac{3\epsilon}{2}$. Hence, from Eq. (29)

$$\mathbb{E}\left[\inf_{t \in [T]} f(\theta_t)\right] \leq \frac{5\epsilon}{2}$$

$\square$

**Corollary 7.** *For logistic regression on linearly separable data with margin $\gamma$, if, for all $i$, $\|x_i\| \leq 1$, for a fixed $\epsilon \in \left(0, \frac{C'}{8}\right)$ where $C' := \frac{1}{\lambda_1} = \frac{1}{648}$, SGD−SLS with $\eta_{\max} = \frac{1}{\epsilon}$ and $c = \frac{2}{3}$ requires*

$$T = O\left(\min\left\{n, \frac{1}{\epsilon}\right\}\frac{1}{\gamma^2}\left[\ln\left(\frac{n}{\epsilon^2}\right)\right]^2\right)$$

*iterations to ensure that $\mathbb{E}[\inf_{t \in [T]} f(\theta_t)] \leq \frac{5\epsilon}{2}$.*

*Proof.* Combining Thm. 4 and Thm. 5 completes the proof. $\square$

**Theorem 6.** *For logistic regression on linearly separable data with margin $\gamma$, if, for all $i$, $\|x_i\| \leq 1$, for a fixed $\epsilon \in \left(0, \frac{\min\{\frac{1}{2}, C'\}}{8}\right)$ where $C' := \frac{1}{\lambda_1} = \frac{1}{648}$, SGD−SLS with $\eta_{\max} = \frac{1}{\epsilon}$ and $c = \frac{2}{3}$ and guaranteeing that for all $t \in [T]$, $\Pr\left[f_t(\theta_t) \geq \frac{\epsilon}{2}\right] = 1$ requires $T = O\left(\frac{1}{\gamma^2}\left[\ln\left(\frac{1}{\epsilon^2}\right)\right]^2\right)$ iterations to ensure that $\mathbb{E}[f(\theta_T)] \leq \frac{3\epsilon}{2}$.*

*Proof.* Note that from Lemma 1, we know that $\eta_t \geq \min\left\{\eta_{\max}, \frac{C'}{f_t(\theta_t)}\right\}$ where $C' := \frac{1}{\lambda_1}$. Since $f_t(\theta) \geq 0$, combining this with Lemma 5, we get that,

$$\|\theta_{t+1} - u\|_2^2 \leq \|\theta_t - u\|_2^2 - \left(2 - \frac{1}{c}\right)\min\{\eta_{\max}f_t(\theta_t), C'\} + 2\eta_{\max}[f_t(u)]$$

$$= \|\theta_t - u\|_2^2 - \frac{\min\{\eta_{\max}f_t(\theta_t), C'\}}{2} + 2\eta_{\max}[f_t(u)] \qquad \text{(Setting } c = \frac{2}{3}\text{)}$$

Since the algorithm guarantees that $f_i(\theta_t) \geq \frac{\epsilon}{2}$ for all $t$, $f_t(\theta_t) \geq \frac{\epsilon}{2}$. Hence,

$$\leq \|\theta_t - u\|_2^2 - \frac{1}{2} \min\left\{\frac{\eta_{\max}\epsilon}{2}, C'\right\} + 2\eta_{\max}[f_t(u)]$$

$$= \|\theta_t - u\|_2^2 - \frac{1}{2} \min\left\{\frac{1}{2}, C'\right\} + \frac{2f_t(u)}{\epsilon} \qquad \text{(Setting } \eta_{\max} = \frac{1}{\epsilon}\text{)}$$

Define $u^*$ to be the max-margin solution i.e. $\|u^*\| = 1$ and $\gamma$ to be the corresponding margin, i.e.

$$\gamma := \min_i y_i \langle x_i, u^* \rangle \tag{32}$$

For a scalar $\beta > 0$,

$$f(\beta u^*) = \frac{1}{n} \sum_{i=1}^n \ln(1 + \exp(-y_i \langle x_i, \beta u^* \rangle)) \leq \frac{1}{n} \sum_{i=1}^n \exp(-y_i \langle x_i, u^* \rangle) \leq \exp(-\beta\gamma) \tag{33}$$

Consider $u = \beta u^*$ where $\beta = \frac{1}{\gamma} \ln\left(\frac{1}{\epsilon^2}\right)$ implies that $f(u) \leq \epsilon^2$. Since $\epsilon \leq 1$, $f(u) \leq \epsilon$. Using this relation with the above inequality, we get,

$$\leq \|\theta_t - u\|_2^2 - \frac{1}{2} \min\left\{\frac{1}{2}, C'\right\} + 2\epsilon$$

$$\leq \|\theta_t - u\|_2^2 - \frac{1}{4} \min\left\{\frac{1}{2}, C'\right\} \qquad \text{(Since } \epsilon \leq \frac{1}{8} \min\left\{\frac{1}{2}, C'\right\}\text{)}$$

Taking expectation w.r.t. the randomness from iterations $t = 0$ to $T - 1$,

$$\mathbb{E}[\|\theta_{t+1} - u\|_2^2] \leq \mathbb{E}[\|\theta_t - u\|_2^2] - \frac{\min\left\{\frac{1}{2}, C'\right\}}{4}$$

$$\implies \mathbb{E}[\|\theta_T - u\|_2^2] \leq \|\theta_0 - u\|_2^2 - \frac{\min\left\{\frac{1}{2}, C'\right\}}{4} T \qquad \text{(Summing from } t = 0 \text{ to } T - 1\text{)}$$

$$\leq \beta^2 - \underbrace{\frac{\min\left\{\frac{1}{2}, C'\right\}}{4} T}_{:=C} \qquad \text{(Since } \theta_0 = 0 \text{ and } \|u\| = \beta\text{)}$$

Since $f$ satisfies Assn. 1 to 3, $f$ is 1 uniform smooth and $f(u) \leq \epsilon$, then, using Lemma 4 with $M = \frac{1}{8}$,

$$\mathbb{E}[f(\theta_T) - f(u)] \leq \frac{\epsilon}{2} + \underbrace{[\nu^2 + 1]}_{:=L} \mathbb{E}\frac{\|\theta_T - u\|_2^2}{2} \leq \frac{\epsilon}{2} + \frac{L}{2} \left[\beta^2 - CT\right]$$

$$\implies \mathbb{E}[f(\theta_T)] \leq \frac{3\epsilon}{2} + \frac{L}{2} \left[\beta^2 - CT\right] \qquad \text{(Since } f(u) \leq \epsilon\text{)}$$

In order to ensure that $\mathbb{E}[f(\theta_T)] \leq \frac{3\epsilon}{2}$, it is sufficient to set $T$ as:

$$T \geq \frac{\beta^2}{C} = \frac{4}{\min\{C', \frac{1}{2}\}\gamma^2} \left[\ln\left(\frac{1}{\epsilon^2}\right)\right]^2$$

$\square$

