# OpenReview forum: "Armijo Line-search Can Make (Stochastic) Gradient Descent Provably Faster"
_ICML.cc/2025/Conference — ICML 2025 poster_

### Official Review · Reviewer_8uuj · 2025-02-28

**Overall Recommendation:** 4

**Summary:**

Establishes rates for Armijo line search in several settings.

**Claims And Evidence:**

There are some interesting results in this paper, and I like the expansion of our understanding of the (L0, L1)-smoothness condition and it's generalizations.

I object to the use of stochastic to refer to analysis under interpolation condition. These are profoundly different settings and they should not be confused. The title of the paper is misleading and would suggest to readers that there is some novelty in the paper that addresses the full stochastic setting.

My major concern with this paper is that it doesn't have a related work section, and seems to only cite very old papers concerning line searches, as well as a nearly 20 year old text book. Research papers on line searches are normally published in the optimization literature not so much the machine learning literature, and I am not familiar with the state of the art here, but it's an enormous and heavily studied research area and there are surely related work that should be cited in the last 50 years since the Armijo line search was introduced.

I would like to ask the authors to write a 1 page related work section that includes some recent state-of-the art research (at least some citations from the last few years), as well as older works. Please include this in your rebuttal and add it to the final camera ready.

EDIT: The additional literature review is reasonable. Please also include the missing references mentioned by other reviewers also. I have updated my score.

**Essential References Not Discussed:**

See above.

**Experimental Designs Or Analyses:**

N/A

**Methods And Evaluation Criteria:**

N/A

**Other Comments Or Suggestions:**

Typo line 146 “satisfiesAssumptions”

**Other Strengths And Weaknesses:**

N/A

**Questions For Authors:**

N/A

**Relation To Broader Scientific Literature:**

See Above.

**Theoretical Claims:**

Established results seem solid.

---

> ### Author Rebuttal · Authors · 2025-03-30
>
> We thank the reviewer for their helpful feedback and address their concerns below.
>
> > *(1) My major concern with this paper is that it doesn't have a related work section, and seems to only cite very old papers concerning line searches, as well as a nearly 20 year old text book...Please include this in your rebuttal and add it to the final camera ready.*
>
> Even though we do not have an explicit related work section, we believe that we have cited the most relevant work. This includes recent work on Armijo line-search (Vaswani et al., 2019a; Galli et al., 2024; Hubler et al., 2024), Polyak step-sizes (Loizou et al., 2021) and variants of normalized gradient descent (Mei et al., 2021; Axiotis & Sviridenko, 2023; Taheri & Thrampoulidis, 2023). Moreover, we have added the most relevant and recent references for each example – logistic regression (Axiotis & Sviridenko, 2023; Freund et al., 2018; Wu et al., 2024), generalized linear models (Mei et al., 2020; Hazan et al., 2015), softmax policy gradient (Mei et al., 2020; Asad et al., 2024; Lu et al., 2024), two layer neural networks (Taheri & Thrampoulidis, 2023). In the final version of the paper, we will add the following extended comparison and the corresponding references.
>
> **Comparison to GD with adaptive step-sizes on uniform smooth functions**: There has been substantial work on designing adaptive step-sizes for GD. However, most of this literature (for example (Orabona & Pal, 2016; Malitsky & Mishchenko, 2019; Carmon & Hinder, 2022; Khaled et al., 2023)) considers either the uniform-smooth, convex or non-smooth but Lipschitz, convex settings. In the uniform-smooth, convex case, the focus of these papers is to design more efficient ways to adapt to the smoothness constant, and the resulting methods achieve the same rate as GD with constant step-size. In contrast, we consider Armijo line-search which is the classic way to set the step-size for GD on smooth functions. The focus of our work is to identify when $\texttt{GD-LS}$ can be *provably* faster than $\texttt{GD(1/L)}$. To that end, we have identified a class of non-uniform smooth functions and shown that numerous examples in machine learning satisfy these properties. We believe that we are the first to identify such a broad range of examples and demonstrate the provable improvement of $\texttt{GD-LS}$ over $\texttt{GD(1/L)}$.
>
> **Comparison to GD on non-uniform smooth functions:** There has been recent work on analyzing GD (with/without clipping) on functions satisfying (L0, L1) non-uniform smoothness (Zhang et al., 2019; 2020; Koloskova et al., 2023). This work requires the knowledge of L0 and L1. In contrast, as explained in Contribution 1 and in lines 133-142 (after Proposition 1) of our paper, the proposed non-uniform smoothness condition is different and the $\texttt{GD-LS}$ algorithm does not require the knowledge of L0, L1 or other problem-dependent constants.
>
> **Comparison to $\texttt{GD-LS}$ on non-uniform smooth functions**: The most relevant paper to our setup is (Hubler et al., 2024) and we have compared to this in Lines 58-64. In particular, this paper analyzes the convergence of $\texttt{GD-LS}$ on functions satisfying a different notion of (L0, L1) non-uniform smoothness. However, their algorithm requires the knowledge of L1 and the resulting rate is the same as that of $\texttt{GD(1/L)}$
>
> We hope that this addresses the reviewer’s concerns and helps them better contextualize our contributions. If the reviewer is aware of a key reference we have missed, we would be happy to include and compare to it.
>
> > *(2) I object to the use of stochastic to refer to analysis under interpolation condition... The title of the paper is misleading and would suggest to readers that there is some novelty in the paper that addresses the full stochastic setting.*
>
> As is evidenced by the numerous papers analyzing SGD under interpolation, we first note that the interpolation setting is an important special case. Moreover, this setting can serve as a building block towards analyzing the full stochastic setting. For example, the stochastic line-search (Vaswani et al., 2019b) and stochastic Polyak step-sizes (Loizou et al., 2021) were originally designed for the interpolation setting, and have been adapted to the full stochastic case by combining these techniques with appropriately decreasing step-sizes (Orvieto et al., 2022; Vaswani et al., 2022; Jiang & Stich, 2023). Moreover, the proof for $\texttt{SGD-SLS}$ in Section 6 is more involved compared to the standard SGD proofs. Given this, we think it is important to highlight that our paper also considers the stochastic setting, and hence we added "stochastic” to the title. Note that we clarify that we are only considering the interpolation setting in the abstract
>
> We do understand the reviewer's point, and will change the title to *Armijo Line-search **Can** Make (Stochastic) Gradient Descent Go Fast* to suggest that the speedup is for some special settings.

---

### Official Review · Reviewer_8bAV · 2025-03-07

**Overall Recommendation:** 2

**Summary:**

The paper analyzes functions where the local smoothness constant is given by \\( L(\\theta) = L_0 + L_1 f(\\theta) \\), which is satisfied by common objectives like logistic regression or regression problems with generalized linear models. The authors prove that Gradient Descent with Armijo Line-Search (GD-LS) converges faster on these functions by showing that the step size selected is lower-bounded by \\( 1/L(\\theta) \\). They extend their analysis to non-convex functions such as generalized linear models (GLMs) with a logistic link function and demonstrate empirically that GD-LS outperforms standard GD with a fixed step size. The paper also discusses the benefits of using a line-search in the stochastic setting by analyzing SGD with a stochastic variant of the Armijo line-search (SGD-SLS). By introducing assumptions on non-uniform smoothness and relating gradient norms to function values, the authors provide a framework that facilitates the analysis of GD with Armijo line-search for functions with exponential-type losses.

**Claims And Evidence:**

Yes.

**Essential References Not Discussed:**

Not applicable.

**Experimental Designs Or Analyses:**

The experiment designs look good.

**Methods And Evaluation Criteria:**

Yes.

**Other Comments Or Suggestions:**

The description of taking $\eta_\max=\infty$ in GD-LS is very unclear. At line 3 in Algorithm 1, the author initializes $\tilde \eta_t$ by $\eta_\max$. By taking $\eta_\max=\infty$, every step $\tilde \eta_t$ in Algorithm 1 will simply be $\infty$. From line 193-194 at the right column and from the proof, it seems that this assumption on $\eta_\max$ is somehow equivalent to taking a sufficiently large $\eta_\max$. Moving the explanation on taking $\eta_\max=\infty$ before Theorem 1 will minimize the confusion.

**Other Strengths And Weaknesses:**

Overall this paper provides a solid theoretical contribution for the study of Armijo line search for proposed (L0-L1) non-uniformly smooth optimization problems.

Nevertheless, the writing of the paper can be improved. For instance, the authors list 6 contributions in the introduction section, but each of them look like the organization of each of section. To be specific, the contribution 1 (i.e., the content of Section 2) is simply the problem formulation. In other words, the main contribution of this work is not easy to find. One potential improvement for this is that the author may list one or two informal version of their main theorem and provide a clear and succinct description of their contributions.

The targeted optimization problem in this paper is limited in terms of application perspective. The authors primarily focus on logistic regression and generalized linear models. Although the authors showed that object for softmax policy gradient also satisfies the assumptions in appendix, the application sides of this paper to real-world use cases are not significant enough for a venue like ICML.

**Questions For Authors:**

No.

**Relation To Broader Scientific Literature:**

The main difference of this paper and other paper analyzing Armijo line search is that this paper consider a specific setting by Assumption 2. It can be viewed as a relaxed version of  symmetric (L0, L1) smoothness discussed in [1]. As this setting is novel, the theoretical contribution of this paper is new.

[1] Gorbunov, Eduard, et al. "Methods for convex $(l_0, l_1) $-smooth optimization: Clipping, acceleration, and adaptivity." arXiv preprint arXiv:2409.14989 (2024).

**Theoretical Claims:**

I browse the proof of Theorem 1 and it looks correct to me.

---

> ### Author Rebuttal · Authors · 2025-03-30
>
> We thank the reviewer for their helpful feedback and address their concerns below.
>
> > *(1) The targeted optimization problem in this paper is limited in terms of application perspective. The authors primarily focus on logistic regression and generalized linear models. Although the authors showed that object for softmax policy gradient also satisfies the assumptions in appendix, the application sides of this paper to real-world use cases are not significant enough for a venue like ICML.*
>
> It is not required for papers in ICML to have "application sides...to real-world use cases''. In fact, the call for papers lists "Theory of Machine Learning (statistical learning theory, bandits, game theory, decision theory, etc.)'' explicitly as a topic of interest.
>
> Having said that, we believe that our paper does have practical implications. In particular, our paper analyzes when $\texttt{GD-LS}$ (a standard optimization method widely used in practice) can be *provably* faster than $\texttt{GD(1/L)}$. Besides achieving adaptivity to the smoothness in a principled manner and hence requiring minimal hyper-parameter tuning (an important consideration from a practical standpoint), $\texttt{GD-LS}$ can result in faster convergence on numerous convex (e.g. logistic regression) and non-convex problems (e.g. generalized linear models, 2 layer neural networks, softmax policy gradient) important in machine learning. We have argued that $\texttt{GD-LS}$ can thus replace more specialized algorithms tailored for these specific problem settings. *Having a single algorithm that can work well across different applications is important from a practical standpoint.*
>
> Furthermore, as evidenced by the experimental results in [Vaswani et al, 2019], $\texttt{SGD-SLS}$ (analyzed in Section 6 of our paper) results in good empirical performance on convex problems that satisfy our assumptions. Under the interpolation assumption, Vaswani et al, 2019 can only prove an $O(1/\epsilon)$ rate for $\texttt{SGD-SLS}$ on smooth convex losses such as logistic regression. However, the empirically, the convergence seems to be faster than $O(1/\epsilon)$ (for example, see the linear convergence on the $\texttt{mushrooms}$ dataset in Figure 3 of [Vaswani et al, 2019]. Our paper (specifically Corollary 5) can explain this fast convergence. *We believe that it is important to understand the behaviour of optimization algorithms, and that these insights can often lead to developing better methods in practice.*
>
> Finally, it is important to note that this same method - $\texttt{SGD-SLS}$ and its non-monotone variants [Galli et al, 2024] result in strong empirical performance on non-convex losses with deep neural networks [Vaswani et al, 2019, Galli et al 2024] where our assumptions are not necessarily satisfied. *This demonstrates that developing methods that are theoretically principled in simple settings can result in empirical gains for real-world use cases.*
>
> > *(2) One potential improvement for this is that the author may list one or two informal version of their main theorem and provide a clear and succinct description of their contributions.*
>
> Thank you for the good suggestion. We will consider making this change in the final version of the paper.
>
> > *(3) Moving the explanation on taking $\eta_{\max} = \infty$ before Theorem 1 will minimize the confusion.*
>
> We will make the change.

---

### Official Review · Reviewer_sPj3 · 2025-03-10

**Overall Recommendation:** 3

**Summary:**

This paper studies gradient descent using Armijo line search to choose the stepsize. They prove convergence rates for the algorithm under a non-uniform smoothness condition, and specialize these results to logistic regression, softmax policy gradient for multi-armed bandit problems, and GLMs with the logistic link function. They also have convergence guarantees for a stochastic version of the algorithm for finite-sum structured separable logistic regression.

**Claims And Evidence:**

For the most part, the claims made in this paper are clear and backed up by convincing evidence. There are three main places where this is not the case:

1) At various times in the paper, the authors claim that GD-LS is better than GD-(1/L) by comparing the upper bounds they prove for GD-LS against upper bounds for GD-(1/L) derived in other papers. This is not a sound way of making the comparison---how do we know that those guarantees for GD-(1/L) are tight? Without lower bounds for GD-(1/L), it is possible that GD-(1/L) is just as good (or even better) than GD-LS in some or all of the cases considered in this paper. I agree with the authors that this seems unlikely given that GD-(1/L) needs to use a fixed stepsize, but to know this definitively, we'd need an actual proof.

2) Proposition 3 seems to require that the reward vector (and therefore the optimal action) are known---otherwise, it would not be possible to implement GD-LS to sole the problem. The authors later argue that GD-LS is superior to some alternatives because they require knowledge of r(a^*). Even though this is true, GD-LS requires knowledge of the reward vector (and therefore the optimal action), so there isn't much difference. I understand what they are trying to get at---their algorithm is less "specialized" to the problem---but I think the advantage would be more clear if this distinction was made a little bit more clearly. I believe they are trying to gesture towards GD-LS being a good method for more interesting cases when we don't just know the reward vector / optimal arm, but I think the problem setup in this section could use a little more explanation. In general, you wouldn't know the reward vector, so you'd probably need to do some kind of online/stochastic optimization here---is there reason to think GD-LS would be good for that? Does the line search work well enough when you only have online/stochastic gradients and can only estimate the function value? Section 6 and the last sentence of page 6 suggests that they think the answer is yes, but it's not completely clear to me precisely what they are showing vs just claiming for the specific problem here.

3) A lot of the claims in this paper are based on logistic regression or similar objectives where, for most gradient methods, convergence is initially fast and then slows down as the loss / gradients / hessian become very small. This is a rare example in optimization where very large stepsizes are possible / desirable, and it is not necessarily that clear that these gains from GD-LS would extend to more typical problems where the maximum possible stepsize is not so large.

**Essential References Not Discussed:**

See above.

**Ethical Review Concerns:**

No concerns.

**Experimental Designs Or Analyses:**

There is not a ton of detail given for the experiments in the paper (this is probably fine given their simplicity), so it is hard to say.

**Methods And Evaluation Criteria:**

Experimental evidence is provided for GD-LS versus other alternatives, but I think the stochastic setting is much more interesting for typical ML applications, and there are no experiments covering this. Experiments for SGD-SLS on logistic regression is one thing, but given  that the upshot of Lemma 3 is not very clear outside of the application to Corollary 5, it would probably be interesting to see how well SGD-SLS works on other (non-logistic regression) supervised learning problems, both convex and non-convex.

At the bottom of pg 5, the authors point out that GD with the Polyak stepsize can achieve the same thing as Theorem 2. This raises the question of whether the Polyak stepsize can also match the guarantee, e.g., in Corollary 1. On the one hand, to implement the Polyak stepsize you need to know $f^*$, which is not necessarily available. On the other hand, you only need to evaluate the function value once (versus multiple times for the backtracking line search). I feel making this comparison more thoroughly is important context for this work.

**Other Comments Or Suggestions:**

Small comments:

Assumption 2: Are parts (a) and (b) equivalent statements, or are these separate conditions that need to both be satisfied? It is not obvious whether or not (a) <=> (b).

Theorem 1: probably have $\| \nabla f(\theta) \|_2^2 \geq [f(\theta) - f^*]^2 / R$ be "Assumption 4" rather than sneaking it into the theorem statement? Then for e.g. Corollary 1 I would just note that convexity -> assumption 4.

Theorem 1: it might be better to write this as $\max\{ 2 R \lambda_1, 1 \}\left( \frac{f^* + \frac{\lambda_0}{\lambda_1}}{\epsilon} + 1 \right) \ln\left( \frac{f(\theta_0) - f^*}{\epsilon} \right)$ in all cases. In case 1, the bound is a little looser, but the lambda_0 / lambda_1 term is less than f^*, so by at most a factor of 2. And in case two, similarly, this is only looser by a log factor. On the other hand, it is easier to read and understand if there is only one case. Relatedly, in the last full paragraph on page 4, you describe the guarantee as exhibiting two phases, but unless you are considering the ln(1/eps) term to be "large", I don't see what is so different in the first/second phase---it is 1/eps or ln(1/eps)/eps convergence in all cases. The proof of Theorem 1 involves breaking things up into two phases, but that is more about the analysis than the actual convergence guarantee, at least as it is written.

**Other Strengths And Weaknesses:**

Nothing else to add.

**Questions For Authors:**

See other comments.

**Relation To Broader Scientific Literature:**

This paper is trying to show that GD-LS compares favorably to alternatives (especially GD-(1/L)). I am honestly not very familiar with prior analysis of Armijo line search beyond the most basic cases, so this work appears novel to the best of my knowledge.

Given the emphasis of this paper, I think it would be useful to spend some additional time putting this work into the context of related work.  E.g. see my comments above about the Polyak stepsize rule. Also, this paper puts an emphasis on the fact that GD-LS is adaptive to the problem parameters like the smoothness constant, so it would probably be useful to make some comparison to recent work on adaptive stepsizes for GD (e.g. [Carmon and Hinder 2022] and many more).

**Theoretical Claims:**

I read the proofs and they appear basically sound as far as I can tell.

The only issues I see are:

- I believe Corollary 2 and 6 are supposed to be the same thing? But, Corollary 6 in the appendix appears to be slightly different than Corollary 2 in the main text.

- Proposition 3: Shouldn't the reward gap be defined as \min_a rather than \max_a?

- Assumption 4: As stated, this appears to just a complicated way of saying that $f$ has no stationary points that are not global minima (for arbitrary \zeta, just take $\mu(\theta) = \|\nabla f(\theta)\|^\zeta / [f(\theta) - f^*]$, which is strictly positive for such $f$). I think it would be better to say that "$f$ satisfies the $(\zeta,\mu)$-gradient domination condition if..." because the $\mu$ function plays an important quantitative role (beyond just the fact that it exists).

- Pg 4: I'm being a little pedantic here, but this sentence "As a concrete example, consider the case when $f^* = \delta \epsilon$ where $\delta \geq 1$ is a constant independent of $\epsilon$" does not make very much sense. Of course $\delta$ depends on $\epsilon$, it is exactly $\delta = f^* / \epsilon$! Calling it a "constant" let's you ignore it in the big-O convergence rate $O(R \ln 1/\epsilon)$, but I would say this is cheating a little bit. To make a similar statement, I would phrase it differently, something like: if you are only trying to reach accuracy $\epsilon = \Theta(f^*)$, then GD-LS will result in $O(R \ln 1/\epsilon)$ convergence. This isn't a big deal and I apologize for the pedantry, but I am not a fan of using a sleight of hand to hide things in big-O notation.

---

> ### Author Rebuttal · Authors · 2025-03-30
>
> We thank the reviewer for their helpful feedback and address their concerns below. **For all unaddressed comments, we agree with the reviewer's suggestion and will make the corresponding change.**
>
> > *(1) Lower-bounds for $\texttt{GD(1/L)}$*
>
> For logistic regression, Theorem 3 in Wu et al, 2024 shows that $\texttt{GD(1/L)}$ (or more generally, GD with any constant step-size that guarantees monotonic descent) cannot have a convergence rate faster than $\Omega(1/\epsilon)$. Hence, both $\texttt{GD-LS}$ and $\texttt{SGD-SLS}$ on logistic regression are provably faster than their constant step-size analogs.
>
> For the softmax policy gradient in Section 5.1, Theorems 9, 10 in Mei et al, 2020 show that GD with any constant step-size cannot achieve a rate faster than $\Omega(1/\epsilon)$ on both bandit and MDP problems. We mention this after Corollary 4 in the paper. For the GLM problem in Section 5.2, we are not aware of a lower bound for $\texttt{GD(1/L)}$ or normalized GD. We will include these comparisons.
>
> > *(2) Softmax PG - Setup*
>
> We note that the linear convergence result in Corollary 3 also applies to the general MDP problem. See Proposition 6 for the problem setting. $\texttt{GD-LS}$ in Section 5.1 is equivalent to softmax policy gradient in the "deterministic'' or "exact'' setting. This setting is commonly used as a testbed for analyzing policy gradient algorithms [Mei et al 2020; Section 4, Lu et al 2024]. For general MDPs, the deterministic setting corresponds to knowing the rewards and transition probabilities, and is the same setting under which classic RL algorithms such as value iteration or policy iteration are analyzed. This was our motivation to consider the problem setup in Section 5.1.
>
> > *(3) Softmax PG - Handling stochasticity*
>
> For the general MDP problem, in cases where the rewards/transition probabilities are unknown, the policy gradient is estimated by interacting with the environment. Standard policy gradient results (e.g [Theorem 29, Agarwal et al, 2020] can then prove convergence up to an error incurred because of the estimation.
>
> For the specific case of stochastic multi-armed bandits with unknown rewards, recent work [Mei et al, 2023] has proven stronger global convergence results. In particular, Mei et al, 2023 use importance sampling to construct the stochastic softmax policy gradient and use it with constant step-size SGD. The resulting algorithm can be proven to converge to the optimal arm. From Proposition 3, we know that the objective function satisfies Assumptions 2, 3, and 4. It also satisfies the interpolation condition [Theorem 4.2 in Mei et al, 2023], required for the $\texttt{SGD-SLS}$ analysis in Section 6. Hence, we believe that a variant of $\texttt{SGD-SLS}$ could be a good algorithm for the stochastic multi-armed bandit problem. However, since the softmax policy objective is non-convex, we cannot directly use the results in Section 6 and leave this interesting direction to future work.
>
> > *(4) ....not clear if gains from GD-LS would extend to more typical problems..*.
>
> We have tried to argue that losses that satisfy our non-uniform smoothness assumptions are in fact, not as rare. To illustrate this, we have given numerous examples ranging from classification in supervised learning to policy gradient in reinforcement learning. In general, applications where we use losses with an exponential tail (e.g. exponential, logistic loss) or those that use a softmax function to parameterize probabilities can benefit from using a line-search. Another example that we did not study in detail is noise contrastive estimation [Lu et al 2021], where GD is slow because of the flatness of the landscape, and $\texttt{GD-LS}$ can potentially improve the convergence.
>
> We agree that large gains from $\texttt{GD-LS}$ are not always possible (e.g. for quadratics) and if the function is only uniformly smooth, $\texttt{GD-LS}$ will converge at the same rate as $\texttt{GD(1/L)}$. However, even in this case, $\texttt{GD-LS}$ enables setting the step-size without estimating or conservatively bounding the smoothness constant and thus has a practical benefit.
>
> > (5) *Comparison to Polyak step-size*
>
> The proof of Theorem 1 (and non-convex results) relies on the monotonic decrease in the function values guaranteed by $\texttt{GD-LS}$. GD with the Polyak step-size is not a monotonic descent method, and it is unclear how to extend the proofs.
>
> > (6) *Experimental Evaluation*
>
> See Point (4) in our response to Rev. jUfU
>
> > *(7) Comparison to recent work on adaptive step-sizes*
>
> See Point (1) in our response to Rev. 8uuj
>
> > (8) Cor. 2 vs Cor. 6:
>
> Cor. 2 is for $\texttt{GD-LS}$, Cor. 6 is for GD with the Polyak step-size.
>
> > (9) Proposition 3: Def. of reward gap
>
> It is the difference between the rewards of the best (optimal) and second-best arm [see Lemma 17 in Mei et al, 2020].
>
> > (10) Assumption 2 (a) and (b)
>
> These are separate (but related) conditions that need to both be satisfied.

---

### Official Review · Reviewer_jUfU · 2025-03-14

**Overall Recommendation:** 3

**Summary:**

This paper investigates the effectiveness of the Armijo line-search (Armijo-LS) method for step-size selection in gradient descent (GD) algorithms. The authors introduce a class of functions that satisfy a non-uniform smoothness condition and show that GD with Armijo-LS (GD-LS) can adapt to local smoothness, leading to faster convergence compared to standard GD with a fixed step-size of $1/L$ (GD(1/L)). The analysis is further extended to the stochastic setting.

## update after rebuttal

Thanks the authors for answering my questions. I decide to keep my score.

**Claims And Evidence:**

Yes.

**Essential References Not Discussed:**

No.

**Experimental Designs Or Analyses:**

Almost no experiment.

**Methods And Evaluation Criteria:**

Yes.

**Other Comments Or Suggestions:**

See above.

**Other Strengths And Weaknesses:**

Weaknesses:

* Independence of Assumptions: Assumption 2(b) is a direct consequence of the generalized smoothness condition used in prior literature. The proof primarily relies on Assumptions 2(c), 3, and 5, with Assumption 2(c) being implied by the other two. Since assumptions should ideally be independent, I suggest presenting only Assumptions 3 and 5 to clarify their distinct roles in the proof. Alternatively, would it be possible to provide an example satisfying Assumption 2(c) and 3 without Assumption 5?

* Positivity Assumption (Assumption 1): The paper assumes that $f$ is positive over the entire space. While it is possible to shift
$f$ by subtracting $f^*=\min f$, in most cases, $f^*$ is unknown. Would it be possible to remove Assumption 1 and instead consider
$|f|$ in the relevant conditions?

* Missing Reference: The paper should reference [1], which provides gives tight rates for GD(LO) that are faster than for GD(1/L) in strongly convex setting. This aligns with Assumption 4 when $\xi=2$.

[1] De Klerk E, Glineur F, Taylor AB. On the worst-case complexity of the gradient method with exact line search for smooth strongly convex functions. Optimization Letters, 2017, 11:1185-99.

* Experimental Validation: More experiments should be conducted to further validate the efficiency of the proposed line-search algorithm, particularly in multi-armed bandit (MAB) problems and stochastic settings.

Strengths:

* The newly proposed conditions are well-discussed and provide valuable insights.
* The paper reveals an interesting result: GD-LS provably adapts to local smoothness, leading to improved convergence rates.
* The paper is well-written and easy to follow.

**Questions For Authors:**

See above.

**Relation To Broader Scientific Literature:**

See below.

**Theoretical Claims:**

Yes.

---

> ### Author Rebuttal · Authors · 2025-03-30
>
> We thank the reviewer for their helpful feedback and address their concerns below.
>
> > *(1) Independence of Assumptions*
>
> Assumptions 3 and 5 do indeed imply Assumption 2(b). The reason we did not use Assumption 5 as our main assumption is because Assumption 5 cannot be satisfied for logistic regression, multi-class classification with $L_c = 0$. We explained this in Lines 137-142, and note that having $L_c = 0$ is essential to derive a linear convergence rate for logistic regression in Corollary 2. To see this from a technical perspective, consider $f$ to be a finite sum $f(\theta) = \frac{1}{n}\sum_i f_i(\theta)$ (for example, the logistic regression loss in Proposition 1). If $f_i$ satisfies Assumption 5 with constants $L_c, L_g$, it does necessarily imply that $f$ also satisfies Assumption 5 with the same constants (see Proposition 7 for a counter-example).
>
> In contrast, using Assumption 2 as our main assumption enables us to prove that the losses corresponding to logistic regression, multi-class classification satisfy this assumption with $L_0 = 0$, and this enables us to prove the linear convergence rate in Corollary 2. Again, from a technical perspective, we can do this because if $f_i$ satisfies Assumption 2 with constants $L_0$ and $L_1$, then $f$ also satisfies this assumption with the same constants (please refer to the proof of Lemma 5).
>
> In summary, we only use Assumption 5 for $f_i$ (and not $f$) corresponding to common finite-sum losses as a way (using Lemma 5) to show that $f$ satisfies Assumption 2. For example, see the proof of Proposition 1, where we instantiate this logic for logistic regression.
>
> > *(2) Positivity Assumption (Assumption 1)*
>
> From an algorithmic perspective, Assumption 1 is quite benign. As the reviewer suggests, it is possible to shift $f$ by subtracting $f^*$ and ensure that the resulting function $h(\theta) := f(\theta) - f^\ast$ is always non-negative. Since $f^\ast$ is a constant, $\nabla h(\theta) = \nabla f(\theta)$, meaning that we do not require the knowledge of $f^\ast$ to implement GD. Moreover, the Armijo condition on $h(\theta)$ has $f^\ast$ on both sides, implying that implementing $\texttt{GD-LS}$ on the non-negative function $h(\theta)$ does not require the knowledge of $f^*$. We use this property in Section 5.1 (see Proposition 3 and the corresponding discussion). Finally, we note that most common losses in machine learning (cross-entropy, exponential, and squared) are non-negative.
>
> > *(3) Missing Reference*
>
> Thank you for directing us to this paper. The assumption in their paper is more restrictive than ours, as they assume strong convexity, whereas we only rely on gradient domination. This enables us to consider non-convex losses such as generalized linear models. Additionally, they focus on exact line search, while we consider Armijo line search. We will cite this paper in the final version.
>
> > *(4) Experimental Validation*
>
> In the deterministic setting (with access to full gradients), $\texttt{GD-LS}$ is a standard algorithm for (non)-convex minimization and is widely used in practice. In the stochastic setting, Vaswani et al. (2019b); Galli et al. (2024) have empirically evaluated the performance of $\texttt{SGD-SLS}$ and its non-monotone variant for convex losses corresponding to linear and logistic regression and non-convex losses with deep neural networks. These supervised learning experiments have demonstrated the efficacy of $\texttt{SGD-SLS}$. As we explain in Section 6, under the interpolation assumption, Vaswani et al. (2019a) can only prove an O(1/ϵ) rate for SGD-SLS on smooth convex losses such as logistic regression. However, empirically, the convergence seems to be faster than O(1/ϵ) (for example, see the linear convergence on the mushrooms dataset in Vaswani et al. (2019b, Figure 3). Our paper (specifically Corollary 5) can explain this fast convergence.
>
> For the softmax policy optimization problem in RL, we note that Lu et al. (2024, Figure 1) have shown the linear convergence of $\texttt{GD-LS}$ on tabular Markov decision processes. However, they could only prove an O(1/ϵ) convergence rate for the resulting algorithm (Lu et al., 2024, Theorem 1). In order to attain a linear convergence rate in theory, they propose a non-standard line-search scheme that requires the knowledge of the optimal value function. On the other hand, Corollary 4 in our paper proves a linear convergence rate for $\texttt{GD-LS}$ (the standard line-search) on bandit problems, and the same proof extends to the MDP setting (using Proposition 6 in our paper).
>
> We will include these comparisons and explanations in the final version of the paper.

---

### Decision · Program_Chairs · 2025-05-01

**Decision:**

Accept (poster)

**Comment:**

The paper analyzes Armijo line descent under a new generalization of (L_0,L_1)-smoothness that bounds the spectra of the Hessian by a constant plus a multiple of the function value. The paper also assumes in addition that the gradient norm is bounded by a constant plus a multiple of the function value. The assumptions are properly justified by presenting three problems where they (non-trivially) hold: logistic regression, a particular non-convex GLM, and softmax policy gradient  for multi-armed bandits and tabular Markov decision processes. This is sufficient to justify the assumptions. The paper has a few meta-theorems, which are specialized in the setting of logistic regression for separable data (both SGD and GD),  and softmax policy optimization for multi-armed bandits. I think the contributions and clarity are enough to warrant publication. But I would like to see the authors add additional references to other methods (in particular polyak based stepsizes) and address some minor issues I've compiled below.

1- **Descent property, Polyak stepsize and additional comparison.**  Regarding your answer to Reviewer sPj3 on if Theorem 1 and Corollary 1 could be applied to the Polyak stepsize, note that GD with a Polyak stepsize *is monotonically decreasing*, even in the stochastic setting, so long as the loss is convex, see for instance Theorem 2.3 in [Garrigos2022]. In this regard, in both your full batch (when assuming $f(x^*) =0$) and stochastic setting you are competing directly with the Polyak stepsize.  Furthemore, the resulting stepsize of Armijo line search and the polyak stepsize are both local estimates of smoothness. Moreover, several closely related (though not the same) types of smoothness conditions have now been used to study the Polyak step size, see [Yuki2024, Vankov2025]. These are very recent works (online since October 2024), but I think it would enrich the reader's perspective if you point out some of these connections.

2- **The growth condition you assume in Theorem 1** sometimes goes by the name of the weak gradient domination condition see (Agarwal et al., 2021;Mei et al., 2020, Yuan et al 2022). It might be worth naming it, and pointing where exactly to find proofs of convergence under this condition.

3- **Corollary 1** you show that this weak gradient domination holds **along the trajectory** so long as the method is monotonic and the function is convex. But here you need to adjust the assumption of Theorem 1 to assume weak domination over the trajectory, instead of globally as is written now.

4- Clarify if $f(x^*) =0$ is assumed for Theorem 2, I wasn't sure when reading. Though it is clear in Corollary 2 you assume f(x^*)=0.

5- Lemma 3, calling $\chi^2$ the noise of the stochastic gradients is confusing. Calling the "function noise" is much more accurate, as done in  Definition 4.13 in [Garrigos2023, Section 4.3.2]

6- I see a small (inconsequential) mistake in two of the proofs: "For normalized data, s.t. $\vert x\rvert =1$ the logistic regression loss is convex, is uniform smooth with $L = \lambda_{\max}(X^\top X) =1$ satisfies Assumption 2 with $L1=1$." This should be instead   $L \leq \lambda_{\max}(X^\top X) =n$ or   $L \leq (1/n) \lambda_{\max}(X^\top X) \leq 1$.

7- I partly agree with Reviewer  8uuj, the title is a little bit misleading, in that, when I read the title, I was hoping for a rate of convergence of Armijo linear for the truly stochastic setting. By assuming interpolation, you are still in the stochastic setting, but it's the setting where SGD achieves the same rates as full batch gradient descent, so its *almost* like analysing the full batch case (but not really). A clearer title that would better fit the contributions would be

`` Armijo Line-search Makes Gradient Descent Go Fast: Rates for Full batch and Stochastic Interpolation''

But this is just a suggestion, the title you have now is ok.


[Yuki2024] Yuki Takezawa and Han Bao and Ryoma Sato and Kenta Niwa and Makoto Yamada, Parameter-free Clipped Gradient Descent Meets Polyak, Neurips 2024

[Vankov2025] Daniil Vankov and Anton Rodomanov and Angelia Nedich and Lalitha Sankar and Sebastian U Stich,  Optimizing \$(L\_0, L\_1)\$-Smooth Functions by Gradient Methods,
ICLR, 2025

[Yuan et all 2022] ] Yuan, R., Gower, R. M., & Lazaric, A. (2022). A general sample complexity analysis of vanilla policy gradient. Proceedings of The 25th International Conference on Artificial Intelligence and Statistics, PMLR 151, 3332–3380

[Agarwal et al., 2021] Agarwal, A., Kakade, S. M., Lee, J. D., & Mahajan, G. (2021). On the Theory of Policy Gradient Methods: Optimality, Approximation, and Distribution Shift. Journal of Machine Learning Research, 22(98), 1-76

[Mei et al., 2020] J Mei, C Xiao, C Szepesvari, D Schuurmans. On the global convergence rates of softmax policy gradient methods
 - ICML 2020

[Garrigos2022] Guillaume Garrigos, Robert M. Gower, Fabian Schaipp, Function Value Learning: Adaptive Learning Rates Based on the Polyak Stepsize and Function Splitting in ERM, 2022

[Garrigos2023] Handbook of Convergence Theorems for (Stochastic) Gradient Methods
Guillaume Garrigos, Robert M. Gower, 2023